# Decentralized Bandits without Global Clock for Dynamic Matching Market

**Mengtong Gao** [* 1] **Zhenhe Zhang** [* 1] **Jichen Li** [1] **Wentao Zhou** [1] **Xuanzhi Xia** [1] **Jing Chen** [1]

## Abstract

Two-sided matching markets are pervasive in numerous real-world applications, ranging from labor markets to online advertising. A rich line of research has studied the matching bandit problem, where participants learn their preferences through iterative interactions. However, existing works assume a static environment with fixed participants and require synchronized learning, in which all participants start simultaneously and have access to a global clock. In reality, matching markets are inherently dynamic: participants may enter and leave at arbitrary time steps without any global signal, creating coordination challenges. To study the dynamic setting, we first investigate one-sided learning under uncoordinated player arrivals, where only the players need to learn their preferences. We propose the Way-SE algorithm, which achieves a regret of $O(\frac{K^2 \log T}{\Delta_{\min}^2})$, where $K$ is the number of arms, $T$ is the time horizon, and $\Delta_{min}$ is the minimum utility gap. This is done through a distributed exploration mechanism that coordinates exploration implicitly via just local clocks. More importantly, we extend our work to fully decentralized dynamic two-sided learning, where both sides need to learn their preferences, and players arrive or depart arbitrarily. We introduce Way-SE-2S, the first algorithm to achieve sublinear regret $O\left(\frac{K T^{1-1/K} (\log T)^{2/K}}{\Delta_{\min}^2}\right)$ in this challenging environment, without requiring global signals, restrictive preference structures, or observability of the results of competing agents. Our work provides the first theoretical guarantee for stable matching in fully decentralized and uncoordinated bandit markets.

## 1. Introduction

Two-sided matching is a fundamental framework with wide-ranging applications in modern economic markets, such as labor markets (Roth, 1984a; Mehta et al., 2007), school admissions (Abdulkadiroğlu & Sönmez, 2003), online advertising (Varian, 2007), two-sided recommendation platforms (Li et al., 2024), etc. The classical literature (Roth & Sotomayor, 1992; Gale & Shapley, 1962; Roth, 2002) focuses on how to generate a stable outcome, while more recent studies (Das & Kamenica, 2005; Liu et al., 2020; 2021; Sankararaman et al., 2021; Kong et al., 2022; 2024; Zhang & Fang, 2024) started investigating the important *matching bandit* problem. In the latter, participants on one or both sides must learn their unknown preferences about the other side through iterative interactions and stochastic rewards. In a static setting with a fixed set of players and arms, existing studies were able to design online learning algorithms to reach a stable matching, while almost minimizing the regret.

However, in many applications, matching markets are inherently dynamic: players may enter and leave the market arbitrarily without giving any global signal (Doval, 2022; Ashlagi et al., 2023; Akbarpour et al., 2020). For instance, in labor markets or crowdsourcing platforms, applicants (players) frequently enter and exit the market while companies (arms) remain long-term. Because these arrivals and departures occur without any centralized schedule, there is no shared global clock to align their actions. This leads to a decentralized learning process where participants initiate preference learning at different, uncoordinated time steps, raising an important question: Is it possible to reach a stable matching with minimum regret under such decentralized, uncoordinated learning conditions?

This uncoordinated dynamic setting introduces fundamental challenges in both exploration and exploitation, which render existing state-of-the-art algorithms (Kong & Li, 2023; Zhang et al., 2022; Kong et al., 2024; Zhang & Fang, 2024; Wang & Li, 2024) inapplicable. For exploration, the uncoordinated arrivals break the common assumption of phased exploration (e.g., round-robin or explore-then-commit), as participants are at different stages of their learning progress. Indeed, all existing decentralized protocols rely on players being "synchronized" to avoid collisions: they enter the market at the same time and start learning in the same

---

*Equal contribution [1]Department of Computer Science and Technology, Tsinghua University, Beijing, China. Correspondence to: Jichen Li <jichenli@tsinghua.edu.cn>, Jing Chen <jchencs@tsinghua.edu.cn>.

*Proceedings of the 43rd International Conference on Machine Learning*, Seoul, South Korea. PMLR 306, 2026. Copyright 2026 by the author(s).

round. In particular, a global clock is publicly known to all participants and leveraged by the algorithm. Without this, collisions become pervasive and persistent. For exploitation, the arbitrary departure of players implies that the optimal stable matching is not a fixed target but a changing one. In static markets, once a stable matching is found, players can commit to their choices and maintain them indefinitely. However, in the dynamic setting, if a high-ranking player leaves, the entire stable-matching structure may shift. Algorithms that stop exploring after an initial convergence period will inevitably suffer a linear regret: without external signals, they fail to detect and adapt to the new equilibrium of the system.

To address these challenges, we first tackle the exploration problem by investigating the one-sided learning scenario under dynamic player arrivals, where there is no departure and only players need to learn their preferences about arms. For players learning with $K$ arms in time horizon $T$, we propose the Way-SE algorithm that achieves a regret of $O(\frac{K^2 \log T}{\Delta_{\min}^2})$ for player-optimal stable matching (POSM) in the dynamic setting, where $\Delta_{\min}$ denotes the minimum utility gap among arms for the players. Our key technical contribution lies in a novel distributed exploration mechanism that enables participants to only rely on their local clocks and de facto coordinate their exploration phases without global synchronization, effectively resolving collisions even when players arrive arbitrarily at different time steps.

Next and more importantly, we extend our study to the most general and complex case: a fully decentralized, dynamic two-sided learning setting. In this scenario, both players and arms need to learn their preferences about the other side, and players may enter or leave at any time, without knowing the global time or the actions or rewards of others, and without sending or receiving any public signal. We introduce the Way-SE-2S algorithm, which achieves a sublinear regret of $O\left( \frac{KT^{1-1/K}(\log T)^{2/K}}{\Delta_{\min}^2} \right)$ with respect to POSM, for arbitrary sequences of player arrivals and departures, again without using a global clock for synchronization. To our best knowledge, this is the first learning algorithm to achieve a sublinear regret in this challenging environment. To overcome the challenges brought by player departures, we introduce a novel re-exploration mechanism termed "Restless Ping"[1], which is specifically designed to restore the matching structure after departure-induced structural shifts. It inherently empowers a decentralized market to continuously track and recover the evolving POSM under uncoordinated population dynamics. Ultimately, our work provides the first theoretical guarantee for stable matching in fully decentralized and uncoordinated bandit markets, making a genuine step in this hard open problem and building a new bridge between

___
[1]This is not the "restless bandit" model in the literature. Rather, it highlights the way how our algorithm works.

matching bandit theory and real-world dynamic systems.

The rest of the paper is organized as follows. Section 2 discusses related work. We then introduce the two-sided dynamic matching market model in Section 3. Section 4 presents the Way-SE algorithm and Section 5 presents the Way-SE-2S algorithm, as well as their theoretical guarantees. In Section 6, we empirically evaluate our algorithms in various dynamic matching market scenarios and compare their performance against state-of-the-art algorithms. Due to page limits, proofs and additional experimental results are deferred to the Appendix in the Supplemental Material.

## 2. Related Work

**Decentralized Bandit Learning in Matching Markets.** Existing literature on decentralized matching bandits is primarily categorized by the feedback available to players. Earlier works (Liu et al., 2021; Kong et al., 2022; Kong & Li, 2023) assume a full observation setting where the entire matching outcome is revealed, allowing players to achieve a logarithmic regret. In the more restrictive partial observation setting, where players only observe their own matching outcome, coordination must be achieved either implicitly or explicitly. Implicit coordination typically relies on specific market structures, such as serial dictatorship or $\alpha$-conditions (Sankararaman et al., 2021; Basu et al., 2021; Maheshwari et al., 2022), with Wang & Li (2024) recently establishing optimal regret bounds under these assumptions. To accommodate general markets, recent advances introduce explicit protocols that emulate the Gale-Shapley algorithm (Gale & Shapley, 1962), such as ML-ETC (Zhang et al., 2022) and Adaptive Online GS (Kong et al., 2024), which utilize phase-based synchronization to secure collision-free samples.

**Dynamic Matching Markets.** Research on dynamic environments has largely focused on reward non-stationarity, where players' and arms' preferences evolve over time. Proposed solutions include restart strategies (Muthirayan et al., 2023) and successive elimination protocols such as DNCB (Ghosh et al., 2022). However, these methods assume a static set of agents and fail to account for population dynamics (entry/exit), where the market structure itself is fluid. As highlighted by Li et al. (2025), designing algorithms for markets with structurally changing populations remains a critical open challenge. Our work fills this gap by considering arbitrary player arrivals and departures, a setting significantly more complex than simple reward drift.

**Two-sided Unknown Preferences.** In markets where preferences are unknown to both sides, achieving stable matching becomes a moving target due to the coupled learning processes of all participants. While recent data-driven formulations attempt to learn these uncertain preferences from

historical data (Dai & Jordan, 2021), their analytical frameworks are restricted to static environments with synchronized timelines. Current theoretical approaches, such as CA-ETC (Pagare & Ghosh, 2023) and Round-Robin ETC (Zhang & Fang, 2024), also rely on synchronized, epoch-based strategies to identify the POSM. Although recent studies explore uncoordinated learning via probabilistic methods (Pokharel & Das, 2023; Etesami & Srikant, 2025) or extended Gale-Shapley variants (Basu, 2025), they predominantly assume a fixed participant set with a synchronized starting clock. Parallel to these algorithmic constraints, existing lower bounds (Sankararaman et al., 2021) are also for synchronized settings, leaving theoretical lower bounds under dynamic environments entirely unknown. In contrast, our work breaks these assumptions by addressing a fully decentralized market characterized by population dynamics and uncoordinated learning start times.

## 3. The Model

This paper considers decentralized bandit learning in two-sided dynamic matching markets over a time horizon $T$. Time is discrete, and there is an abstract global clock only for the purpose of problem description and analysis, whereas the value of the clock is unknown to the players or arms. Players enter the market in an uncoordinated way and may leave at any time. Throughout time, the market has $N$ players in total and $K$ fixed arms, denoted by $\mathcal{N} = \{p_0, p_1, \ldots, p_{N-1}\}$ and $\mathcal{K} = \{a_0, \ldots, a_{K-1}\}$. Here $K$ is publicly known while $N$ is unknown to the players or arms. For each player $p_i$, it has an enter time $t_i^{in}$ and leave time $t_i^{out}$, such that $1 \le t_i^{in} < t_i^{out}$. If it never leaves, then $t_i^{out} \triangleq T + 1$. Player $p_i$ pulls arms at time steps $t \in [t_i^{in}, t_i^{out})$. That is, $p_i$ no longer pulls any arm in the time step where it leaves.

At each global time $t \in [1, T]$, the set of active players is $\mathcal{N}_t \triangleq \{p_i | t_i^{in} \le t < t_i^{out}\} \subseteq \mathcal{N}$. To formalize the eventual stabilization of the player set, we define the *global stable time* $T_{\text{stable}} \in [1, T]$ as the earliest time such that the active player set is finally stable: that is, $\forall p_i \in \mathcal{N}, t_i^{in} \le T_{\text{stable}}$ and we have that either $t_i^{out} \le T_{\text{stable}}$ or $t_i^{out} = T + 1$. Thus $\mathcal{N}_t = \mathcal{N}_{T_{\text{stable}}}$ for all $t \ge T_{\text{stable}}$.

Importantly, neither $T_{\text{stable}}$ nor the time steps when other players enter or leave is known to a player. Upon entering, a player $p_i$ only has a local clock $t_i$ to count its own decisions, which starts from 1 at time $t_i^{in}$. Indeed, $p_i$ doesn't know $t_i^{in}$, $t_i^{out}$, or how long itself will be in the system either. Following prior works (Liu et al., 2020; 2021; Sankararaman et al., 2021; Basu et al., 2021; Kong et al., 2022; Zhang et al., 2022), we assume $1 \le |\mathcal{N}_t| \le K$ for all $t$, so that at every time step, the market is non-empty and each active player can be matched to an arm.

**Preferences and Utilities.** Each side of the market has a strict preference ranking over the other, represented by fixed but unknown utilities. For player $p_i$, the utility of arm $a_k$ is $\mu_{i,k} \in [0, 1]$. We define the utility gap for player $p_i$ as $\Delta_i \triangleq \min_{k_1, k_2 \in \mathcal{K}, k_1 \ne k_2} |\mu_{i,k_1} - \mu_{i,k_2}|$. For arm $a_k$, the utility of player $p_i$ is $\pi_{k,i} \in [0, 1]$, and the utility gap for arm $a_k$ is $\Delta_k^{(a)} \triangleq \min_{i_1, i_2 \in \mathcal{N}, i_1 \ne i_2} |\pi_{k,i_1} - \pi_{k,i_2}|$. Following previous works (Liu et al., 2020; 2021), we assume $\min_i \Delta_i > 0$ and $\min_k \Delta_k^{(a)} > 0$, and denote the minimum gaps for the player side and the arm side as $\Delta_{\min}^p \triangleq \min_{i \in \mathcal{N}} \Delta_i$ and $\Delta_{\min}^a \triangleq \min_{k \in \mathcal{K}} \Delta_k^{(a)}$, respectively. The definition of the global minimum utility gap $\Delta_{\min}$ then adapts to the specific learning setting: in Setting I (Section 4), where arms do not learn, the complexity relies solely on the player side, so $\Delta_{\min} \triangleq \Delta_{\min}^p$; whereas in the fully decentralized Setting II (Section 5), convergence is constrained by the hardest instance on either side, thus $\Delta_{\min} \triangleq \min\{\Delta_{\min}^p, \Delta_{\min}^a\}$. Both $\mu_{i,k}$ and $\pi_{k,i}$ are initially unknown and must be learned through repeated interactions. Note that $\Delta_{min}$ is used in the algorithm only to simplify the latter's description. In Appendix F we discuss how to remove it: the algorithm becomes more complex without significantly new techniques.

Following literature convention (Zhang & Fang, 2024; Kong et al., 2024), the players propose and the arms accept. In each round $t$, each active player $p_i \in \mathcal{N}_t$ proposes to an arm $A_i(t) \in \mathcal{K}$. Each arm $a_k$ collects the set of proposers and accepts the one according to its learning algorithm. We denote the player successfully matched to arm $a_k$ at time $t$ as $I_k(t)$. When a match is successful between player $p_i$ and arm $a_k$ at time $t$, player $p_i$ will receive a stochastic utility $X_i(t) \in [0, 1]$ with $\mathbb{E}[X_i(t)] = \mu_{i,k}$, and arm $a_k$ will receive a stochastic utility $Y_k(t) \in [0, 1]$ with $\mathbb{E}[Y_k(t)] = \pi_{k,i}$. The player rejected by the arm or the arm with no one proposing at time $t$ will receive utility $X_i(t) = 0$ and $Y_k(t) = 0$ at that round. Players do not observe any information about the actions or utilities of others.

**Stability and Regrets.** Stability is the central solution concept in two-sided markets (Gale & Shapley, 1962; Roth, 1984b; Roth & Sotomayor, 1992; Gusfield & Irving, 1989). At any time $t$, a matching is *stable* if all active players are accepted by their proposed arms and there exists no player-arm pair $(p_i, a_k)$ that *blocks* it —meaning that both prefer each other over their current partners, i.e., $\mu_{i,k} > \mu_{i,A_i(t)}$ and $\pi_{k,i} > \pi_{k,I_k(t)}$.

With respect to an arbitrary sequence of entering and leaving actions of the players, let $\mathcal{M}_t$ denote the set of all stable matchings at time $t$. Among them, the *player-optimal stable matching* (POSM), denoted as $m_t^*$, yields the highest utility for every active player $p_i \in \mathcal{N}_t$ simultaneously. The objective of the proposed algorithms is to drive the players to converge toward the POSM. A player $p_i$'s cumulative regret

with respect to the POSMs through time is defined as

$$R_i(T) \triangleq \mathbb{E}[\sum_{t \in [t_i^{in}, t_i^{out})} \mu_{i,m_t^*(i)} - X_i(t)]. \tag{1}$$

Here, the expectation is taken over the randomness in $p_i$'s utility generation and the players' randomized strategies according to the learning algorithm.

As typical in the literature, our learning algorithms try to minimize each player's cumulative regret, and we are interested in regret upper-bounds that are sublinear in $T$, so that the time-averaged regret goes to $0$ as $T$ grows. By doing so, the individualized regret minimization collectively drives the decentralized system to track the dynamic POSM target. Of course, our algorithms work in the dynamic decentralized setting and each player runs the algorithm individually, without any global signal or any observation about others.

## 4. Algorithm for One-sided Learning with Arrival-Only Players

To address the fundamental challenges of uncoordinated learning in dynamic matching markets, in this section, we first investigate the *one-sided* learning problem under uncoordinated player *arrivals.* Here, only players need to learn their preferences while arms have fixed, known preferences. Also, the players never leave, so $t_i^{out} = T + 1$ for all $p_i$ and is omitted in the discussion.

We develop a decentralized algorithm that enables players to systematically explore arms and identify their optimal stable matches without requiring a global clock. Our Algorithm 1, called the Way-SE (Wait-and-Yield with Stable Elimination) algorithm, combines a cyclic exploration mechanism with an elimination strategy to achieve an $O(\frac{K^2 \log T}{\Delta_{\min}^2})$ regret in this environment.

### 4.1. Algorithm Overview

The algorithm operates in two phases at each time step $t$: a *State Update* phase and an *Exploration/Exploitation* phase. In the state update phase, each player $p_i$ first computes confidence bounds for all arms based on historical observations, then decides whether to explore or exploit. Then, in the exploration/exploitation phase, the player proposes to the target arm, observes the utility, and updates the statistical estimates.

**State Update.** At the beginning of each round $t$, player $p_i$ computes the upper/lower confidence bounds (UCB/LCB) for each arm $a_k$ based on historical observations, which consist of the number of successful pulls $N_{i,k}(t)$ and the empirical mean $\hat{\mu}_{i,k}(t)$. More specifically, let $\delta_g \in (0, 1)$ denote the maximum tolerable global failure probability

---

**Algorithm 1** Way-SE (from the view of player $p_i$)

1: **Input:** Arm set $\mathcal{K} = \{a_k\}_{k=0}^{K-1}$.
2: **Init:** $\mathcal{S}_i \leftarrow \mathcal{K}$, $ptr_i \sim \mathcal{U}(K)$.
3: **Init:** $\hat{\mu}_{i,k} \leftarrow 0$, $N_{i,k} \leftarrow 0$, $\text{UCB}_{i,k}(0) \leftarrow 1$, $\text{LCB}_{i,k}(0) \leftarrow 0, \forall k \in \{0, \ldots, K-1\}$.
4: **for** $t = 1, \ldots, T$ **do**
5:    // State Update
6:    Update $\text{UCB}_{i,k}(t)$ and $\text{LCB}_{i,k}(t)$ for all $a_k \in \mathcal{K}$ by Equations 3 and 4;
7:    Let $k^* \leftarrow \arg\max_{k:a_k \in \mathcal{S}_i} \text{LCB}_{i,k}(t)$;
8:    **if** $\text{LCB}_{i,k^*}(t) > \max_{k:a_k \in \mathcal{S}_i \setminus \{a_{k^*}\}} \text{UCB}_{i,k}(t)$ **then**
9:      **Pull** $a_{k^*}$;       //Exploitation
10:      **if** $X_i(t) > 0$ **then**
11:        $\hat{\mu}_{i,k^*} \leftarrow \frac{X_i(t)+N_{i,k^*}\hat{\mu}_{i,k^*}}{N_{i,k^*}+1}$; $N_{i,k^*} \leftarrow N_{i,k^*} + 1$;
12:      **else**
13:        $\mathcal{S}_i, \hat{\mu}_{i,k^*}, N_{i,k^*}, t \leftarrow$
14:        $\textsc{WaitingArm}(a_k^*, \mathcal{S}_i, \hat{\mu}_{i,k^*}, N_{i,k^*}, t)$;
15:      **end if**
16:    **else**
17:      **Pull** $a_{ptr_i}$;       //Exploration
18:      **if** $X_i(t) > 0$ **then**
19:        $\hat{\mu}_{i,ptr_i} \leftarrow \frac{X_i(t)+N_{i,ptr_i}\hat{\mu}_{i,ptr_i}}{N_{i,ptr_i}+1}$;
20:        $N_{i,ptr_i} \leftarrow N_{i,ptr_i} + 1$;
21:        $ptr_i \leftarrow (ptr_i + 1) \mod K$;
22:      **else**
23:        **if** $a_{ptr_i} \in \mathcal{S}_i$ **then**
24:          $\mathcal{S}_i, \hat{\mu}_{i,ptr_i}, N_{i,ptr_i}, t \leftarrow$
25:          $\textsc{WaitingArm}(a_{ptr_i}, \mathcal{S}_i, \hat{\mu}_{i,ptr_i}, N_{i,ptr_i}, t)$;
26:        **else**
27:          $ptr_i \leftarrow (ptr_i + 1) \mod K$;
28:        **end if**
29:      **end if**
30:    **end if**
31: **end for**

---

across all players, arms, and time steps. To control the cumulative error via the union bound, we allocate this total budget over both arms and time. Namely, for each time step $t$, we define a time-dependent failure probability budget $\delta_t \triangleq \frac{6\delta_g}{\pi^2 K^2 t^2}$. Following the definition of self-normalized martingale bound (Abbasi-Yadkori et al., 2011), the confidence radius is defined as:

$$\text{Rad}_{i,k}(t, \delta_t) \triangleq$$

$$\frac{\sqrt{1 + N_{i,k}(t)}}{2N_{i,k}(t)} \cdot \sqrt{1 + 2\log\left(\frac{K\sqrt{1 + N_{i,k}(t)}}{\delta_t}\right)}. \tag{2}$$

When $N_{i,k}(t) = 0$ (no historical samples), the radius is defined as $\text{Rad}_{i,k}(t, \delta_t) = 1$.

Based on the empirical mean $\hat{\mu}_{i,k}(t)$ and the confidence radius $\text{Rad}_{i,k}(t, \delta_t)$ computed from historical information,

---

**Algorithm 2** WAITINGARM (for player $p_i$)

---
1: **Input:** Target arm $a_k, \mathcal{S}_i, \hat{\mu}_{i,k}, N_{i,k}, t$.
2: **Init:** $c_{wait} \leftarrow 0$.
3: **for** $\tau = 1, \ldots, K$ **do**
4:   Pull $a_k$;
5:   **if** $X_i(\tau) > 0$ **then**
6:     Update $\hat{\mu}_{i,k}, N_{i,k}$;
7:     **Return** $\mathcal{S}_i, \hat{\mu}_{i,k}, N_{i,k}, t + \tau$
8:   **else**
9:     $c_{wait} \leftarrow c_{wait} + 1$;
10:   **end if**
11: **end for**
12: **if** $c_{wait} \geq K$ **then**
13:   $\mathcal{S}_i \leftarrow \mathcal{S}_i \setminus \{a_k\}$;                    //Elimination
14: **end if**
15: **Return** $\mathcal{S}_i, \hat{\mu}_{i,k}, N_{i,k}, t + K$

---

player $p_i$ computes the confidence bounds:

$$\text{UCB}_{i,k}(t) \triangleq \min\{1, \hat{\mu}_{i,k}(t) + \text{Rad}_{i,k}(t, \delta_t)\}, \quad (3)$$

$$\text{LCB}_{i,k}(t) \triangleq \max\{0, \hat{\mu}_{i,k}(t) - \text{Rad}_{i,k}(t, \delta_t)\}. \quad (4)$$

Using these confidence bounds, $p_i$ first picks an arm $a_{k^*}$ that has the highest LCB among the remaining candidate set $\mathcal{S}_i$, i.e., $k^* = \arg\max_{k:a_k \in \mathcal{S}_i} \text{LCB}_{i,k}(t)$. If the confidence intervals are sufficiently separated so that $\text{LCB}_{i,k^*}(t) > \max_{k:a_k \in \mathcal{S}_i \setminus \{a_{k^*}\}} \text{UCB}_{i,k}(t)$, then the algorithm exploits $a_{k^*}$; otherwise, it follows the cyclic exploration algorithm (Line 17 of Algorithm 1).

**Index-Free Cyclic Exploration.** The key challenge in uncoordinated learning is that players enter and explore at different, unsynchronized times, so naive exploration strategies can lead to persistent collisions. Our index-free cyclic exploration algorithm addresses this by letting each player maintain a local pointer $ptr_i$, initialized uniformly at random from $\{0, \ldots, K-1\}$, and advance it deterministically modulo $K$. Thus, as long as two players' pointers are offset from each other, they keep sweeping through all arms without colliding in exploration.

When a collision occurs and player $p_i$ got rejected (i.e., $X_i(t) = 0$), $p_i$ does not immediately give up, but instead continues to propose the same arm for up to $K$ consecutive rounds while maintaining a waiting-time counter $c_{\text{wait}}$ that increments up to $K$ (Algorithm 2). Here, the $K$-round waiting window spans a full exploration cycle: temporary explorers will vacate the arm within $K$ rounds. Therefore, if the arm yields zero utility throughout the entire $K$-round waiting period, $p_i$ deduces that a superior player is persistently occupying this arm, eliminates it from $\mathcal{S}_i$, and resumes the cyclic scan. Otherwise, if $p_i$ obtains positive utility before the $K$-round timeout, it concludes that the arm

is not occupied by a superior player and retains it within the cyclic exploration process.

**Exploitation with Elimination.** To find the POSM, the algorithm initializes each player $p_i$'s candidate set to the full arm set $\mathcal{S}_i \leftarrow \mathcal{K}$. During exploitation, the player targets the arm $a_{k^*}$ with the highest LCB among remaining candidates. Similar to the exploration part, if rejected enough times, then $p_i$ permanently eliminates $a_{k^*}$ from $\mathcal{S}_i$. The elimination strategy is safe because any eliminated arm, although preferred by the player, cannot belong to the POSM: the arm is persistently occupied by some player whom the arm prefers over $p_i$, so this player would block any matching between $p_i$ and $a_{k^*}$. Combined with the fact that newly entering players can only cause each existing player's assignment unchanged or worse, this ensures that the eliminated arms are precisely those unattainable to $p_i$ in the POSM.

### 4.2. Regret Analysis

**Theorem 4.1.** *Following Algorithm 1, the expected cumulative regret for any player $p_i$ satisfies:*

$$\mathbb{E}[R_i(T)] \leq O\left(\frac{K^2 \log T}{\Delta_{\min}^2}\right), \quad (5)$$

*and the system converges to POSM with high probability.*

The complete proof of Theorem 4.1 is presented in Appendix A. To provide the readers with some intuition, the core idea is to decompose the cumulative regret into Learning Regret and Structural Adjustment Regret, conditioned on the validity of time-uniform concentration bounds. Our Index-Free Cyclic Exploration method ensures that any player, regardless of their arrival time or the existing population, maintains a minimum sampling frequency of $\Omega(1/K^2)$ for each candidate arm. This guarantee enables players to distinguish suboptimal arms from the optimal one within $O(\frac{K^2 \log T}{\Delta_{\min}^2})$ learning steps. For the residual structural adjustment regret, the analysis hinges on the validity of the elimination strategy: the timeout condition ($c_{wait} \geq K$, Line 12 of Algorithm 2) serves as a precise discriminator. It guarantees that even when players arrive at different times, each player eventually eliminates all infeasible arms, while the system ultimately converges to the POSM.

It is worth noting that the $\Delta_{\min}^{-2}$ dependence in Theorem 4.1 reflects a fundamental challenge inherent to asynchronous environments. Specifically, when a collision occurs, it stems from two indistinguishable situations: either the arm is only temporarily occupied by another exploring player, or it has already been stably matched with a player it prefers. Consequently, unlike in classic static settings, players cannot safely eliminate arms based solely on confidence bounds. How to establish a more refined $\Delta_{\min}^{-1}$-dependent bound under asynchronous arrivals remains an open challenge.

## 5. Algorithm for Two-sided Learning

Having analyzed the one-sided learning environment with player arrivals, we now advance to the most general and complex scenario: a fully decentralized, dynamic two-sided learning setting. In this setting, both players and arms need to learn their preferences regarding the opposing side, and the players may enter or leave at any time. The paramount difficulty here is that a player's departure may change the established matching structure, thereby incurring regret even among players who had already found their optimal arms in previous time. We overcome this challenge via a randomized algorithmic design and, for the first time, establish sublinear regret guarantees under such a fully dynamic and uncoordinated environment.

### 5.1. Algorithm Overview

Specifically, in the two-sided case, each arm maintains confidence bounds for players following the mainstream framework (Abbasi-Yadkori et al., 2011), always selecting the player that has the highest upper confidence bound. Formally, each arm $a_k$ estimates its preference for player $p_i$ using a confidence budget of $\delta_t = \frac{3\delta_g}{\pi^2 K^2 t^2}$; the bounds are defined as:

$$\text{UCB}_{k,i}^{(a)}(t) \triangleq \min\left(1, \hat{\pi}_{k,i}(t) + \text{Rad}_{i,k}\left(t, \delta_t\right)\right),$$

$$\text{LCB}_{k,i}^{(a)}(t) \triangleq \max\left(0, \hat{\pi}_{k,i}(t) - \text{Rad}_{i,k}\left(t, \delta_t\right)\right),$$

where $\hat{\pi}_{k,i}(t)$ is the empirical mean and $\text{Rad}_{i,k}\left(t, \delta_t\right)$ is as in Equation (2) with $\delta_t$ above. Each player's confidence bounds are as before, except using the new $\delta_t$ above.

Having defined the confidence intervals, we now determine the minimum number of samples required to ensure that the confidence intervals of each arm, and also those of each player, become disjoint, which is essential for correctly identifying the optimal arms. To this end, we define a dynamic sample separation threshold $L(t)$ as:

$$L(t) \triangleq \left\lceil \frac{48 \log t}{\Delta_{\min}^2} \right\rceil + 1, \tag{6}$$

where $\Delta_{\min}$ is the global minimum utility gap. We emphasize that $\Delta_{\min}$ is used here to simplify the theoretical exposition; it is neither leveraged for player synchronization nor for detecting structural market changes. The algorithm can be naturally extended to operate purely on localized information. We refer the reader to Appendix F for such a decoupled variant, where a player-arm specific threshold $L_{i,k}(t)$ is used.

For the players, our Way-SE-2S algorithm (Algorithm 3) operates in three sequential phases in each time step: State Update, Target Arm Selection, and Execution and Conflict Handling, with the core procedures of each phase defined in

---

**Algorithm 3** Way-SE-2S (from the view of player $p_i$)

1: **Input:** Arm set $\mathcal{K} = \{a_k\}_{k=0}^{K-1}$, parameters $\Delta_{\min}, \delta_g$.
2: **Global Variables:** Time step $t \leftarrow 1$, waiting counter $c_{wait} \leftarrow 0$, $\mathcal{S}_i \leftarrow \mathcal{K}$, $\mathcal{O}_i \leftarrow \emptyset$, $M_i \leftarrow \texttt{EXPLORE}$, $timer \leftarrow K^2$, $a_{\text{held}} \leftarrow \perp$, $ptr_i \leftarrow 0$.
3: **Global Variables:** $\hat{\mu}_{i,k} \leftarrow 0$, $N_{i,k} \leftarrow 0$, $\text{UCB}_{i,k} \leftarrow 1$, $\text{LCB}_{i,k} \leftarrow 0$, $\forall k \in \{0, \dots, K-1\}$.
4: **while** $t \leq T$ **do**
5:     *// Phase 1: State Update*
6:     $(\mathcal{P}_i, L(t)) \leftarrow \textsc{StateUpdate}(\mathcal{K}, \Delta_{\min}, \delta_g)$; //Alg. 4
7:     $timer \leftarrow \max(0, timer - 1)$;
8:     **if** $M_i = \texttt{EXPLORE} \wedge timer = 0$ **then**
9:         $M_i \leftarrow \texttt{EXPLOIT}$; $c_{wait} \leftarrow 0$;
10:         $a_{\text{held}} \leftarrow \arg\max_{a_k \in \mathcal{S}_i} \text{UCB}_{i,k}$;
11:     **else if** $M_i = \texttt{EXPLOIT} \wedge a_{\text{held}} \notin \mathcal{P}_i$ **then**
12:         $M_i \leftarrow \texttt{EXPLORE}$; $timer \leftarrow L(t)$; $c_{wait} \leftarrow 0$;
13:         $a_{\text{held}} \leftarrow \perp$;
14:     **end if**
15:     *// Phase 2: Target Arm Selection*
16:     $(Target, IsPing) \leftarrow \textsc{Select}(\mathcal{K}, \mathcal{P}_i)$;     //Alg. 5
17:     *// Phase 3: Execution & Conflict Handling*
18:     **Pull** $Target$; Observe $X_i(t)$;
19:     **if** $X_i(t) > 0$ **then**
20:         Update $\hat{\mu}_{i,Target}$; $N_{i,Target} \leftarrow N_{i,Target} + 1$; $c_{wait} \leftarrow 0$;
21:         **if** $IsPing$ **then**
22:             $a_{\text{held}} \leftarrow Target$;         //Ping success
23:         **end if**
24:         **if** $(M_i = \texttt{EXPLORE}) \wedge (|\mathcal{P}_i| = 1) \wedge (N_{i,Target} \geq L(t))$ **then**
25:             $a_{\text{held}} \leftarrow Target$; $M_i \leftarrow \texttt{EXPLOIT}$; // Stabilize
26:         **end if**
27:     **else**
28:         $\textsc{Conflict}(Target, IsPing, L(t), \mathcal{K})$.     //Alg. 6
29:     **end if**
30:     $t \leftarrow t + 1$;
31: **end while**

---

Algorithms 4, 5, and 6, respectively. See Appendix D.1 for a flow diagram. Variables shared across all procedures are declared as global variables, while other variables are local. In particular, each player $p_i$ computes four arm sets, which are updated during the algorithm:

- Candidate Set $\mathcal{S}_i$: Arms believed to be available, initialized as $\mathcal{K}$.
- Potential Set $\mathcal{P}_i$: Arms that cannot yet be statistically ruled out as optimal, defined as $\mathcal{P}_i \triangleq \{a_k \in \mathcal{S}_i \mid \text{UCB}_{i,k} \geq \max_{a_j \in \mathcal{S}_i} \text{LCB}_{i,j}\}$.
- Omitted Set $\mathcal{O}_i$: Arms temporarily eliminated due to conflict.
- Dream Set $\mathcal{D}_i$: Arms in $\mathcal{S}_i \cup \mathcal{O}_i$ that are possibly better than current held arm $a_{\text{held}}$.

**Algorithm 4** STATEUPDATE

1: **Input:** Arm set $\mathcal{K}$, parameters $\Delta_{\min}, \delta_g$.
2: **Global Input:** Candidate set $\mathcal{S}_i$, current time $t$.
3: $\delta_t \leftarrow \frac{3\delta_g}{\pi^2 K^2 t^2}$; $L(t) \leftarrow \lceil \frac{48}{\Delta_{\min}^2} \log t \rceil + 1$;
4: **for** $a_k \in \mathcal{K}$ **do**
5:    **if** $N_{i,k} > 0$ **then**
6:       $\text{Rad}_{i,k} \leftarrow \frac{1}{2}\sqrt{\frac{1+N_{i,k}}{N_{i,k}^2}\left(1 + 2\log\left(\frac{K\sqrt{1+N_{i,k}}}{\delta_t}\right)\right)}$;
7:       $\text{UCB}_{i,k} \leftarrow \min(1, \hat{\mu}_{i,k} + \text{Rad}_{i,k})$;
8:       $\text{LCB}_{i,k} \leftarrow \max(0, \hat{\mu}_{i,k} - \text{Rad}_{i,k})$;
9:    **else**
10:       $\text{UCB}_{i,k} \leftarrow 1$; $\text{LCB}_{i,k} \leftarrow 0$;
11:    **end if**
12: **end for**
13: $\text{LCB}_{\max} \leftarrow \max_{j \in \mathcal{S}_i} \text{LCB}_{i,j}$;
14: $\mathcal{P}_i \leftarrow \{a_k \in \mathcal{S}_i \mid \text{UCB}_{i,k} \geq \text{LCB}_{\max}\}$;
15: **Return** $\mathcal{P}_i, L(t)$.

**State Update.** This phase is governed by two key variables: a sample threshold $L(t)$ and a $timer$. In each time step, the player first invokes STATEUPDATE to refresh the confidence bounds and construct the potential arm set $\mathcal{P}_i$, and subsequently decreases the timer. If the player is currently in the exploration state ($M_i = \text{EXPLORE}$) and the timer reaches zero, the player transits to the exploitation state and holds onto the empirically best arm $a_{\text{held}}$. On the other hand, if a player in the exploitation state finds that its held arm $a_{\text{held}}$ has dropped out of the potential set $\mathcal{P}_i$ due to statistical updates, it reverts to the exploration state, resets the timer $timer = L(t)$ and releases $a_{\text{held}}$.

**Target Arm Selection.** In this phase, player $p_i$ determines a target arm $Target$ by executing SELECT (Algorithm 5). When $p_i$ is in the exploration state with the waiting counter $c_{wait} = 0$, the player updates its pointer $ptr_i$ to cyclically pull arms within $\mathcal{P}_i$ (Line 4–6).

Conversely, if the player is in the exploitation state, the procedure first constructs the dream set $\mathcal{D}_i$. Under normal conditions, the player maintains its current position by targeting $a_{\text{held}}$ with $IsPing \leftarrow$ False. However, to handle unobserved departures and prevent players from being trapped in a suboptimal stable matching, the player deviates to initiate a *Restless Ping* mechanism with a dynamic probability $p(t)$, defined as:

$$p(t) = \frac{1}{t^{1/K}(\log t)^{(K-2)/K}}. \tag{7}$$

Specifically, this mechanism selects a random arm from $\mathcal{D}_i$ as the target and reactivates it from the omitted set $\mathcal{O}_i$ to the candidate set $\mathcal{S}_i$ (Lines 13–17).

This design allows players to re-evaluate previously elimi-

nated options and escape from outdated beliefs about matching arms. Notably, multiplying $p(t)$ by any constant factor $c > 0$ (such that $p(t) < 1$) preserves the same asymptotic regret bound. A detailed analysis of $p(t)$'s alternative forms is provided in Appendix C.

**Algorithm 5** SELECT

1: **Input:** Arm set $\mathcal{K}$, Potential Set $\mathcal{P}_i$.
2: **Global Input:** $\mathcal{S}_i$, $\mathcal{O}_i$, $a_{\text{held}}$, $t$, $ptr_i$, $c_{wait}$, $M_i$, $\text{UCB}_{i,k}$.
3: **if** $M_i = \text{EXPLORE}$ **then**
4:    **if** $(c_{wait} = 0) \vee (a_{ptr_i} \notin \mathcal{P}_i)$ **then**
5:       $d^* \leftarrow \min\{d \geq 1 \mid a_{(ptr_i+d)\,(\text{mod } K)} \in \mathcal{P}_i\}$;
6:       $ptr_i \leftarrow (ptr_i + d^*)\,(\text{mod } K)$; $c_{wait} \leftarrow 0$;
7:    **end if**
8:    **Return** $(a_{ptr_i}, \text{False})$.
9: **else**
10:    *// 1. Construct Dream Set (Potential Upgrades)*
11:    $\mathcal{D}_i \leftarrow \{a_k \in (\mathcal{S}_i \cup \mathcal{O}_i) \mid \text{UCB}_{i,k} > \text{UCB}_{i,a_{\text{held}}}\}$;
12:    *// 2. Probabilistic Trigger for Restless Ping*
13:    $p(t) \leftarrow \frac{1}{t^{1/K}(\log t)^{(K-2)/K}}$;
14:    **if** $\mathcal{D}_i \neq \emptyset \wedge \text{Rand}(\mathcal{U}(0,1)) < p(t)$ **then**
15:       $Target \leftarrow RandomSample(\mathcal{D}_i)$;
16:       $\mathcal{S}_i \leftarrow \mathcal{S}_i \cup \{Target\}$; $\mathcal{O}_i \leftarrow \mathcal{O}_i \setminus \{Target\}$;
17:       **Return** $(Target, \text{True})$.      *// Initiate Ping*
18:    **end if**
19:    **Return** $(a_{\text{held}}, \text{False})$.
20: **end if**

**Execution and Conflict Handling.** In this phase, player $p_i$ proposes to the $Target$ arm and observes whether the proposal is accepted, as well as the utility $X_i(t)$.

If accepted (i.e., $X_i(t) > 0$), the player updates its empirical information for this arm and resets the waiting counter $c_{wait}$ (Line 20 in Algorithm 3). Furthermore, if this accepted proposal was initiated via the *Restless Ping* mechanism (which can only happen when it is in exploitation), the player updates its current held position to this newly activated arm by setting $a_{\text{held}} \leftarrow Target$ (and it remains in exploitation). When the player is in the exploration state, SELECT ensures that $Target$ is from $\mathcal{P}_i$. If it is the only arm in the latter, and if the number of samples for $Target$ is large enough, player $p_i$ holds onto this unique arm by setting $a_{\text{held}} \leftarrow Target$ and transits to exploitation (Lines 24–26).

If the proposal is not accepted by the target arm, a conflict has occurred and player $p_i$ invokes CONFLICT (Algorithm 6) to handle it. First, the waiting counter $c_{wait}$ is incremented. Then, depending on the current state and the type of the target arm, CONFLICT executes the following update rules:

- If the rejected target was selected via the *Restless Ping* mechanism (under exploitation), the conflict indicates that the previously omitted arm remains unavailable.

Consequently, the player releases its currently held arm $a_{\text{held}}$, clears the counter $c_{wait}$, and returns to the exploration state with the timer reset to $L(t)$.

- If the player is in the exploitation state (but did not trigger Restless Ping), then the rejected proposal was to the currently held arm $a_{\text{held}}$, and the player will wait for at most $K$ time steps. Once $c_{wait} \geq K$, a persistent blocking by a player currently more preferred by the arm is confirmed. In this case, $a_{\text{held}}$ is removed from the candidate set $\mathcal{S}_i$ and moved to the omitted set $\mathcal{O}_i$. The player then releases $a_{\text{held}}$ and returns to exploration with $timer \leftarrow L(t)$. In addition, the candidate set $\mathcal{S}_i$ is reset if it becomes empty.

- Finally, if a rejection occurs while the player is in the exploration state, the counter $c_{wait}$ continues to increment. The player waits at this position for at most $K$ time steps. Once $c_{wait} \geq K$, it is reset to 0. This reset ensures that in the next time step, SELECT successfully advances the cyclic pointer to sample alternative arms.

---

**Algorithm 6** CONFLICT

1: **Input:** $Target, IsPing, L(t)$, arm set $\mathcal{K}$.
2: **Global Input:** $\mathcal{S}_i, \mathcal{O}_i, a_{\text{held}}, c_{wait}, M_i, timer$.
3: $c_{wait} \leftarrow c_{wait} + 1$;
4: **if** $IsPing$ **then**
5:     $a_{\text{held}} \leftarrow \perp; c_{wait} \leftarrow 0; M_i \leftarrow$ EXPLORE;
6:     $timer \leftarrow L(t)$;           // Ping failed: Reset
7: **else if** $M_i =$ EXPLOIT $\wedge c_{wait} \geq K$ **then**
8:     $\mathcal{S}_i \leftarrow \mathcal{S}_i \setminus \{a_{\text{held}}\}; \mathcal{O}_i \leftarrow \mathcal{O}_i \cup \{a_{\text{held}}\}$;
9:     $a_{\text{held}} \leftarrow \perp; M_i \leftarrow$ EXPLORE; $timer \leftarrow L(t)$;
10:     **if** $\mathcal{S}_i = \emptyset$ **then**
11:         $\mathcal{S}_i \leftarrow \mathcal{K}; \mathcal{O}_i \leftarrow \emptyset$;     // Candidate Reset
12:     **end if**
13: **else if** $M_i =$ EXPLORE $\wedge c_{wait} \geq K$ **then**
14:     $c_{wait} \leftarrow 0$;
15: **end if**

---

### 5.2. Regret Analysis

**Theorem 5.1.** *Following Algorithm 3, the expected cumulative regret for any player $p_i$ satisfies*

$$\mathbb{E}[R_i(T)] \leq O\left(\frac{KT^{1-1/K}(\log T)^{2/K}}{\Delta_{\min}^2}\right). \quad (8)$$

The complete proof of Theorem 5.1 is in Appendix B. Intuitively, we condition the analysis on the high-probability event that the time-uniform confidence bounds hold on both the player side and the arm side. Under this event, the cumulative regret is decomposed into the following three components. (1) Statistical learning regret, incurred while learning the preferences of players and arms. (2) POSM-reaching regret, which measures the utility loss before the market first reaches POSM after the population stabilizes.

And (3) restless exploration regret, induced by the Restless Ping mechanism, which is the most difficult to analyze and ultimately determines the upper-bound in the theorem.

## 6. Simulation

We empirically validate the guarantees of our algorithms: Way-SE under player arrivals, whereas Way-SE-2S under both player arrivals and departures.

### 6.1. Experimental Setup

Simulations are conducted for $T \in \{2 \times 10^5, 2.5 \times 10^5\}$ steps with sub-Gaussian utilities with variance $\sigma^2 = 0.5$. During the experiment, the Restless Ping rate $p(t)$ is scaled by a constant $c = 0.1$ by Appendix C. The means of agent utilities are mostly generated by sampling uniformly from $[0.1, 0.4]$. In addition, to simulate preferred matches with high-utility partners, for each agent (player or arm, whenever applicable), the means of two randomly selected agents from the other side are uniformly drawn from $[0.6, 0.8]$ and $[0.8, 0.95]$, respectively. Meanwhile, a minimum utility gap of $\Delta_{\min} = 0.01$ is maintained in the main body of the paper.

### 6.2. Way-SE: Logarithmic Regret

We evaluate the Way-SE algorithm in a dynamic market initialized with 15 arms and 6 players at time 1. New players arrive at time $t \in \{20k, 60k, 80k, 100k, 150k\}$ (where k = $10^3$) to simulate a dynamic player arrival.

As shown in Figure 1, the cumulative regret (blue line) closely tracks the $O(\log T)$ benchmark (red line), validating the upper-bound in Theorem 4.1. Specifically, despite the disruptions caused by new player arrivals (the vertical green dashed lines), the system exhibits strong resilience by rapidly reconverging to the new POSM after a brief, temporary increase in regret. Appendix E.1 in addition demonstrates the instantaneous regret, which increases upon player arrivals and quickly goes back to 0 compared with POSM.

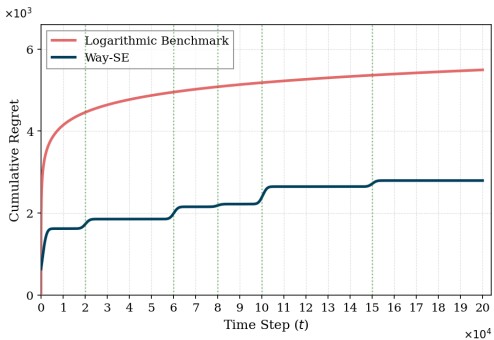

*Figure 1.* Way-SE: The cumulative regret tracks the $O(\log T)$ benchmark under one-sided learning with 15 arms.

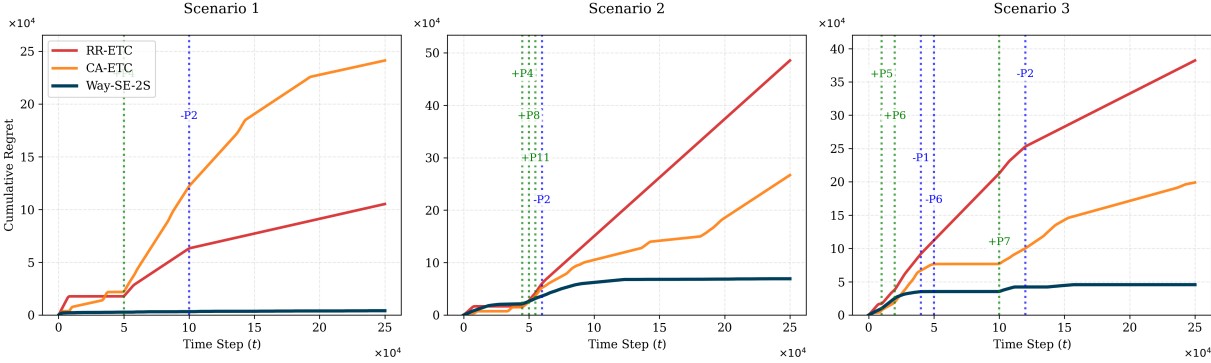

*Figure 2.* Way-SE-2S: The cumulative regret across three scenarios with $K = 15$, in comparison with two baseline algorithms.

### 6.3. Way-SE-2S: Sublinear Regret

We first evaluate the Way-SE-2S algorithm in a dynamic market initialized with 15 arms and 3 players at time 1. We consider 3 scenarios:

1. A new player enters at $t = 50k$ and a different player exits at $t = 100k$: a setting with a large time gap between arrival and departure events.
2. Three player entries at $t \in \{45k, 50k, 55k\}$, followed by a different player's departure at $t = 60k$: a setting with fast events before the system converges.
3. Player entries at $t \in \{10k, 20k, 100k\}$ and different player exits at $t \in \{45k, 50k, 120k\}$: a volatile market with intervening entries and exits.

In each scenario, our algorithm is compared with two static baseline algorithms: RR-ETC (Zhang & Fang, 2024) and Epoch-based CA-ETC (Pagare & Ghosh, 2023).

Figure 2 demonstrates that for all scenarios, our algorithm achieves sublinear cumulative regret, significantly outperforming baseline algorithms. Furthermore, the results show that our algorithm adapts to both player arrivals and departures (blue dashed lines): in each scenario, the system converges to the newly established POSM following the events. In contrast, even though the baseline algorithms initially converge faster prior to a population change, the uncoordinated arrivals breaks the synchronization and the departures shifts the POSM. As a result, the systems become permanently trapped in suboptimal matchings, causing their cumulative regrets to diverge. Appendix E.2 further shows the instantaneous regrets for the three scenarios.

**Scalability and Sensitivity Simulation.** To evaluate the robustness of Way-SE-2S in macroscopic environments, Appendix E.4 considers larger market sizes: (1) $K = 100$ and $N_{\text{init}} = 20$, and (2) $K = 200$ and $N_{\text{init}} = 30$. Dynamic entries and exits are modeled via Poisson processes to generate volatile events, and multiple arrivals or departures may occur simultaneously. For each market

size, we investigate parameter sensitivity by considering $\Delta_{\min} \in \{0.0005, 0.001, 0.002\}$. The empirical results robustly corroborate our theoretical analysis, sustaining sublinear regret growth even in volatile dynamic settings.

## 7. Conclusion and Future Directions

This work addresses the critical challenge of arbitrary arrival and departure sequences in decentralized matching markets. We introduce *Way-SE*, the first algorithm for one-sided learning under dynamic arrivals. Using the new Index-Free Cyclic Exploration technique, the proposed algorithm achieves self-organized coordination without relying on a global clock. It achieves a logarithmic regret, approaching the lower-bound for static markets (Sankararaman et al., 2021). More importantly, for the general two-sided learning setting involving both arrival and departure dynamics, we propose the *Way-SE-2S* algorithm, which leverages the new Restless Ping mechanism to handle arbitrary departures and achieves the first sublinear regret in this hard setting.

**Future Directions.** Firstly, we conjecture that the $T^{1-1/K}$ dependence in the general two-sided setting is actually almost tight, and an answer to this conjecture in either direction will be very interesting. Secondly, it would be nice to collect real-world data in dynamic environments and conduct data-driven studies under our model. Several possible extensions to our model are also interesting, such as to incorporate incentive compatibility for individual behaviors beyond regret-minimization. Also, it is worth studying how to move beyond the fixed stable time $T_{\text{stable}}$, when arrivals and departures occur in a decelerating way but continue forever. While our current algorithm remains applicable when the deceleration speed exceeds the convergence speed, any outpaced scenario will require new algorithms with accelerated convergence. Finally, extending the current framework to simultaneously accommodate reward non-stationarity (e.g., drifting preferences) alongside population dynamics is another practical consideration for real-world environments.

## Acknowledgements

The authors would like to thank several anonymous reviewers and the area chair for their valuable comments.

## Impact Statement

This paper advances the study of decentralized matching bandits, especially in multi-agent learning environments where agents arrive and leave asynchronously and no global coordination signal is available. This work provides the first algorithmic guarantee for stable matching in a fully decentralized, uncoordinated market, with significant potential impact across a wide range of real-world applications. The main practical relevance of this work lies in applicant-driven matching settings, such as decentralized job application and admission-like processes, where the applicants in the market can change over time and previously stable decisions may become obsolete after hidden arrivals or departures. At the same time, the present paper is primarily theoretical: it does not involve deployment, human-subject experimentation, or immediate policy recommendations, and its broader societal impact would depend on how future systems choose to incorporate such decentralized matching mechanisms.

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

# A. Proof of Theorem 4.1

To establish the regret guarantees of Way-SE algorithm, we first characterize the statistical properties of the utility noise, which form the foundation for constructing confidence intervals and analyzing the algorithm's performance.

**Sub-Gaussianity Noise.** Let $\mathcal{F}_{t-1}$ denote the $\sigma$-algebra generated by the interaction history up to the beginning of round $t$. This represents the filtration of the stochastic process. For each player $p_i$ and time step $t$, we define the noise term as $\eta_i(t) \triangleq X_i(t) - \mu_{i,A_i(t)}$. This term captures the instantaneous deviation of the observed utility from its expected value given the chosen arm. The following lemma establishes that $\{\eta_i(t)\}_{t \geq 1}$ forms a sub-Gaussian martingale difference sequence, which allows for the application of concentration inequalities.

**Lemma A.1.** *For any player $p_i$ and time step $t$, the noise sequence $\{\eta_i(t)\}_{t \geq 1}$ is a conditionally $\frac{1}{2}$-sub-Gaussian martingale difference sequence with respect to the filtration $\{\mathcal{F}_{t-1}\}_{t \geq 1}$. That is, for all $\lambda \in \mathbb{R}$,*

$$\mathbb{E}\left[\exp(\lambda \eta_i(t)) \mid \mathcal{F}_{t-1}\right] \leq \exp\left(\frac{\lambda^2}{8}\right).$$

*Proof.* To prove the lemma, we must verify that the sequence has a zero conditional mean and its moment generating function is bounded. By definition, the expected utility of arm $A_i(t)$ is $\mu_{i,A_i(t)} = \mathbb{E}[X_i(t) \mid \mathcal{F}_{t-1}]$, as utility $X_i(t)$ is independent generated. Therefore, the conditional expectation of the noise term is:

$$\mathbb{E}[\eta_i(t) \mid \mathcal{F}_{t-1}] = \mathbb{E}[X_i(t) - \mu_{i,A_i(t)} \mid \mathcal{F}_{t-1}] = 0.$$

Thus, $\{\eta_i(t)\}_{t \geq 1}$ is a martingale difference sequence with respect to $\{\mathcal{F}_{t-1}\}_{t \geq 1}$.

Meanwhile, since the observed utilities are bounded such that $X_i(t) \in [0, 1]$, the noise term $\eta_i(t)$ is bounded within the interval $[-\mu_{i,A_i(t)}, 1 - \mu_{i,A_i(t)}]$. Hoeffding's Lemma (Lattimore & Szepesvári, 2020) states that for any zero-mean random variable $Z$ bounded in an interval $[a, b]$, the moment generating function satisfies $\mathbb{E}[e^{\lambda Z}] \leq \exp(\frac{\lambda^2(b-a)^2}{8})$. Applying this to $\eta_i(t)$, we have:

$$\mathbb{E}\left[\exp(\lambda \eta_i(t)) \mid \mathcal{F}_{t-1}\right] \leq \exp\left(\frac{\lambda^2(1)^2}{8}\right) = \exp\left(\frac{\lambda^2}{8}\right).$$

This confirms that the noise sequence is conditionally $\frac{1}{2}$-sub-Gaussian. $\square$

With the established sub-Gaussian property, we now construct confidence intervals for the true utilities and analyze their validity. To this end, we define the Good Event, which captures the scenario where all confidence intervals simultaneously contain their corresponding true utilities.

**Definition A.2** (Good Event)**.** For each time step $t \in [1, T]$, we define the Good Event at time $t$ as:

$$\mathcal{G}_t \triangleq \{\forall p_i \in \mathcal{N}_t, \forall a_k \in \mathcal{K}, |\mu_{i,k} - \hat{\mu}_{i,k}(t)| \leq \text{Rad}_{i,k}(t, \delta_t)\},$$

where $\text{Rad}_{i,k}(t)$ denotes the confidence radius for player-arm pair $(i, k)$ at time $t$. The global Good Event is defined as the intersection over all time steps, that is, $\mathcal{G} := \bigcap_{t=1}^{\infty} \mathcal{G}_t$.

Under the event $\mathcal{G}_t$, all confidence intervals are valid simultaneously, meaning $\mu_{i,k} \in [\text{LCB}_{i,k}(t), \text{UCB}_{i,k}(t)]$ for all player-arm pairs $(i, k)$. The complementary event $\neg \mathcal{G}_t$ is referred to as the Bad Event at time $t$, which occurs when at least one confidence bound fails.

The following lemma establishes that the Good Event holds with high probability, which is crucial for the regret analysis.

**Lemma A.3.** *For any time step $t \in \mathbb{N}$, the probability of the Bad Event is bounded by:*

$$\mathbb{P}(\neg \mathcal{G}_t) \leq \frac{6\delta_g}{\pi^2 t^2}.$$

*Proof.* We analyze the probability of the Bad Event by applying the Union Bound over the set of players and arms. Based on the construction of $\text{Rad}_{i,k}(t, \delta_t)$ derived from standard concentration inequalities (e.g., Hoeffding-Azuma or self-normalized

bounds (Abbasi-Yadkori et al., 2011)), the probability that the empirical mean deviates from the true mean by more than the radius for a fixed pair $(i, k)$ is bounded by $\delta_t$:

$$\mathbb{P}\left(|\hat{\mu}_{i,k}(t) - \mu_{i,k}| > \mathrm{Rad}_{i,k}(t, \delta_t)\right) \leq \delta_t = \frac{6\delta_g}{\pi^2 K^2 t^2}.$$

The Bad Event $\neg\mathcal{G}_t$ occurs if this deviation inequality holds for *any* active player $p_i \in \mathcal{N}_t$ or any arm $a_k \in \mathcal{K}$. We express this as a union of failure events:

$$\neg\mathcal{G}_t = \bigcup_{p_i \in \mathcal{N}_t} \bigcup_{a_k \in \mathcal{K}} \{|\hat{\mu}_{i,k}(t) - \mu_{i,k}| > \mathrm{Rad}_{i,k}(t, \delta_t)\}.$$

Applying the Union Bound yields:

$$\mathbb{P}(\neg\mathcal{G}_t) \leq \sum_{p_i \in \mathcal{N}_t} \sum_{a_k \in \mathcal{K}} \mathbb{P}\left(|\hat{\mu}_{i,k}(t) - \mu_{i,k}| > \mathrm{Rad}_{i,k}(t, \delta_t)\right) \leq \sum_{p_i \in \mathcal{N}_t} \sum_{a_k \in \mathcal{K}} \delta_t = |\mathcal{N}_t| \cdot K \cdot \frac{6\delta_g}{\pi^2 K^2 t^2}.$$

Since the number of active players $|\mathcal{N}_t|$ is bounded by the total number of arms $K$, we conclude:

$$\mathbb{P}(\neg\mathcal{G}_t) \leq \frac{6\delta_g}{\pi^2 t^2}.$$

This completes the proof. $\qquad\square$

Having established the validity of confidence intervals, we now determine the minimum number of samples required to ensure that the confidence intervals of different arms become disjoint, which is essential for correctly identifying the optimal arms. To this end, we define a dynamic sample separation threshold $L(t)$ as:

$$L(t) \triangleq \left\lceil \frac{48 \log t}{(\Delta^p_{\min})^2} \right\rceil + 1, \tag{9}$$

where $\Delta^p_{\min}$ is the global minimum player-side utility gap.

The following lemma guarantees that once the number of samples for candidate arms exceeds this threshold, their confidence intervals become disjoint, thereby enabling the correct ranking of arms.

**Lemma A.4.** *Under global Good Event $\mathcal{G}$, for any time $t \geq \max\{\frac{48K^2}{(\Delta^p_{\min})^2}, \frac{96}{(\Delta^p_{\min})^2} \log \frac{96}{(\Delta^p_{\min})^2} + 3, \frac{e\pi^4 K^6}{36\delta_g^2}\}$, any player $p_i \in \mathcal{N}_t$ and the best arm $a_{k^*} \in \mathcal{S}_i(t)$ can be disjoint with other arms $a_k \in \mathcal{S}_i(t) \setminus \{a_{k^*}\}$ or be eliminated within $K^2 C_L \sqrt{t + \frac{1}{4} K^4 C_L^2} + \frac{1}{2} K^4 C_L^2 + 1$ rounds, where $C_L = \frac{49}{(\Delta^p_{\min})^2}$.*

*Proof.* The set of available arms $\mathcal{S}_i(t)$ is dynamic. Whenever an arm is eliminated whether due to $K$ consecutive rejections during the EXPLOIT state or the EXPLORE state, the set $\mathcal{S}_i(t)$ shrinks immediately.

Consequently, the identity of the best arm in $\mathcal{S}_i(t)$ changes instantaneously. Let $t_{start}$ be the moment player $p_i$ is forced into the EXPLORE state (due to the removal of the previous incumbent). At this exact moment $t_{start}$, a new optimal arm is defined:

$$a_{k^*} = \arg\max_{a_k \in \mathcal{S}_i(t_{start})} \mu_{i,k}.$$

The player immediately begins the exploration process for this new optimal arm $a_{k^*}$. The sampling rate in EXPLORE state is constrained by two independent factors.

1. Player $p_i$ explores the arm set $\mathcal{K}$ via a cyclic strategy with random start, targeting $a_k \in \mathcal{S}_i$ with frequency at least $1/K$.

2. Upon proposing to $a_k$, player $p_i$ may compete with up to $K - 1$ other players in EXPLORE state. For the arms that are eliminated during the exploration, we wait at most $K$ steps; for the other arms, we wait at most $K - 1$ steps. Therefore, for each target arm in the exploration phase, we sample with frequency at least $1/K$ in each round.

Combining these factors, the worst-case sampling rate is $\eta = \frac{1}{K} \cdot \frac{1}{K} = \frac{1}{K^2}$. The EXPLORE state phase for $a_{k^*}$ terminates in one of two ways:

1. **Identification:** $a_{k^*}$ is statistically distinguished from suboptimal arms $a_k \in \mathcal{S}_i(t) \setminus \{a_{k^*}\}$ and enters in EXPLOIT state at most at time $t'$, defined and bounded below.

2. **Elimination:** $a_{k^*}$ is rejected $K$ times. Because it didn't have time to enter the EXPLOIT state phase, the time cost is strictly bounded by the maximum identification time $t' - t$.

We set $T_0$ as the solution of the following equation, which can also be seen as a function of $t$.

$$T_0 = K^2 \left(3 + \frac{48 \log(t + T_0)}{(\Delta_{\min}^p)^2}\right).$$

Then we have $T_0 < K^2 C_L \log(t + T_0) < K^2 C_L \sqrt{t + T_0}$, where $C_L = \frac{49}{(\Delta_{\min}^p)^2}$. Therefore, $(T_0 - \frac{1}{2} K^4 C_L^2)^2 < K^4 C_L^2 t + \frac{1}{4} K^8 C_L^4$,

$$\lceil T_0 \rceil < K^2 C_L \sqrt{t + \frac{1}{4} K^4 C_L^2} + \frac{1}{2} K^4 C_L^2 + 1.$$

Next we will prove that the EXPLORE state phase will terminate at most at time $\lceil T_0 \rceil + t$. To simplify the notation, we denote $t' = t + \lceil T_0 \rceil$. Since $t \geq \frac{48 K^2}{(\Delta_{\min}^p)^2}$, $f(T) = \frac{T}{K^2} - (3 + \frac{48 \log(t+T)}{(\Delta_{\min}^p)^2})$ is increasing for all $T > 0$. We have for all $a_k \in \mathcal{S}_i(t)$,

$$N_{i,k}(t') - L(t') \geq \left\lfloor \frac{\lceil T_0 \rceil}{K^2} \right\rfloor - \left\lceil \frac{48 \log t'}{(\Delta_{\min}^p)^2} \right\rceil - 1 > \frac{\lceil T_0 \rceil}{K^2} - \frac{48 \log(t')}{(\Delta_{\min}^p)^2} - 3$$

$$= f(\lceil T_0 \rceil) \geq f(T_0) = \frac{T_0}{K^2} - \frac{48 \log(t + T_0)}{(\Delta_{\min}^p)^2} - 3 = 0.$$

$f(0) < 0 = f(T_0)$, so $\lceil T_0 \rceil \geq 1$.

Based on the above condition, we further prove that $N_{i,k}(t') \geq L(t')$ is a sufficient condition for these intervals to be disjoint before $t'$. Consider two arms $a_{k^*}, a_{k'} \in \mathcal{S}_i(t)$ with a true mean gap $\Delta \geq \Delta_{\min}^p$. Under the global Good Event $\mathcal{G}$, the true means lie within their respective confidence intervals $[\hat{\mu} - \text{Rad}, \hat{\mu} + \text{Rad}]$.

Since the function $\text{Rad}_{i,k}(t')$ is decreasing in $N_{i,k}(t')$, and $N_{i,k}(t') \geq L(t')$, we have

$$\text{Rad}_{i,k^*}(t') + \text{Rad}_{i,k'}(t') \leq \sqrt{\frac{1 + L(t')}{L(t')^2} \left(1 + 2 \log\left(\frac{K \sqrt{1 + L(t')}}{\delta_{t'}}\right)\right)}$$

$$< \sqrt{\frac{2}{L(t')} \left(1 + 2 \log\left(\frac{\pi^2 K^3}{6 \delta_g}\right) + 4 \log(t') + \log(1 + L(t'))\right)}.$$

Given $t' > t \geq \max\{\frac{96}{(\Delta_{\min}^p)^2} \log \frac{96}{(\Delta_{\min}^p)^2} + 3, \frac{e \pi^4 K^6}{36 \delta_g^2}\}$, we have $t' \geq 1 + L(t')$ and $\log t' \geq 1 + 2 \log(\frac{\pi^2 K^3}{6 \delta_g})$. Therefore,

$$2\text{Rad}_{i,k^*}(t') + 2\text{Rad}_{i,k'}(t') < 2 \sqrt{\frac{12 \log t'}{L(t')}} < \Delta_{\min}^p,$$

the sum of the two confidence interval lengths is strictly less than the reward gap, two arms are disjoint.

$\square$

Thus, we ensure that the separation condition is met with high probability.

Now we begin proving that the algorithm will eventually converge to POSM. We characterize an important property of POSM with the following lemma.

**Lemma A.5.** *In a POSM, for any $k \neq m_t^*(i)$ such that $\mu_{i,k} > \mu_{i,m_t^*(i)}$, there must exists another player $p_{i'}$, $k = m_t^*(i')$ and $\pi_{k,i'} > \pi_{k,i}$.*

*Proof.* Assume $a_{m_t^*(i)}$ is the arm matched to player $p_i$ in the unique POSM. For any super-optimal arm $a_k$ such that $p_i$ strictly prefers $a_k$ to $a_{m_t^*(i)}$ (i.e., $a_k \succ_{p_i} a_{m_t^*(i)}$), suppose for contradiction that $a_k$ is either unmatched or matched to some player $p_{i'}$ with $p_i \succ_{a_k} p_{i'}$. Then $(p_i, a_k)$ forms a blocking pair, contradicting the stability of the POSM. Hence $a_k$ must be occupied by an incumbent $p_{i'}$ who satisfies $p_{i'} \succ_{a_k} p_i$. $\square$

We establish the convergence of Way-SE by proving that infeasible super-optimal arms are eventually eliminated, and the arm in POSM is never erroneously eliminated.

**Lemma A.6.** *Under the Good Event $\mathcal{G}$, for any player $p_i$ targeting arm $a_k$ over a $K$-step waiting window, the event that $c_{wait}$ reaches $K$ occurs if and only if, throughout that window, $a_k$ is continuously occupied by exploit-state incumbents who all have higher priority than $p_i$ with respect to $a_k$.*

*Proof.* Let $t_0$ be the time step when $p_i$ initiates the waiting algorithm for arm $a_k$. The event $c_{wait} = K$ implies that $a_k$ rejection $p_i$ for all $t \in [t_0, t_0 + K - 1]$.

**Sufficiency ($\Leftarrow$):** Assume arm $a_k$ is initially occupied by a stable incumbent $p_j$ ($p_j \succ_{a_k} p_i$). In the only enter setting, while $p_j$ may be displaced by a new entrant $p_{new}$, the stability condition requires $p_{new} \succ_{a_k} p_j$. By transitivity ($p_{new} \succ_{a_k} p_j \succ_{a_k} p_i$), the new incumbent is also superior to $p_i$. Since stable incumbents in the exploit state do not voluntarily abandon their optimal arms, arm $a_k$ remains continuously occupied by a chain of superior players throughout the interval. Consequently, $p_i$ receives rejection at every step $t \in [t_0, t_0 + K - 1]$, inevitably leading to $c_{wait} = K$.

**Necessity ($\Rightarrow$):** We prove this by contradiction: if arm $a_k$ is not continuously occupied by superior exploit-state incumbents during $[t_0, t_0 + K - 1]$, then player $p_i$ must successfully propose at least once in this window. Suppose no stable incumbent exists. Then any rejection at time $t \in [t_0, t_0 + K - 1]$ comes from a transient competitor $p_j \in \mathcal{N}_t$ who is in the exploration state. According to the cyclic exploration algorithm, any transient winner $p_j$ advances their pointer after success: $ptr_j(t+1) = (ptr_j(t) + 1) \pmod{K}$. For $p_j$ to block arm $a_k$ again, they must traverse the full cycle of length $K$. Thus, any specific player can win arm $a_k$ at most once within a window of $K$ steps. Let $\mathcal{P}_t$ be the set of distinct winners during the interval $[t_0, t_0 + K - 1]$. The size of the competitor pool is bounded by $|\mathcal{N}_{t'} \setminus \{p_i\}| \leq K - 1$ for all $t'$. By the pigeonhole principle, it is impossible for $K - 1$ potential competitors to generate $K$ distinct wins in consecutive steps. This contradicts the assumption that $c_{wait}$ reached $K$. $\square$

This result ensures that the elimination acts as a precise discriminator for the presence of a superior stable incumbent. Therefore, let $\mathcal{N}_{T_{\text{stable}}}$ denote the final set of active players after the global stable time $T_{\text{stable}}$, and $m^*$ be the POSM for the final set of active players. We have the following theorem.

**Theorem A.7.** *There exist a finite convergence round number $T_{con}$, such that When $t > T_{stable} + T_{con}$, all players enter the* EXPLOIT *state targeting the POSM arm $a_{m^*(i)}$, with probability at least $1 - \delta_g$.*

*Proof.* The proof relies on the fact that under the Good Event $\mathcal{G}$, the estimated confidence intervals validly bound the correct means. We analyze the evolution of $\mathcal{S}_i$ relative to the final stable state defined by $\mathcal{N}_{T_{\text{stable}}}$.

Completeness: we first show that $a_{m^*(i)}$ is never eliminated, even during the transient phase $t < T_{\text{stable}}$. In the only entering setting, the set of active competitors grows monotonically ($\mathcal{N}_t \subseteq \mathcal{N}_{T_{\text{stable}}}$). By definition, $a_{m^*(i)}$ is the stable match for $p_i$ against the full competition $\mathcal{N}_{T_{\text{stable}}}$. This implies $a_{m^*(i)}$ is not blocked by any superior player $p_j \in \mathcal{N}_{T_{\text{stable}}}$ (otherwise $p_i$ could not be stably matched to $a_{m^*(i)}$). Since the set of potential blockers at any time $t$ is a subset of $\mathcal{N}_{T_{\text{stable}}}$, $a_{m^*(i)}$ remains strictly unblocked by a superior player for all $t$. Applying Lemma A.6, the timeout condition is never triggered for $a_{m^*(i)}$. Thus, $a_{m^*(i)}$ is permanently retained in $\mathcal{S}_i$.

Soundness: consider any arm $a_k$ such that $a_k \succ_{p_i} a_{m^*(i)}$. By Lemma A.5, within the final population $\mathcal{N}_{T_{\text{stable}}}$, arm $a_k$ is strictly blocked by a exploiting player $p_j$ ($p_j \succ_k p_i$). Under global Good Event $\mathcal{G}$, Lemma A.4 guarantees that player $p_j$ will eventually learn the true values and enter the EXPLOIT state on arm $a_k$. Then, applying Lemma A.6, the persistent presence of $p_j$ guarantees that $p_i$'s waiting counter $c_{wait}$ reaches $K$. Thus, $p_i$ correctly identifies $a_k$ as infeasible and eliminates it from $\mathcal{S}_i$.

Consider the system state at $t > T_{\text{stable}}$. Based on the Soundness argument, all super-optimal arms are eliminated in turn. Based on the Completeness argument, $a_{m^*(i)}$ is retained. Thus, the available set $\mathcal{S}_i(t)$ converges such that $a_{m^*(i)}$ becomes the unique optimal arm:

$$a_{m^*(i)} = \arg \max_{a \in \mathcal{S}_i(t)} \mu_{i,a}.$$

By inductively using Lemma A.4, the total number of rounds after $T_{stable}$ for player $p_i$ to accumulate sufficient samples such that the confidence intervals for $m^*(i)$ become disjoint from any remaining suboptimal arms can be bounded by a constant related to $T_{stable}$. The algorithm satisfies the condition to enter the EXPLOIT state targeting $a_{m^*(i)}$. Therefore, by the recursion function theory (Cutland, 1980), there exist a finite round number $T_{con}$, such that for any $t > T_{\text{stable}} + T_{con}$, with at least $1 - \delta_g$ probability that the Good Event happened, all players converge to POSM.     □

We consider regret under good and bad events, respectively.

**Lemma A.8.** *The total expected regret $R^{bad} \triangleq \sum_{t=1}^{T} \mathbb{E}[r_t \mathbf{1}_{\mathcal{G}_t^c}]$ incurred due to Bad Event $\neg\mathcal{G}$ is bounded by $R^{bad} < \delta_g$.*

*Proof.* By Lemma A.3 and the convergence of the Riemann zeta function $\zeta(2) = \sum_{t=1}^{\infty} \frac{1}{t^2} = \frac{\pi^2}{6}$, we have

$$\mathbb{P}(\neg\mathcal{G}) \le \sum_{t=1}^{T} \mathbb{P}(\neg\mathcal{G}_t) \le \sum_{t=1}^{T} \frac{6\delta_g}{\pi^2 t^2} < \delta_g.$$

Since the maximum regret incurred at any single step is bounded by 1,

$$R_{\text{bad}} = \sum_{t=1}^{T} \mathbb{E}[r_t \mathbf{1}_{\mathcal{G}_t^c}] \le \sum_{t=1}^{T} \mathbb{P}(\mathcal{G}_t^c) < \delta_g.$$

□

**Combining the auxiliary results.**  We now prove Theorem 4.1 by assembling the guarantees above. Lemma A.3 and Lemma A.8 control the regret from confidence failures. Under the Good Event, Lemma A.4 bounds the cost of learning enough samples to separate feasible arms, while Lemma A.5, Lemma A.6, and Theorem A.7 show that infeasible super-optimal arms are eliminated and the process converges to the POSM. Thus it remains to bound the learning regret and the residual elimination regret.

*Proof.* We decompose the total regret into two components: the learning regret $R^{learn}$, which is incurred while statistically distinguishing suboptimal arms, and the residual regret $R^{Res}$, which is incurred during the elimination process for infeasible arms. By the process of the algorithm, it has:

$$R_i(T) = R^{\text{Learn}} + R^{\text{Res}}.$$

From Lemma A.4, we know that to distinguish a suboptimal arm $a_k$ from the optimal arm $a_{k^*}$, the algorithm requires at most $L(\lceil T_0(t) \rceil) < L(T)$ samples. Let $C_L = \frac{49}{(\Delta_{\min}^p)^2}$, the explicit bound is:

$$L(T) < C_L \log T.$$

We use $\Delta_{\min}^p$ here rather than the player's gap $\Delta_i$ because the waiting time of $p_i$ is not determined only by its own statistical difficulty. Even if $p_i$ can rank its arms quickly, an arm relevant to $p_i$ may still be temporarily occupied by a stronger player whose own learning problem is harder. Such transient blocking slows down the effective samples collected by $p_i$. The uniform threshold based on $\Delta_{\min}^p = \min_j \Delta_j$ gives a single horizon that safely covers both $p_i$'s own learning and the slowest possible blocker.

Due to the Index-Free Cyclic Exploration algorithm, player $p_i$ does not sample arms arbitrarily but follows a fixed cyclic order. In the worst-case scenario, obtaining one effective sample from arm $a_k$ requires traversing the entire cycle of length $K$. Thus, the system time $T_{\text{spend}}(k)$ required to accumulate $L(T)$ samples is dilated by a factor of $K$, bounded by:

$$T_{\text{spend}}(k) < K \cdot L(T) = KC_L \log T.$$

Meanwhile, the player must statistically identify and prune all suboptimal arms $a_k \in \mathcal{S}_i \setminus \{a_{m^*(i)}\}$. The regret accumulated from arm $a_k$ is the product of its utility gap $\Delta_{i,k}$ and the time spent evaluating it $T_{\text{spend}}(k)$. Summing over all suboptimal arms:

$$R^{\text{Learn}} = \sum_{k \neq a_{m^*(i)}} \Delta_{i,k} \cdot T_{\text{spend}}(k) \leq \sum_{k \neq a_{m^*(i)}} T_{\text{spend}}(k) < K^2 C_L \log T. \tag{10}$$

The component $R^{\text{Res}}$ incurred when player $p_i$ attempts to access an arm $a_k$ occupied by a superior stable incumbent. The upper bound for this cost is derived based on the following logic:

- Due to the monotonicity of the active player set process $\mathcal{N}_t$ in this setting, player $p_i$ eliminates at most $K - 1$ arms throughout the horizon.

- Each elimination is triggered after $K$ consecutive rejections.

- During these $K$ waiting periods per elimination, the player receives zero reward. Consequently, the instantaneous regret is bounded by the expected reward of the optimal arm, i.e., $\mu_{i,a_{m^*(i)}} \leq 1$.

- By Theorem A.7, under good event, after at most $K$ number of elimination, the system converges to POSM.

By aggregating these factors, the total structural cost $R^{\text{Res}}$ is bounded by the product of the maximum eliminations, the rejection threshold, and the maximal instantaneous regret:

$$R^{\text{Res}} \leq K \cdot K \cdot \mu_{i,a_{m^*(i)}} \leq K^2. \tag{11}$$

Combining Eq. (10) and Eq. (11) and Lemma A.8, the total expected regret for player $p_i$ is upper bounded by:

$$\mathbb{E}[R_i(T)] \leq \mathbb{P}(\mathcal{G}) \left[ R^{\text{Learn}} + R^{\text{Res}} \right] + R^{\text{bad}} < K^2 C_L \log T + K^2 + \delta_g = O\left( \frac{K^2 \log T}{\Delta_{\min}^2} \right).$$

Thus Theorem 4.1 holds. $\qquad\square$

## B. Proof of Theorem 5.1

### B.1. Learning Convergence Analysis

In this section, after $T_{\text{stable}}$ is reached, we analyze the time horizon at which every player in $\mathcal{N}_{T_{\text{stable}}}$ has accumulated enough samples to distinguish all arms. We call this state cognitive stabilization and define its time as the last time at which some such player-arm pair remains under-sampled:

$$T_{cog} \triangleq \sup\{t \in \mathbb{N} \mid \exists p_i \in \mathcal{N}_{T_{\text{stable}}}, \exists a_k \in \mathcal{K}, \ N_{i,k}(t) < L(t)\}.$$

Because the sample counts $N_{i,k}(t)$ are generated by the algorithm's randomized exploration process, $T_{cog}$ is treated as a random variable throughout the analysis. Time is counted from $T_{\text{stable}}$, and the player set is fixed as $\mathcal{N}_{T_{\text{stable}}}$. In applying the sampling lower bound below, we use it only for player-arm comparisons that are decision-relevant under Algorithm 3. Accordingly, subsequent uses of "for any/every player-arm pair" in the sampling and cognitive-stabilization arguments are understood in this decision-relevant sense; this is only a proof convention and does not modify the algorithm.

We further show that, under these conditions, the system converges to POSM as the algorithm continues to run. To begin with, we introduce a classical inequality that will be used in the following proofs.

**Definition B.1** (Two-Sided Good Event). Set $\delta_t = \frac{3\delta_g}{\pi^2 K^2 t^2}$, where $\delta_t$ is the two-sided failure budget for a single preference estimate; that is, the absolute-value confidence failure of one player-side or arm-side estimate is at most $\delta_t$. We define the player-side Good Event at time $t$, denoted as $\mathcal{G}_t^{(p)}$, and the arm-side Good Event $\mathcal{G}_t^{(a)}$ as follows:

$$\mathcal{G}_t^{(p)} := \{\forall p_i, \forall a_k, |\mu_{i,k} - \hat{\mu}_{i,k}(t)| \leq Rad\left(N_{i,k}(t), \delta_t\right)\},$$
$$\mathcal{G}_t^{(a)} := \{\forall a_k, \forall p_i, |\pi_{k,i} - \hat{\pi}_{k,i}(t)| \leq Rad\left(N_{i,k}(t), \delta_t\right)\}.$$

The Good Event at time $t$ is $\mathcal{G}_t := \mathcal{G}_t^{(p)} \cap \mathcal{G}_t^{(a)}$, and the global Good Event $\mathcal{G}$ is defined as the intersection of these events over the entire horizon: $\mathcal{G} := \bigcap_{t=1}^{\infty} \mathcal{G}_t$. Since there are at most $2K^2$ player-side and arm-side estimates at each time $t$, the union bound gives

$$\mathbb{P}(\mathcal{G}_t^c) \leq 2K^2 \delta_t = \frac{6\delta_g}{\pi^2 t^2}, \qquad \mathbb{P}(\mathcal{G}^c) \leq \sum_{t=1}^{\infty} \mathbb{P}(\mathcal{G}_t^c) \leq \delta_g.$$

Under event $\mathcal{G}$, the true utilities satisfy $\mu_{i,k} \in [\text{LCB}_{i,k}(t), \text{UCB}_{i,k}(t)]$ (and symmetrically for arms) for all interaction steps.

**Lemma B.2** (Inversion Inequality). *For any constant $Q \geq 3$, if $x \geq 2Q \log Q$, then the inequality $\frac{x}{\log x} \geq Q$ always holds.*

*Proof.* Let $f(x) = \frac{x}{\log x}$. The first derivative is given by $f'(x) = \frac{\log x - 1}{\log^2 x}$. For $x > e$, $f'(x) > 0$, implying that $f(x)$ is strictly increasing. Given $Q \geq 3$, set $x_Q = 2Q \log Q$. Then $x_Q \geq 2 \cdot 3 \log 3 > e$, so $f(x)$ is monotonically increasing for all $x \geq x_Q$. To prove the lemma, it suffices to verify the boundary condition $f(x_Q) \geq Q$. Substituting $x_Q = 2Q \log Q$, we have:

$$f(x_Q) = \frac{2Q \log Q}{\log(2Q \log Q)} = \frac{2Q \log Q}{\log 2 + \log Q + \log \log Q}.$$

Dividing both numerator and denominator by $\log Q$ yields:

$$f(x_Q) = Q \cdot \frac{2}{1 + \frac{\log 2 + \log \log Q}{\log Q}}.$$

For $Q \geq 3$, the term $\frac{\log 2 + \log \log Q}{\log Q}$ is strictly less than 1, which implies $f(x_Q) > Q$. By the monotonicity of $f(x)$, the inequality $\frac{x}{\log x} \geq Q$ holds for all $x \geq x_Q$. $\square$

We now establish a lower bound for the expected cumulative sample count $\Lambda_{i,k}(t) \triangleq \sum_{\tau=1}^{t} \mathbb{E}[X_{i,k}(\tau) \mid \mathcal{F}_{\tau-1}]$ for any player $p_i$ and arm $a_k$. This in turn enables us to prove that the actual number of samples $N_{i,k}(t)$ exceeds $\frac{1}{2}\Lambda_{i,k}(t)$ with high probability.

Throughout the subsequent analysis, the time-dependent sample separation threshold is

$$L(t) = \left\lceil \frac{48 \log t}{\Delta_{\min}^2} \right\rceil + 1, \quad C_L \triangleq \frac{49}{\Delta_{\min}^2}.$$

In what follows, we abuse notation slightly and write $L(t) = C_L \log t$, since the ceiling and max in the definition only shift $L(t)$ by an additive constant, which is irrelevant for all asymptotic statements below. Whenever a player pings a particular arm from its Dream Set $\mathcal{D}_i(t)$, we use the lower bound $1/|\mathcal{D}_i(t)| \geq 1/K$.

**Lemma B.3** (Effective Sample Accumulation). *The cumulative expected samples $\Lambda_{i,k}(t)$ satisfy the following lower bounds:*

1. *When $K = 2$, for all $t \geq T_2 \triangleq 8C_L^2 \left[\log(\sqrt{2}C_L)\right]^2$:*

$$\Lambda_{i,k}(t) \geq \frac{C_L}{8\sqrt{2}} \left((\log t)^2 - (\log T_2)^2\right),$$

2. *When $K > 2$, for all $t \geq 4$ with:*

$$\Lambda_{i,k}(t) \geq \frac{K^{2-K}}{K-2}(\log t)^{-\frac{2(K-2)}{K}} \left(t^{1-2/K} - 4^{1-2/K}\right).$$

*Meanwhile, with probability at least $1 - \exp\left(-\frac{3}{28}\Lambda_{i,k}(t)\right)$, the realized cumulative sample satisfies $N_{i,k}(t) \geq \frac{1}{2}\Lambda_{i,k}(t)$.*

*Proof.* For each $\tau \in [1, t]$, let $q_{i,k}(\tau) = \mathbb{E}[X_{i,k}(\tau) \mid \mathcal{F}_{\tau-1}]$ be the conditional success probability at time $\tau$. We analyze the lower-bound for $\Lambda_{i,k}(t)$ using the Restless Ping rate $p(\tau) = \tau^{-1/K}(\log \tau)^{-\frac{K-2}{K}}$.

**Case 1** ($K = 2$)**:** When $K = 2$, the logarithmic exponent becomes $\frac{K-2}{K} = 0$. Therefore, the Restless Ping rate simplifies exactly to $p(\tau) = \tau^{-1/2}(\log \tau)^0 = \tau^{-1/2}$. A success event is lower bounded by player $p_i$ Restless Ping the arm $a_k$, and

the incumbent player yields arm $a_k$ at least once within the contention window $[\tau, \tau + L(\tau)]$. The probability that the incumbent yields is:

$$P_{\text{yield}} = 1 - \prod_{s=\tau}^{\tau+L(\tau)} (1 - p(s)) \geq 1 - \exp\left(-\sum_{s=\tau}^{\tau+L(\tau)} p(s)\right).$$

By the definition of $T_2$, we have $C_L \log \tau \leq \sqrt{2\tau}$ for all $\tau \geq T_2$. This condition is used in two ways. First, it implies $L(\tau) = C_L \log \tau \leq \tau$, hence $\tau + L(\tau) \leq 2\tau$ and $p(\tau + L(\tau)) \geq (2\tau)^{-1/2}$. Second, it directly gives $(C_L \log \tau) \cdot (2\tau)^{-1/2} \leq 1$. Lower-bounding the sum by the minimum rate within the window gives

$$\sum_{s=\tau}^{\tau+L(\tau)} p(s) \geq L(\tau) \cdot p(\tau + L(\tau)) \geq (C_L \log \tau) \cdot (2\tau)^{-1/2}.$$

Since this quantity lies in $[0, 1]$, we apply $1 - e^{-x} \geq x/2$ (valid for $x \in [0, 1]$) to obtain, for all $\tau \geq T_2$,

$$P_{\text{yield}} \geq \frac{C_L}{2\sqrt{2}} \tau^{-1/2} \log \tau.$$

Combining this with the probability $p(\tau)/K = \tau^{-1/2}/2$ that player $i$ pings arm $a_k$, the joint success probability is:

$$q_{i,k}(\tau) \geq \frac{p(\tau)}{K} \cdot P_{\text{yield}} \geq \frac{\tau^{-1/2}}{2} \cdot \left(\frac{C_L}{2\sqrt{2}} \tau^{-1/2} \log \tau\right) = \frac{C_L}{4\sqrt{2}} \frac{\log \tau}{\tau}.$$

Integrating this probability over time $t$, we have for all $t \geq T_2$:

$$\Lambda_{i,k}(t) \geq \int_{T_2}^{t} \frac{C_L}{4\sqrt{2}} \frac{\log x}{x} dx = \frac{C_L}{8\sqrt{2}} \left((\log t)^2 - (\log T_2)^2\right).$$

**Case 2 ($K > 2$):** The success sample probability $q_{i,k}(\tau)$ is lower bounded by the joint event where player $p_i$ initiates a Restless Ping to arm $a_k$, the current player $p_j$ utilizes arm $a_k$ to ping other arms at the same time, and no other player proposes arm $a_k$; since each other player proposes $a_k$ with probability at most $p(\tau)$, this last event has probability at least $(1 - p(\tau))^{K-2}$. We have:

$$q_{i,k}(\tau) \geq \frac{p(\tau)}{K} p(\tau)(1 - p(\tau))^{K-2} = \frac{1}{K} \tau^{-2/K} (\log \tau)^{-\frac{2(K-2)}{K}} \left(1 - \tau^{-1/K} (\log \tau)^{-\frac{K-2}{K}}\right)^{K-2}.$$

For $\tau \geq 4$ and $K \geq 2$, it holds that $\log \tau > 1$, hence $p(\tau) \leq \tau^{-1/K} \leq 4^{-1/K} \leq 1 - 1/K$. Thus, the interference term is strictly lower-bounded by a constant: $(1 - p(\tau))^{K-2} \geq (1/K)^{K-2} = K^{-(K-2)}$. Substituting this into the integral to compute the expected cumulative samples:

$$\Lambda_{i,k}(t) \geq \int_{4}^{t} q_{i,k}(\tau) d\tau \geq K^{-(K-1)} \int_{4}^{t} \tau^{-2/K} (\log \tau)^{-\frac{2(K-2)}{K}} d\tau.$$

Since the logarithmic penalty term $(\log \tau)^{-\frac{2(K-2)}{K}}$ is monotonically decreasing for $\tau \geq 4$, we can extract it at the upper limit $\tau = t$ to obtain a strict closed-form lower bound:

$$\Lambda_{i,k}(t) \geq K^{-(K-1)} (\log t)^{-\frac{2(K-2)}{K}} \int_{4}^{t} \tau^{-2/K} d\tau$$

$$= K^{-(K-1)} (\log t)^{-\frac{2(K-2)}{K}} \left[\frac{K}{K-2} \tau^{\frac{K-2}{K}}\right]_{4}^{t}$$

$$= \frac{K^{2-K}}{K-2} (\log t)^{-\frac{2(K-2)}{K}} \left(t^{1-2/K} - 4^{1-2/K}\right).$$

**Concentration of $N_{i,k}(t)$.** Having established a lower bound on $\Lambda_{i,k}(t)$, we now prove a concentration lower bound for $N_{i,k}(t)$. Let $Z_{i,k}(\tau) = X_{i,k}(\tau) - q_{i,k}(\tau)$ be the martingale difference sequence, where $|Z_{i,k}(\tau)| \leq 1$. The conditional variance is bounded by $V_{i,k}(\tau) = \text{Var}(X_{i,k}(\tau) \mid \mathcal{F}_{\tau-1}) \leq q_{i,k}(\tau)$. Hence, the total variance is bounded by the mean:

$\sum V_{i,k}(\tau) \le \Lambda_{i,k}(t)$. Applying the lower-tail form of Freedman's inequality with increment bound $b = 1$, variance proxy $v = \Lambda_{i,k}(t)$, and deviation level $\lambda = \frac{1}{2}\Lambda_{i,k}(t)$:

$$P\left(\sum_{\tau=1}^{t} Z_{i,k}(\tau) \le -\frac{1}{2}\Lambda_{i,k}(t)\right) \le \exp\left(-\frac{\lambda^2}{2(v + b\lambda/3)}\right)$$

$$= \exp\left(-\frac{(\Lambda_{i,k}(t)/2)^2}{2(\Lambda_{i,k}(t) + \Lambda_{i,k}(t)/6)}\right)$$

$$= \exp\left(-\frac{3}{28}\Lambda_{i,k}(t)\right).$$

$\square$

Since $N_{i,k}(t) = \Lambda_{i,k}(t) + \sum Z_{i,k}(\tau)$, this implies $N_{i,k}(t) \ge \frac{1}{2}\Lambda_{i,k}(t)$ with high probability. Therefore, to guarantee enough realized samples, it remains to identify a deterministic time after which this high-probability lower bound already exceeds the sample separation threshold $L(t)$. The stochastic fluctuation has been controlled by the concentration bound above, so the next step is the deterministic comparison between $\frac{1}{2}\Lambda_{i,k}(t)$ and $L(t)$.

We use $T_{alg}$ to denote a deterministic sufficient crossing threshold. Its role is to mark a time after which the high-probability sample lower bound dominates the sample separation threshold:

$$T_{alg} \triangleq \inf\{ t_0 \ge 4 \mid \forall \text{ integer } t > t_0, \ \tfrac{1}{2}\Lambda_{i,k}(t) > L(t) \}.$$

**Lemma B.4.** *An explicit sufficient value of the deterministic sufficient crossing threshold is*

$$T_{alg} = \begin{cases} \exp\left(\max\{\sqrt{2}\log T_2, \ 32\sqrt{2}\}\right) & \text{if } K = 2, \\ \max\left\{4 \cdot 2^{\frac{K}{K-2}}, \ (2\tilde{Q}\log\tilde{Q})^{\frac{3K-4}{K-2}}\right\} & \text{if } K > 2, \end{cases}$$

*where $T_2 = 8C_L^2[\log(\sqrt{2}C_L)]^2$ and $\tilde{Q} \triangleq \frac{3K-4}{K-2}\left(\frac{4C_L(K-2)}{K^2-K}\right)^{\frac{K}{3K-4}}$.*

*Proof.* **Case 1** ($K = 2$): Using the lower bound from Lemma B.3, the condition $\frac{1}{2}\Lambda_{i,k}(t) > L(t)$ becomes:

$$\frac{1}{2} \cdot \left[\frac{C_L}{8\sqrt{2}}\left((\log t)^2 - (\log T_2)^2\right)\right] > C_L \log t.$$

Since $C_L > 0$, we divide both sides by $C_L \log t$ (assuming $t > 1$):

$$\frac{(\log t)^2 - (\log T_2)^2}{16\sqrt{2}\log t} > 1.$$

A sufficient way to ensure this inequality is to require $\log t \ge \sqrt{2}\log T_2$ and $\log t > 32\sqrt{2}$. Thus, it is sufficient to take

$$T_{alg}^{(K=2)} = \exp\left(\max\{\sqrt{2}\log T_2, \ 32\sqrt{2}\}\right).$$

**Case 2** ($K > 2$): Using the updated lower bound derived in Lemma B.3, we have

$$\Lambda_{i,k}(t) \ge \frac{K^{2-K}}{K-2}(\log t)^{-\frac{2(K-2)}{K}}\left(t^{\frac{K-2}{K}} - 4^{\frac{K-2}{K}}\right).$$

The crossing condition $\frac{1}{2}\Lambda_{i,k}(t) > L(t)$ becomes:

$$\frac{1}{2} \cdot \frac{K^{2-K}}{K-2}(\log t)^{-\frac{2(K-2)}{K}}\left(t^{\frac{K-2}{K}} - 4^{\frac{K-2}{K}}\right) > C_L \log t.$$

Rearranging the inequality to isolate the polynomial-logarithmic terms:

$$t^{\frac{K-2}{K}} - 4^{\frac{K-2}{K}} > \frac{2C_L(K-2)}{K^{2-K}}(\log t)^{\frac{3K-4}{K}}.$$

For $t > 4 \cdot 2^{K/(K-2)}$, we can absorb the constant $4^{\frac{K-2}{K}}$ by introducing a factor of 2. It is sufficient to require both

$$t^{\frac{K-2}{K}} > 2 \cdot 4^{\frac{K-2}{K}}$$

and

$$t^{\frac{K-2}{K}} > \frac{4C_L(K-2)}{K^{2-K}}(\log t)^{\frac{3K-4}{K}}.$$

For the second condition, raising both sides to the power of $\frac{K}{3K-4}$ yields:

$$t^{\frac{K-2}{3K-4}} > \left(\frac{4C_L(K-2)}{K^{2-K}}\right)^{\frac{K}{3K-4}} \log t.$$

Let $u = t^{\frac{K-2}{3K-4}}$, which implies $\log t = \frac{3K-4}{K-2} \log u$. Substituting this into the inequality:

$$u > \frac{3K-4}{K-2}\left(\frac{4C_L(K-2)}{K^{2-K}}\right)^{\frac{K}{3K-4}} \log u.$$

Let $\tilde{Q} = \frac{3K-4}{K-2}\left(\frac{4C_L(K-2)}{K^{2-K}}\right)^{\frac{K}{3K-4}}$. Since $\frac{3K-4}{K-2} > 3$ for $K > 2$ and the second factor is at least 1, we have $\tilde{Q} \geq 3$. By Lemma B.2, the inequality $u > \tilde{Q} \log u$ holds if $u \geq 2\tilde{Q} \log \tilde{Q}$. Substituting back $t = u^{\frac{3K-4}{K-2}}$, a sufficient threshold for the second condition is:

$$T_{alg} = (2\tilde{Q}\log\tilde{Q})^{\frac{3K-4}{K-2}}.$$

Therefore, throughout the rest of the proof we may take

$$T_{alg} = \begin{cases} \exp\left(\max\{\sqrt{2}\log T_2, \, 32\sqrt{2}\}\right) & \text{if } K = 2, \\ \max\left\{4 \cdot 2^{\frac{K}{K-2}}, \, (2\tilde{Q}\log\tilde{Q})^{\frac{3K-4}{K-2}}\right\} & \text{if } K > 2, \end{cases}$$

where $T_2$ is the finite threshold used in the $K = 2$ part of Lemma B.3, and

$$\tilde{Q} = \frac{3K-4}{K-2}\left(\frac{4C_L(K-2)}{K^{2-K}}\right)^{\frac{K}{3K-4}}.$$

$\square$

**Remark on $T_{alg}$.** The quantity $T_{alg}$ is used only as a sufficient analytical threshold in the proof. It is neither an input to Algorithm 3 nor is it intended to predict the empirical convergence time. Its role is to make the previously asymptotic crossing requirement explicit: for every integer $t > T_{alg}$, the deterministic lower bound on the realized sample count satisfies

$$\frac{1}{2}\Lambda_{i,k}(t) > L(t) = C_L \log t.$$

Thus the phrase "for sufficiently large $t$" in the proof can be understood through the displayed threshold above.

For $K > 2$, recall $\tilde{Q}$ from the definition of $T_{alg}$ above. Substituting $C_L = \frac{49}{\Delta_{\min}^2}$ into the definition of $\tilde{Q}$ gives

$$\tilde{Q} = \frac{3K-4}{K-2}\left(196(K-2)K^{K-2}\Delta_{\min}^{-2}\right)^{\frac{K}{3K-4}}.$$

Therefore, for fixed $K > 2$ and small $\Delta_{\min}$, the sufficient threshold satisfies

$$T_{alg} = O\left(\Delta_{\min}^{-\frac{2K}{K-2}}\left(\log\frac{1}{\Delta_{\min}}\right)^{\frac{3K-4}{K-2}}\right).$$

When $K$ is large and $\Delta_{\min}$ is fixed independently of $K$, the dominant factor is more transparent. In this regime,

$$\log \tilde{Q} = \frac{K}{3K-4}\Big((K-2)\log K + O(\log K)\Big) = \frac{1}{3}K \log K + O(\log K),$$

and hence

$$T_{alg} = K^{K+O(1)}(\log K)^3.$$

Equivalently,

$$\log T_{alg} = K \log K + O(K + \log K \log \log K).$$

Thus the dominant large-$K$ factor is

$$K^K = \exp(K \log K).$$

This large dependence should be interpreted as a proof-level artifact of the worst-case sample-accumulation lower bound, rather than as an empirical convergence estimate. In particular, it traces back to the conservative interference bound

$$(1 - p(t))^{K-2} \geq K^{-(K-2)},$$

used in Lemma B.3 to obtain a uniform lower bound on the effective sampling probability for every player-arm pair. The resulting $T_{alg}$ is therefore deliberately conservative: it guarantees that the tail-sum argument in Theorem 1 starts after an explicit finite time, but it is not meant to characterize the practical stabilization time. In simulations, the observed cognitive stabilization occurs much earlier than this worst-case threshold.

For $K = 2$, the gap dependence enters through the auxiliary threshold $T_2$ determined by $C_L \log t \leq \sqrt{2t}$. Substituting $C_L = 49/\Delta_{\min}^2$ and choosing an explicit sufficient $T_2$ via the inversion inequality yields $T_{alg}^{(K=2)} = O(\Delta_{\min}^{-4\sqrt{2}}(\log \frac{1}{\Delta_{\min}})^{2\sqrt{2}})$. The $K = 2$ case involves no combinatorial growth in $K$.

**Theorem B.5.** *The expected cognitive stabilization time satisfies the explicit bound*

$$\mathbb{E}[T_{cog}] \leq T_{alg} + \frac{K^2 T_{alg}^{2-\gamma}}{(\gamma-1)(\gamma-2)},$$

*with the decay rate $\gamma \triangleq \frac{3}{14}C_L \geq 10.5$.*

*Proof.* Fix any pair $(p_i, a_k)$ with $p_i \in \mathcal{N}_{T_{\text{stable}}}$ and $a_k \in \mathcal{K}$, and consider any $t > T_{alg}$.

By definition of $T_{alg}$, $\frac{1}{2}\Lambda_{i,k}(t) > L(t)$. Since $L(t) < \frac{1}{2}\Lambda_{i,k}(t)$, the event of insufficient samples is contained in the large-deviation event:

$$\{N_{i,k}(t) < L(t)\} \subseteq \{N_{i,k}(t) < \tfrac{1}{2}\Lambda_{i,k}(t)\}.$$

Applying the concentration inequality from Lemma B.3,

$$P(N_{i,k}(t) < L(t)) \leq \exp\left(-\frac{3}{28}\Lambda_{i,k}(t)\right) < \exp\left(-\frac{3}{28}(2C_L \log t)\right) = \exp\left(-\frac{3}{14}C_L \log t\right).$$

Using the identity $e^{-c \log t} = t^{-c}$ and the above definition $\gamma = \frac{3}{14}C_L$, we obtain the polynomial decay bound

$$P(N_{i,k}(t) < L(t)) < t^{-\gamma}.$$

Since $T_{cog}$ is the last time at which any pair $(p_i, a_k)$ with $p_i \in \mathcal{N}_{T_{\text{stable}}}$ is under-sampled, the event $\{T_{cog} > t\}$ means that there exists some future time $s > t$, some player $p_i \in \mathcal{N}_{T_{\text{stable}}}$, and some arm $a_k \in \mathcal{K}$ such that $N_{i,k}(s) < L(s)$. Since $|\mathcal{N}_{T_{\text{stable}}}| \leq K$, a union bound and the continuous integral test give

$$P(T_{cog} > t) \leq \sum_{s>t} \sum_{p_i \in \mathcal{N}_{T_{\text{stable}}}} \sum_{k=1}^{K} P(N_{i,k}(s) < L(s))$$

$$\leq K^2 \sum_{s>t} s^{-\gamma} \leq K^2 \int_t^\infty x^{-\gamma}dx = \frac{K^2}{\gamma-1}t^{1-\gamma}.$$

The expectation of the cognitive stabilization time is then bounded by

$$
\begin{aligned}
\mathbb{E}[T_{cog}] &\le T_{alg} + \sum_{t > T_{alg}} P(T_{cog} > t) \\
&\le T_{alg} + \frac{K^2}{\gamma - 1} \sum_{t > T_{alg}} t^{1-\gamma} \\
&\le T_{alg} + \frac{K^2}{\gamma - 1} \int_{T_{alg}}^{\infty} x^{1-\gamma} dx \\
&= T_{alg} + \frac{K^2 T_{alg}^{2-\gamma}}{(\gamma - 1)(\gamma - 2)}.
\end{aligned}
$$

$\square$

Notice that this bound depends on the problem parameters, but not on the time horizon $T$.

### B.2. Structure Convergence Analysis

We restrict our analysis to the regime $t > T_{cog}$, a stage at which the cognitive learning process has stabilized, ensuring that players separate arms with high probability. In this context, we aim to establish that the algorithm asymptotically converges to POSM, denoted by $m^*$.

We first formalize the intermediate stable matching states that the system may encounter.

**Definition B.6** (Restricted Stable Matching). A matching $m$ is defined as Restricted Stable if for all players $p_i$, there exists no blocking pair $(p_i, a_k)$ such that $p_i$ prefers $a_k$ to their current match $a_{m(i)}$, $a_k$ prefers $p_i$ over its current occupant (the player matched to $a_k$ under $m$), and $a_k \in \mathcal{S}_i$. That is, the matching is stable within the players' current visibility scope.

If a matching is Restricted Stable but not Globally Stable(without a blocked pair), it implies the existence of hidden blocking pairs located in the omitted sets $\mathcal{O}_i$. If a matching is Globally Stable but not the POSM, it implies the existence of a *Deadlock Cycle* defined as follows.

**Definition B.7** (Deadlock Cycle). Let $m$ be a stable matching. A deadlock cycle is a sequence of pairs $\rho = \{(p_0, a_0), (p_1, a_1), \ldots, (p_{\ell-1}, a_{\ell-1})\} \subseteq m$ (indices modulo $\ell$) that satisfies the structural properties:

- For every player $p_i$, the arm $a_{i+1}$ is strictly preferred to their current match $a_i$. That is:

$$a_{i+1} \succ_{p_i} a_i.$$

- For every target arm $a_{i+1}$, the current incumbent $p_{i+1}$ is preferred over the proposer $p_i$:

$$p_{i+1} \succ_{a_{i+1}} p_i.$$

In the terminology of Lattice Theory, $\rho$ constitutes a rotation which is exactly contrasted with *Exposed Rotation* (Gusfield & Irving, 1989) where proposers move down their list. Eliminating this rotation moves the system state upward in the lattice towards the Player-Optimal Stable Matching.

To establish the convergence of the system toward the POSM, we decompose the structural convergence process into the following three phases: (i) stabilization into a restricted stable state; (ii) elimination of hidden blocking pairs through ping signals; (iii) ascent through the stable matching lattice toward the POSM.

**Lemma B.8** (Convergence to Restricted Stability). *For any time $t > T_{cog}$. If no Restless Ping operations occur, the decentralized market will converge to a Restricted Stable Matching, denoted as $m_{\mathcal{S}}$. Here, $m_{\mathcal{S}}$ is stable with respect to player's the current candidate sets $\{\mathcal{S}_i\}_{i \in \mathcal{N}}$.*

*Proof.* In the absence of Restless Pings, the algorithm operates purely under exploitation logic, which is equivalent to running the simultaneous-proposal Gale-Shapley (GS) algorithm (Roth, 2008) restricted to the sets $\mathcal{S}_i$. Throughout this

process, GS maintains the invariant that the tentative matching at each proposal round is stable with respect to the proposals made so far. This allows us to compare consecutive states via the potential function

$$\Phi(t) = \sum_{p_i \in \mathcal{N}_t} \text{Rank}_{p_i}(A_i(t)),$$

here $\text{Rank}_{p_i}(a)$ denotes the preference rank of arm $a$ (1 being the best). Since the matching at each step is tentatively restricted-stable, the rank of a player's match is well-defined as their position within a valid stable matching at that stage. When a conflict occurs and player $p_i$ is rejected, $p_i$ removes the current arm from consideration and proposes to the next-best arm in $\mathcal{S}_i$, strictly increasing $\text{Rank}_{p_i}(A_i(t+1))$. Since ranks are bounded by $K$, $\Phi(t)$ is strictly monotone and necessarily converges to a restricted-stable equilibrium $m_{\mathcal{S}}$.

At this equilibrium, no player wishes to deviate to any other arm within their current $\mathcal{S}_i$ that would accept them. Thus, the resulting matching $m_{\mathcal{S}}$ satisfies the condition of Restricted Stability (Definition B.6). Furthermore, since players propose in decreasing order of preference, standard stable matching theory implies this convergence leads to the player-optimal matching among all matchings stable within $\mathcal{S}$. $\qquad \square$

**Lemma B.9** (Elimination of Hidden Blocking Pairs). *On the good event $\mathcal{G}$, if there exists a hidden blocking pair $(p_i, a_k)$ such that $a_k \in \mathcal{O}_i$, then the algorithm eliminates this blocking pair with probability at least $1 - \delta_e$ within any $\tau$ satisfying*

$$\frac{(t+\tau)^{\frac{K-1}{K}}}{(\log(t+\tau))^{\frac{K-2}{K}}} \geq \frac{t^{\frac{K-1}{K}}}{(\log t)^{\frac{K-2}{K}}} + (K-1)\ln\frac{1}{\delta_e}.$$

*Proof.* On the good event $\mathcal{G}$, since $t > T_{cog}$, the confidence intervals needed for this hidden blocking comparison have enough samples to separate true gaps of size at least $\Delta_{\min}$. Therefore, for a hidden blocking pair $(p_i, a_k)$, the lower confidence bound $\text{LCB}_{i,a_k}$ exceeds the upper confidence bound of the current match $m_t(i)$. Consequently, $a_k$ is included in player $p_i$'s Dream Set $\mathcal{D}_i$. All probabilities below are conditional on $\mathcal{G}$ and on the history up to time $t$; the remaining randomness is the algorithm's ping randomization. Let $E_s$ be the event that player $p_i$ initiates a Ping towards arm $a_k$ at time step $s$. The size of the Dream Set is bounded by $K$. Substituting the updated Restless Ping rate $p(s) = s^{-1/K}(\log s)^{-\frac{K-2}{K}}$, the probability of the event $E_s$ is bounded below by:

$$P(E_s) = p(s) \cdot \frac{1}{|\mathcal{D}_i(s)|} \geq \frac{p(s)}{K} = \frac{1}{K}s^{-1/K}(\log s)^{-\frac{K-2}{K}}.$$

We analyze the probability that the event $E_s$ *never* occurs in time interval $[t, t+\tau]$. Assuming independence of random draws across time steps, the failure probability is:

$$P(\text{fail}) = \prod_{s=t}^{t+\tau}(1 - P(E_s)).$$

Using the inequality $1 - x \leq e^{-x}$, we have:

$$P(\text{fail}) \leq \exp\left(-\frac{1}{K}\sum_{s=t}^{t+\tau} s^{-1/K}(\log s)^{-\frac{K-2}{K}}\right).$$

To solve for $\tau$, we lower-bound the summation with an integral. Since $f(s) = s^{-1/K}(\log s)^{-\frac{K-2}{K}}$ is monotonically decreasing, we have:

$$\sum_{s=t}^{t+\tau} s^{-1/K}(\log s)^{-\frac{K-2}{K}} \geq \int_t^{t+\tau} x^{-1/K}(\log x)^{-\frac{K-2}{K}}dx.$$

Because the logarithmic penalty term $(\log x)^{-\frac{K-2}{K}}$ is monotonically decreasing, we can extract its minimum value at the upper limit $x = t + \tau$ to obtain a strict algebraic lower bound for the integral:

$$\int_t^{t+\tau} x^{-1/K}(\log x)^{-\frac{K-2}{K}}dx \geq (\log(t+\tau))^{-\frac{K-2}{K}}\int_t^{t+\tau} x^{-1/K}dx$$

$$= (\log(t+\tau))^{-\frac{K-2}{K}}\frac{K}{K-1}\left((t+\tau)^{\frac{K-1}{K}} - t^{\frac{K-1}{K}}\right).$$

Substitute this integration result into the exponent of the failure probability:

$$P(\text{fail}) \leq \exp\left(-\frac{1}{K-1}(\log(t+\tau))^{-\frac{K-2}{K}}\left((t+\tau)^{\frac{K-1}{K}} - t^{\frac{K-1}{K}}\right)\right).$$

We require the probability of failure to be at most $\delta_e$:

$$\exp\left(-\frac{1}{K-1}(\log(t+\tau))^{-\frac{K-2}{K}}\left((t+\tau)^{\frac{K-1}{K}} - t^{\frac{K-1}{K}}\right)\right) \leq \delta_e.$$

Taking the natural logarithm of both sides and multiplying by $-(K-1)$:

$$(\log(t+\tau))^{-\frac{K-2}{K}}\left((t+\tau)^{\frac{K-1}{K}} - t^{\frac{K-1}{K}}\right) \geq (K-1)\ln\frac{1}{\delta_e}.$$

Distributing the denominator yields:

$$\frac{(t+\tau)^{\frac{K-1}{K}}}{(\log(t+\tau))^{\frac{K-2}{K}}} - \frac{t^{\frac{K-1}{K}}}{(\log(t+\tau))^{\frac{K-2}{K}}} \geq (K-1)\ln\frac{1}{\delta_e}.$$

Since $\log(t+\tau) \geq \log t$, we have $\frac{t^{\frac{K-1}{K}}}{(\log(t+\tau))^{\frac{K-2}{K}}} \leq \frac{t^{\frac{K-1}{K}}}{(\log t)^{\frac{K-2}{K}}}$. Thus, replacing the denominator of the subtracted term with $\log t$ makes the subtracted term larger, which strictly decreases the left-hand side. Ensuring this strictly smaller quantity satisfies the bound provides a sufficient condition:

$$\frac{(t+\tau)^{\frac{K-1}{K}}}{(\log(t+\tau))^{\frac{K-2}{K}}} - \frac{t^{\frac{K-1}{K}}}{(\log t)^{\frac{K-2}{K}}} \geq (K-1)\ln\frac{1}{\delta_e}.$$

Rearranging the terms produces the explicit threshold condition for $\tau$:

$$\frac{(t+\tau)^{\frac{K-1}{K}}}{(\log(t+\tau))^{\frac{K-2}{K}}} \geq \frac{t^{\frac{K-1}{K}}}{(\log t)^{\frac{K-2}{K}}} + (K-1)\ln\frac{1}{\delta_e}.$$

The condition derived above exhibits an elegant symmetric algebraic structure. If we define a state function $F(x) \triangleq \frac{x^{\frac{K-1}{K}}}{(\log x)^{\frac{K-2}{K}}}$, the explicit threshold condition can be concisely rewritten as $F(t+\tau) - F(t) \geq (K-1)\ln\frac{1}{\delta_e}$. This implies that the hidden blocking pair is guaranteed to be eliminated once the state function $F(\cdot)$ grows by a constant margin $C = (K-1)\ln\frac{1}{\delta_e}$ over the interval $\tau$.

Furthermore, as $x \to \infty$, the polynomial growth of the numerator $x^{\frac{K-1}{K}}$ strictly dominates the logarithmic term $(\log x)^{\frac{K-2}{K}}$ in the denominator, meaning $F(x)$ strictly diverges to positive infinity. This mathematical property provides a strong physical guarantee for the decentralized dynamics: no matter how large the current time $t$ is (and thus how low the Restless Ping rate $p(t)$ has decayed), there always exists a finite waiting time $\tau$ that satisfies this threshold. Consequently, the algorithm is guaranteed to eliminate any hidden blocking pairs in finite time, driving the system to monotonically climb upward on the stable matching lattice.

Thus, once $E_s$ occurs, since $(p_i, a_k)$ is a blocking pair, $a_k$ accepts $p_i$, and the pair is eliminated. Conditional on $\mathcal{G}$, the condition above guarantees the elimination of the hidden blocking pair with probability at least $1 - \delta_e$. $\qquad\square$

If the matching is $\mathcal{S}$-Stable and no hidden blocking pairs exist, but $m_t \neq m*$, the system is trapped in a Deadlock Cycle (Definition B.7).

To formalize the mechanism by which these deadlock cycles are resolved, we establish an isomorphism between the algorithmic states and the lattice theoretic concept of rotation elimination. We prove that once the system achieves cognitive convergence $t > T_{cog}$, a statistical *synchronized* Restless Ping behavior deterministically triggers a monotonic climb on the stable matching lattice.

**Lemma B.10** (Monotonic Lattice Climbing). *When $t > T_{cog}$, let $\rho \subseteq m$ be a deadlock cycle of length at most $\ell \leq |\mathcal{N}_{T_{\text{stable}}}|$. Let $\mathcal{E}_{overlap}$ be the set of players in the deadlock cycle. If the dynamic windows of all players involved in $\rho$ exhibit a non-empty intersection $\mathcal{T} = \bigcap_{p_i \in \mathcal{E}_{overlap}} [t_i, t_i + L(t_i)] \neq \emptyset$, the system executes an irreversible state transition through the stable matching lattice, denoted by $m \to m'$.*

*Proof.* We establish the causal link between the Restless Ping mechanism and the exploration state. Consider a player $p_i$ in the deadlock attempting to ping the target arm $a_{i+1}$. By the definition, the arm $a_{i+1}$ is occupied by a superior incumbent $p_{i+1} \succ_{a_{i+1}} p_i$ who is currently in the exploitation state (and doesn't execute Restless Ping at the same time). Consequently, any proposal from $p_i$ is rejected.

A failed Ping immediately triggers the player switches to the exploration state and initializes a timer $timer \leftarrow L(t)$. Note that while an immediate success is possible if $p_{i+1}$ happens to be wandering (which would accelerate convergence), relying on the failure-induced cool-down provides a valid conservative lower bound for the transition analysis.

Assume the joint event $\mathcal{E}_{overlap}$ occurs, implying the existence of the window $\mathcal{T}$. During this interval, the following conditions hold for all $p_i$ which $(p_i, a_i) \in \rho$:

- **Unoccupied State:** Every incumbent $p_{i+1}$ is in the exploration state. This means they temporarily release their hold on arm $a_{i+1}$.

- **Cognitive Separation:** On the good event $\mathcal{G}$, since $t > T_{cog}$, the comparison between $a_{i+1}$ and $a_k$ is separated: $\text{LCB}_{i,a_{i+1}} > \text{UCB}_{i,a_i}$. According to the potential set definition and candidate set logic, inferior arms are pruned from the Potential Set $\mathcal{P}_i$. Consequently, $a_{i+1}$ becomes the unique element in $\mathcal{P}_i$.

- **Deterministic Sampling:** In the exploration mode, player $p_i$ polls $\mathcal{P}_i$. Since $a_{i+1}$ is available and statistically dominant, $p_i$ successfully samples $a_{i+1}$. This success triggers climb success routine:

$$M_i \leftarrow \text{EXPLOIT}; \quad a_{held} \leftarrow a_{i+1}.$$

This process occurs concurrently for all $i$, resulting in the matching update $m'(i) = a_{i+1}$. The transition $m \to m'$ is permanent due to preference consistency. In the new matching $m'$, every player $p_i$ holds a strictly preferred arm $(a_{i+1} \succ_{p_i} a_i)$. Since $a_{held}$ has improved to $a_{i+1}$, the previous arm $a_i$ is strictly excluded from $\mathcal{D}_i$, so that players will never transit backwards. $\square$

The overlapping dynamic window, the required target choice, and cognitive convergence ensure a deterministic rotation elimination. Now we analyze the expected time for deadlock resolution for the event $\mathcal{E}_{overlap}$.

**Lemma B.11.** *Consider a deadlock cycle of $\ell$ players from $\mathcal{N}_{T_{\text{stable}}}$, with local clocks $\{t_i\}_{i=1}^{\ell}$. Define $t_{\max} \triangleq \max_{i \in [\ell]} t_i$. If $t_{\max} > \left[ 4C_L^{K/2} \log\left(2C_L^{K/2}\right) \right]^2$, then the probability that all $\ell$ players synchronize Restless Pings to the corresponding arms in the cycle within their overlapping windows satisfies*

$$P_{\text{joint},\ell}(t_{\max}) \geq \left( \frac{(1-e^{-1})C_L}{2K} \right)^{\ell} t_{\max}^{-\ell/K} (\log t_{\max})^{2\ell/K}.$$

*Proof.* Let $P_{\text{active}}(u)$ denote the probability that a player with local clock $u$ initiates at least one ping to the corresponding arm in the cycle within their window $[u, u + L(u) - 1]$. Since $p$ is decreasing, $p(u+j) \geq p(u + L(u))$ for all $0 \leq j < L(u)$. Applying $1 - x \leq e^{-x}$,

$$P_{\text{active}}(u) = 1 - \prod_{j=0}^{L(u)-1} \left(1 - \frac{p(u+j)}{K}\right) \geq 1 - \left(1 - \frac{p(u+L(u))}{K}\right)^{L(u)} \geq 1 - \exp\left(-\frac{L(u)p(u+L(u))}{K}\right).$$

To convert this exponential lower bound into a linear expression in $L(u)p(u)$, we need three properties for every $u > \left[ 4C_L^{K/2} \log\left(2C_L^{K/2}\right) \right]^2$:

1. $L(u) \leq u$. $\left[4C_L^{K/2} \log\left(2C_L^{K/2}\right)\right]^2 > 2C_L \log C_L$ and $u > e$, so Lemma B.2 with $Q = C_L$ gives $C_L \log u \leq u$.

2. $L(u)p(u + L(u)) \leq 1$. $\sqrt{u} > 4C_L^{K/2} \log(2C_L^{K/2})$ and Lemma B.2 with $Q = 2C_L^{K/2}$ give $\sqrt{u} \geq C_L^{K/2} \log u$, i.e., $C_L^K (\log u)^2 \leq u$. With $p(u + L(u)) \leq p(u)$,

$$L(u)p(u + L(u)) \leq C_L u^{-1/K} (\log u)^{2/K} \leq 1.$$

3. $p(u + L(u)) \geq \frac{1}{2}p(u)$. $L(u) \leq u$ implies $u + L(u) \leq 2u$, hence

$$\frac{p(u + L(u))}{p(u)} = \left(\frac{u}{u + L(u)}\right)^{1/K} \left(\frac{\log u}{\log(u + L(u))}\right)^{(K-2)/K} \geq 2^{-\frac{K-1}{K}} > \frac{1}{2}.$$

Applying $1 - e^{-x} \geq (1 - e^{-1})x$ (valid for $0 \leq x \leq 1$) to the exponential bound:

$$P_{\text{active}}(u) \geq \frac{1 - e^{-1}}{K} L(u)p(u + L(u)) \geq \frac{1 - e^{-1}}{2K} L(u)p(u),$$

where the second inequality uses the third property $p(u + L(u)) \geq \frac{1}{2}p(u)$. On the other hand, by the union bound,

$$P_{\text{active}}(u) \leq \sum_{j=0}^{L(u)-1} \frac{p(u + j)}{K} \leq \frac{L(u)p(u)}{K}.$$

Hence for all $u > \left[4C_L^{K/2} \log\left(2C_L^{K/2}\right)\right]^2$,

$$P_{\text{active}}(u) = \Theta\left(\frac{L(u)p(u)}{K}\right).$$

Moreover, $L(u)p(u) = C_L u^{-1/K}(\log u)^{2/K}$ is decreasing on this range, since

$$\frac{d}{du} \log(L(u)p(u)) = \frac{1}{Ku}\left(-1 + \frac{2}{\log u}\right) < 0$$

for $u > \left[4C_L^{K/2} \log\left(2C_L^{K/2}\right)\right]^2 > e^2$. Substituting the definitions of $p(u)$ and $L(u)$,

$$P_{\text{active}}(u) = \Theta\left(\frac{1}{K} u^{-1/K}(\log u)^{2/K}\right).$$

**Joint probability.** A deadlock breaks when all $\ell$ players ping the corresponding arms in the deadlock cycle synchronously within their overlapping windows. Assuming independence of players' ping randomization,

$$P_{\text{joint},\ell}(t_{\max}) = \prod_{i=1}^{\ell} P_{\text{active}}(t_i).$$

Since $L(u)p(u)$ is decreasing on this range, we have $L(t_i)p(t_i) \geq L(t_{\max})p(t_{\max})$ for all $t_i \leq t_{\max}$. Applying the lower bound $P_{\text{active}}(u) \geq \frac{1-e^{-1}}{2K} L(u)p(u)$,

$$P_{\text{active}}(t_i) \geq \frac{1 - e^{-1}}{2K} L(t_{\max})p(t_{\max}).$$

Multiplying over all $\ell$ players and substituting $p(t) = t^{-1/K}(\log t)^{-(K-2)/K}$, $L(t) = C_L \log t$:

$$P_{\text{joint},\ell}(t_{\max}) \geq \left(\frac{1 - e^{-1}}{2K} C_L\right)^{\ell} t_{\max}^{-\ell/K} (\log t_{\max})^{2\ell/K}.$$

This completes the proof. □

**Theorem B.12** (Finite Expected Breaking Time). *Let* $T_{win} \triangleq \max\left\{ \left[ 4C_L^{K/2} \log\left(2C_L^{K/2}\right) \right]^2, \exp(2/A) \right\}$ *and* $A \triangleq$ $C_L^{K-1} \left( \frac{1-e^{-1}}{2K} \right)^K$. *Assume the system enters a deadlock cycle when every involved player has local clock larger than $T_{win}$, and let $t_{\max}$ be the largest local clock among the players in the cycle (as defined in Lemma B.11). The conditional expected time $\tau$ to break the deadlock satisfies:*

$$\mathbb{E}[\tau \mid t_{\max}] \leq \frac{2t_{\max}}{A \log t_{\max}}.$$

*Proof.* To derive a worst-case lower bound, let the cycle length be $\ell$ and let $t_{\max} \triangleq \max_{i \in [\ell]} t_i$ be the largest local clock among the players in the cycle at the entry time; this player has the lowest exploration probability and dominates the waiting time. Taking the worst-case length $\ell = |\mathcal{N}_{T_{\text{stable}}}| = K$ in Lemma B.11 yields the following bound.

From Lemma B.11, the joint synchronization probability satisfies

$$P_{\text{joint},K}(t_{\max}) \geq \left( \frac{(1-e^{-1})C_L}{2K} \right)^K t_{\max}^{-1}(\log t_{\max})^2.$$

We now bound the survival probability by partitioning the waiting interval into consecutive epochs. Let

$$S(\tau \mid t_{\max}) \triangleq \Pr(\text{deadlock not broken by time } t_{\max} + \tau \mid \text{entered deadlock at } t_{\max})$$

denote the conditional survival function of the deadlock-breaking time. Define $t_0 = t_{\max}$ and $t_{k+1} = t_k + L(t_k)$. In the $k$-th epoch $[t_k, t_{k+1})$, the deadlock is broken if all $K$ players synchronize their Restless Pings within overlapping windows. This occurs with probability at least $P_{\text{joint},K}(t_k)$. Hence, after $m$ epochs (covering $\tau = t_m - t_{\max}$ steps),

$$S(\tau \mid t_{\max}) \leq \prod_{k=0}^{m-1} \left( 1 - P_{\text{joint},K}(t_k) \right) \leq \exp\left( -\sum_{k=0}^{m-1} P_{\text{joint},K}(t_k) \right).$$

To handle the summation term in the exponent, we first rewrite it in a form compatible with an integral over time. Define the hazard density

$$h(t) \triangleq \frac{P_{\text{joint},K}(t)}{L(t)},$$

which measures the deadlock-breaking intensity per unit time. Then

$$\sum_{k=0}^{m-1} P_{\text{joint},K}(t_k) = \sum_{k=0}^{m-1} \frac{P_{\text{joint},K}(t_k)}{L(t_k)} L(t_k) = \sum_{k=0}^{m-1} h(t_k)L(t_k).$$

Using $L(t) = C_L \log t$ and the definition of $A$, the lower bound above gives

$$h(t) \geq \left( \frac{1-e^{-1}}{2K} \right)^K C_L^{K-1} \frac{\log t}{t} = A \frac{\log t}{t}.$$

Thus we do not need to impose monotonicity on the exact hazard $h(t)$ itself. It is enough to apply the Riemann-sum comparison to the explicit lower-bound profile $(\log t)/t$, which is decreasing for $t > e$. Since $t_{k+1} - t_k = L(t_k)$, the left-endpoint Riemann sum dominates the corresponding integral:

$$\sum_{k=0}^{m-1} P_{\text{joint},K}(t_k) \geq A \sum_{k=0}^{m-1} \frac{\log t_k}{t_k} L(t_k) \geq A \int_{t_0}^{t_m} \frac{\log s}{s} \, ds$$

$$= A \int_{t_{\max}}^{t_{\max}+\tau} \frac{\log s}{s} \, ds$$

$$= \frac{A}{2} \left[ (\log(t_{\max} + \tau))^2 - (\log t_{\max})^2 \right].$$

Thus, the survival function $S(\tau \mid t_{\max})$ satisfies:

$$S(\tau \mid t_{\max}) \le \exp\left(-\frac{A}{2}\left[(\log(t_{\max}+\tau))^2 - (\log t_{\max})^2\right]\right).$$

The conditional expected waiting time is:

$$\mathbb{E}[\tau \mid t_{\max}] = \int_0^\infty S(\tau \mid t_{\max})d\tau$$

Use the change of variables $y = (\log(t_{\max}+\tau))^2 - (\log t_{\max})^2$. Then $t_{\max} + \tau = \exp(\sqrt{y + (\log t_{\max})^2})$. Differentiating with respect to $y$ gives

$$d\tau = \frac{1}{2}(y + (\log t_{\max})^2)^{-\frac{1}{2}} \exp\left(\sqrt{y + (\log t_{\max})^2}\right) dy.$$

Since $y \ge 0$, we have $(y + (\log t_{\max})^2)^{-\frac{1}{2}} \le (\log t_{\max})^{-1}$. Moreover, by concavity of the square-root function,

$$\sqrt{(\log t_{\max})^2 + y} \le \log t_{\max} + \frac{y}{2\log t_{\max}},$$

and therefore $\exp(\sqrt{(\log t_{\max})^2 + y}) \le t_{\max}\exp(y/(2\log t_{\max}))$. Substituting these bounds into the integral yields

$$\mathbb{E}[\tau \mid t_{\max}] \le \frac{t_{\max}}{2\log t_{\max}} \int_0^\infty \exp\left(-y\left(\frac{A}{2} - \frac{1}{2\log t_{\max}}\right)\right) dy.$$

Since $t_{\max} \ge T_{win} \ge \exp(2/A)$, we have $\frac{1}{2\log t_{\max}} \le \frac{A}{4}$. Therefore $\frac{A}{2} - \frac{1}{2\log t_{\max}} \ge \frac{A}{4}$ and the integral converges to at most $\frac{4}{A}$. Thus:

$$\mathbb{E}[\tau \mid t_{\max}] \le \frac{t_{\max}}{2\log t_{\max}} \cdot \frac{4}{A} = \frac{2t_{\max}}{A\log t_{\max}}.$$

$\square$

Let $H$ denote the lattice distance, the number of rotation eliminations required to reach the POSM from the current matching. By (Gusfield & Irving, 1989), the lattice of stable matchings has height at most $K(K-1)/2$ for a market with $K$ agents on each side, hence $H \le K(K-1)/2 = O(K^2)$.

Finally, we establish that the system converges to the player-optimal stable matching in finite expected time. Let $T_{struct}$ denote the stopping time at which the system first reaches the player-optimal stable matching (POSM).

**Theorem B.13** (Structural Stabilization). *After cognitive stabilization, the system converges to the unique player-optimal stable matching (POSM) in expected time:*

$$\mathbb{E}[T_{struct}] \le \mathbb{E}\left[\left(1 + \frac{2}{A\log T_{cog}}\right)^H T_{cog}\right] + O(1),$$

*where $A$ is defined in Theorem B.12 and $H \le K(K-1)/2$ is the lattice height.*

*Proof.* Each deadlock resolution eliminates one rotation and moves the system strictly upward on the stable matching lattice. Since the lattice height $H$ is finite, the process must terminate at the POSM $m^*$ after at most $H$ steps: at $m^*$, no player-initiated blocking pair exists, so any ping attempt targets an arm that strictly prefers its current occupant and is rejected; the matching can no longer change.

Theorem B.12 requires every involved player's local clock to exceed $T_{win}$. After $T_{stable}$, the active player set is fixed and local clocks increase by one per round. Cognitive separation holds for all $t > T_{cog}$. Starting from $\max\{T_{cog}, T_{win}\}$, within at most $T_{win}$ additional steps every active player's local clock exceeds $T_{win}$. Hence there exists a time $T_0$ satisfying

$$\max\{T_{cog}, T_{win}\} \le T_0 \le \max\{T_{cog}, T_{win}\} + T_{win},$$

from which onward the condition of Theorem B.12 is satisfied for every subsequent deadlock cycle. Since $T_0 \leq T_{cog} + 2T_{win} = T_{cog} + O(1)$, the regret incurred during $[T_{cog}, T_0)$ is $O(1)$ and absorbed into the constant term of the final bound. Moreover $\log T_0 \geq \log \max\{T_{cog}, T_{win}\} \geq \log T_{cog}$.

We analyze the structural convergence time $T_{struct}$ as a sequence of $H$ discrete steps on the lattice. Let $T_k$ denote the timestamp after completing the $k$-th step, with $T_H = T_{struct}$. The temporal evolution is governed by the recurrence:

$$T_{k+1} = T_k + \tau_k,$$

where $\tau_k$ is the waiting time to execute the $(k+1)$-th rotation conditional on the current time $T_k$.

Let $\mathcal{F}_{T_k}$ denote the $\sigma$-algebra generated by the history of the process up to time $T_k$. By Theorem B.12, the expected waiting time satisfies

$$\mathbb{E}[\tau_k \mid \mathcal{F}_{T_k}] \leq \frac{2T_k}{A \log T_k},$$

since at time $T_k$ all local clocks exceed $T_{win}$ and the deadlock cycle is entered with largest local clock at most $T_k$. Taking the conditional expectation of the next timestamp:

$$\mathbb{E}[T_{k+1} \mid \mathcal{F}_{T_k}] = T_k + \mathbb{E}[\tau_k \mid \mathcal{F}_{T_k}] \leq T_k \left(1 + \frac{2}{A \log T_k}\right).$$

Since $T_k \geq T_0 \geq T_{cog}$, we have $\frac{1}{\log T_k} \leq \frac{1}{\log T_0} \leq \frac{1}{\log T_{cog}}$. Hence

$$\mathbb{E}[T_{k+1} \mid \mathcal{F}_{T_k}] \leq T_k \left(1 + \frac{2}{A \log T_{cog}}\right).$$

Now let $\mathcal{F}_{T_{cog}}$ denote the $\sigma$-algebra generated by the history up to $T_{cog}$. Both $T_{cog}$ and the lattice state at $T_{cog}$ (and therefore $H$) are $\mathcal{F}_{T_{cog}}$-measurable. Because $T_k \geq T_0 \geq \max\{T_{cog}, T_{win}\} \geq T_{cog}$ are stopping times, the stopping-time $\sigma$-algebras are nested: $\mathcal{F}_{T_{cog}} \subseteq \mathcal{F}_{T_k}$. Applying the tower property:

$$\begin{aligned}
\mathbb{E}[T_{k+1} \mid \mathcal{F}_{T_{cog}}] &= \mathbb{E}\big[\mathbb{E}[T_{k+1} \mid \mathcal{F}_{T_k}] \mid \mathcal{F}_{T_{cog}}\big] \\
&\leq \mathbb{E}\left[T_k \left(1 + \frac{2}{A \log T_{cog}}\right) \,\Big|\, \mathcal{F}_{T_{cog}}\right] \\
&= \left(1 + \frac{2}{A \log T_{cog}}\right) \mathbb{E}[T_k \mid \mathcal{F}_{T_{cog}}],
\end{aligned}$$

where the last equality holds because $(1 + \frac{2}{A \log T_{cog}})$ is $\mathcal{F}_{T_{cog}}$-measurable (it depends only on $T_{cog}$) and can be pulled out of the conditional expectation.

Iterating this recurrence from $k = 0$ to $H - 1$:

$$\mathbb{E}[T_H \mid \mathcal{F}_{T_{cog}}] \leq \left(1 + \frac{2}{A \log T_{cog}}\right)^H \mathbb{E}[T_0 \mid \mathcal{F}_{T_{cog}}].$$

We do not require $T_0$ itself to be $\mathcal{F}_{T_{cog}}$-measurable; the deterministic bound $T_0 \leq T_{cog} + 2T_{win}$ (from $\max\{T_{cog}, T_{win}\} \leq T_{cog} + T_{win}$) together with the $\mathcal{F}_{T_{cog}}$-measurability of $T_{cog}$ gives $\mathbb{E}[T_0 \mid \mathcal{F}_{T_{cog}}] \leq T_{cog} + 2T_{win}$. Hence

$$\mathbb{E}[T_H \mid \mathcal{F}_{T_{cog}}] \leq \left(1 + \frac{2}{A \log T_{cog}}\right)^H (T_{cog} + 2T_{win}).$$

Since $T_{win} = O(1)$, taking the total expectation yields

$$\mathbb{E}[T_{struct}] \leq \mathbb{E}\left[\left(1 + \frac{2}{A \log T_{cog}}\right)^H (T_{cog} + 2T_{win})\right] \leq \mathbb{E}\left[\left(1 + \frac{2}{A \log T_{cog}}\right)^H T_{cog}\right] + O(1).$$

This completes the proof. □

Theorem B.13 guarantees that the lattice-ascent procedure terminates in finite expected time independent of the horizon $T$: the bound involves only $T_{cog}$, $A$, and $H$, none of which depend on $T$. This ensures the system reaches the POSM almost surely, and it justifies treating the deadlock-breaking portion of $\mathbb{E}[R_{\text{struct}}]$ as $O(1)$ in the regret decomposition below.

### B.3. Combining the Auxiliary Results for Theorem 5.1

We now prove Theorem 5.1 by assembling the learning and structural guarantees established above. The proof decomposes regret into statistical failures, cognitive stabilization, structural recovery, and steady-state exploration.

The learning part is controlled by the two-sided Good Event in Definition B.1. Lemma B.3 and Lemma B.4 show that effective sample counts eventually exceed the separation threshold $L(t)$, and Theorem B.5 converts this into a finite expected cognitive stabilization time $T_{cog}$.

The structural part handles the remaining obstacles after cognitive stabilization. Lemma B.8 gives convergence to restricted stability; Lemma B.9 eliminates hidden blocking pairs; Lemma B.10, Lemma B.11, and Theorem B.12 control deadlock resolution through Restless Pings. Theorem B.13 aggregates these steps into finite expected convergence to the POSM.

It remains to charge regret to these phases: bad-event regret, pre-stabilization learning, post-stabilization structural recovery, and the steady-state cost of persistent Restless Pings. Combining the resulting bounds gives the desired cumulative-regret guarantee.

*Proof of Theorem 5.1.* Without loss of generality, we focus on the players still active in $\mathcal{N}_{\text{stable}}$, as other players' regrets are independent of $T$ by the definition of Eq. (1), and thus a constant compared to the right-hand side of Eq. (8).

At any time $t$, the instantaneous regret is given by $r_t = \mu_{i,m_t^*(i)} - X_i(t)$. Since rewards are bounded in $[0,1]$, we always have $r_t \leq 1$. We decompose the expected regret via the law of total expectation, splitting on the good event $\mathcal{G}$ (all confidence intervals valid). By the law of total expectation,

$$\mathbb{E}[R_i(T)] = \underbrace{\mathbb{E}[R_i(T) \mid \neg\mathcal{G}]\mathbb{P}(\neg\mathcal{G})}_{\text{Statistical Failure Cost}} + \underbrace{\mathbb{E}[R_i(T) \mid \mathcal{G}] \cdot \mathbb{P}(\mathcal{G})}_{\text{Algorithmic Cost}}$$

We first bound the contribution from confidence failures. By Definition B.1,

$$R_{\text{bad}} \triangleq \sum_{t=1}^{T} \mathbb{E}[r_t \mathbf{1}_{\mathcal{G}_t^c}] \leq \sum_{t=1}^{T} \mathbb{P}(\mathcal{G}_t^c) \leq \sum_{t=1}^{T} \frac{6\delta_g}{\pi^2 t^2} \leq \delta_g,$$

and hence $R_{\text{bad}}$ is uniformly bounded independent of $T$.

For the algorithmic term, using $\mathbb{P}(\mathcal{G}) \leq 1$,

$$\mathbb{E}[R_i(T) \mid \mathcal{G}] \cdot \mathbb{P}(\mathcal{G}) \leq \mathbb{E}[R_i(T) \cdot \mathbf{1}_{\mathcal{G}}].$$

On the good event $\mathcal{G}$, the algorithm's progression follows three natural phases: cognitive learning ($t \leq T_{cog}$) for statistical learning regret, structural adjustment until the market reaches the POSM, and steady-state exploration through Restless Pings, yielding the pointwise bound

$$R_i(T) \cdot \mathbf{1}_{\mathcal{G}} \leq R_{\text{cog}} + R_{\text{struct}} + R_{\text{steady}}.$$

Taking its expectation,

$$\mathbb{E}[R_i(T) \mid \mathcal{G}] \cdot \mathbb{P}(\mathcal{G}) \leq \mathbb{E}[R_{\text{cog}}] + \mathbb{E}[R_{\text{struct}}] + \mathbb{E}[R_{\text{steady}}].$$

Combining both terms,

$$\mathbb{E}[R_i(T)] \leq R_{\text{bad}} + \mathbb{E}[R_{\text{cog}}] + \mathbb{E}[R_{\text{struct}}] + \mathbb{E}[R_{\text{steady}}].$$

We now bound each term on the right-hand side.

**Part A: Cognitive Learning Regret ($R_{\text{cog}}$).** This term corresponds to the regret incurred during the initial learning phase ($t \leq T_{cog}$) where players may make sub-optimal decisions due to insufficient samples ($N_{i,k}(t) < L(t)$). By Theorem B.5,

we know the upper bound of expected cognitive convergence time. Meanwhile, as players suffer at most 1 regret at every step until convergence, the expected regret $R_{\text{cog}}$ can be bounded by:

$$\mathbb{E}[R_{\text{cog}}] \leq \mathbb{E}[T_{cog}] \leq T_{alg} + \frac{K^2 T_{alg}^{2-\gamma}}{(\gamma-1)(\gamma-2)},$$

where $T_{alg}$ and $\gamma$ are constant independent with $T$.

**Part B: Structural Adjustment Regret ($R_{\text{struct}}$).** This term of regret captures the cost of adapting to structural changes to climb the lattice from $m_{old}$ to $m_{new}$. We decompose this into historical costs and the post-stabilization recovery costs.

1. Historical Adaptation Cost ($t \leq T_{\text{stable}}$): We denote the regret accumulated during the volatile period before the system reaches a steady state ($T_{\text{stable}}$) as $R_{\text{hist}}(T_{\text{stable}})$. Since $T_{\text{stable}}$ depends solely on the finite sequence of structural shocks (e.g., player entries/exits) and the intrinsic market dynamics, this term is a constant independent of the total horizon $T$. Thus, $\mathbb{E}[R_{\text{hist}}] \leq T_{\text{stable}}$.

2. Post-Stabilization Recovery Cost ($t > T_{\text{stable}}$): After the final structural shock, the system must resolve remaining suboptimalities, primarily through Deadlock Breaking and Candidate Resets:

- Deadlock Breaking Cost ($R_{\text{deadlock}}$): After cognitive convergence ($t > T_{cog}$), deadlock cycles(Definition B.7) may still block the ascent to the POSM. By Theorem B.13, once all local clocks exceed $T_{win}$, which occurs by $T_0$ with $\max\{T_{cog}, T_{win}\} \leq T_0 \leq T_{cog} + 2T_{win}$, the lattice-climbing procedure reaches the POSM in finite expected time. The deadlock-breaking duration proper is $T_{struct} - T_0 \leq T_{struct} - T_{cog}$, and the $O(1)$ gap $T_0 - T_{cog} \leq 2T_{win}$ is absorbed into the constant-order structural cost. Theorem B.13 guarantees $\mathbb{E}[T_{struct}] < \infty$, and this bound is independent of the time horizon $T$. Since each player suffers at most unit regret per step and there are at most $K$ active players, the deadlock-breaking regret over all rotation eliminations satisfies

$$\mathbb{E}[R_{\text{deadlock}}] \leq K \cdot \mathbb{E}[T_{struct}] = O(1).$$

- Candidate Reset Cost ($R_{\text{reset}}$): When a player is forced to reset their candidate set, they incur a cost proportional to the time of continuous rejection $L(t)$ and the time next time they enter the exploitation state $L(t)$. Also, as the number of resets is at most $|\mathcal{N}_{T_{\text{stable}}}| \cdot K \cdot H$, where $H$ is the finite height of the stable matching lattice bounded. Thus

$$\mathbb{E}[R_{\text{reset}}] \leq |\mathcal{N}_{T_{\text{stable}}}| K \cdot H(2L(t) + K) \leq O(C_L \log T) = O\left(\frac{\log T}{\Delta_{\min}^2}\right).$$

Combining the above bounds, the total structural adjustment regret satisfies:

$$\mathbb{E}[R_{\text{struct}}] = \mathbb{E}[R_{\text{hist}} + R_{\text{deadlock}} + R_{\text{reset}}] \leq O(C_L \log T).$$

**Part C: Steady-State Exploration Regret ($R_{\text{steady}}$).** Once the system reaches the POSM, players must maintain *Restless Pings* to ensure liveness against potential future changes. As the Restless Ping rate $p(t)$ and a ping takes time $L(t) = C_L \log t$, we have:

$$\mathbb{E}[R_{\text{steady}}(T)] \leq \sum_{t=1}^{T} K \cdot p(t) \cdot C_L \log t.$$

Substituting $p(t)$ and keeping the dependence on the separation threshold explicit, we get:

$$\mathbb{E}[R_{\text{steady}}(T)] \leq KC_L \int_1^T x^{-1/K} (\log x)^{2/K} \, dx.$$

Using integration by parts:

$$\int_1^T x^{-1/K}(\log x)^{2/K} \, dx = \left[\frac{K}{K-1}x^{1-1/K}(\log x)^{2/K}\right]_1^T - \frac{2}{K-1}\int_1^T x^{-1/K}(\log x)^{2/K-1} \, dx$$

$$\leq \frac{K}{K-1}T^{1-1/K}(\log T)^{2/K}.$$

Therefore,

$$\mathbb{E}[R_{\text{steady}}(T)] \leq O\Big(KC_L T^{1-1/K}(\log T)^{2/K}\Big) = O\Big(\frac{KT^{1-1/K}(\log T)^{2/K}}{\Delta_{\min}^2}\Big).$$

Combining all the above upper bounds together, the regret of each player $i$ is upper bounded by:

$$\begin{aligned}
\mathbb{E}[R_i(T)] &\leq R_{\text{bad}} + \mathbb{E}[R_{\text{cog}}] + \mathbb{E}[R_{\text{struct}}] + \mathbb{E}[R_{\text{steady}}] \\
&\leq \underbrace{O(1)}_{\text{Bad Events}} + \underbrace{O(1)}_{\text{Cognitive}} + \underbrace{O(C_L \log T)}_{\text{Structural Resets}} + \underbrace{O(KC_L T^{1-1/K}(\log T)^{2/K})}_{\text{Steady-State}} \\
&\leq O\Big(\frac{KT^{1-1/K}(\log T)^{2/K}}{\Delta_{\min}^2}\Big).
\end{aligned}$$

Thus, the regret bound in Theorem 5.1 holds. $\qquad\square$

## C. The Restless Ping Rate $p(t)$

As mentioned in Section 5.1, the Restless Ping rate $p(t)$ in Equation (7) can be scaled by any constant factor $c > 0$. In fact, for sufficiently large $t$, $p(t)$ can take from a broad class of functions:

$$p(t) = ct^{-a}(\log t)^{-b_1}(\log \log t)^{-b_2} \cdots (\log^{(m)} t)^{-b_m}, \tag{12}$$

for properly chosen $m, a, b_1, b_2, \ldots, b_m$, and $c$.

Indeed, the analysis in previous sections has proved the regret bound relative to Equation 7 for Algorithm 3. We now explain why this specific formula is chosen, how it relates to the more general family as in Equation 12, and how a fixed multiplicative constant $c$ affects the recovery guarantee. The purpose here is not to repeat the derivation of $P_{\text{active}}(t)$, $P_{\text{joint},K}(t)$, or the epoch-level hazard $h(t)$, which has already been established before. Instead, we start from the consequence needed by the proof: for a deadlock cycle, the Restless Ping schedule should make the induced hazard large enough to ensure both almost-sure recovery and finite expected deadlock breaking time.

### C.1. Sufficient Conditions

Following Lemma B.11, $P_{\text{active}}(t) = \Theta(L(t)p(t)/K)$ and $P_{\text{joint},K}(t) = \Omega((L(t)p(t)/K)^K)$. Using the hazard density $h(t) \triangleq P_{\text{joint},K}(t)/L(t)$ as in Theorem B.12, we obtain

$$h(t) = \Omega\Big(\frac{L(t)^{K-1}p(t)^K}{K^K}\Big) = \Omega((\log t)^{K-1}p(t)^K).$$

For deadlock recovery, we are concerned with two levels of conditions.

**Almost-sure recovery.** The deadlock is broken in the almost-sure sense, i.e., $\Pr(\tau < \infty \mid T) = 1$. From the survival function upper bound

$$S(\tau \mid T) \leq \exp\Big(-\int_T^{T+\tau} h(s)\, ds\Big),$$

this is certified if $S(\tau \mid T) \to 0$ as $\tau \to \infty$, for which a sufficient condition is that the cumulative hazard diverges:

$$\int_T^\infty h(t)\, dt = \infty.$$

**Finite expected recovery.** A stronger requirement is a finite expected recovery time, namely

$$\mathbb{E}[\tau \mid T] < \infty.$$

Since $\mathbb{E}[\tau \mid T] = \int_0^\infty S(\tau \mid T)\, d\tau$, this is guaranteed when the survival function decays sufficiently fast. The integral analysis of Theorem B.12 shows that a sufficient condition is that the hazard density eventually strictly exceeds the critical scale $1/t$ with a strict constant margin: there exist constants $\varepsilon > 0$ and $T_0$ such that, for all $t \geq T_0$,

$$h(t) \geq \frac{1+\varepsilon}{t}.$$

Indeed, under this condition, the cumulative hazard satisfies, for all sufficiently large $x$,

$$\int_T^x h(s)\, ds \geq (1+\varepsilon)\log x - O(1).$$

Consequently, the survival function is bounded by

$$S(\tau \mid T) \leq O\left((T+\tau)^{-(1+\varepsilon)}\right),$$

which is integrable over $\tau \in [0, \infty)$.

The scale $1/t$ is critical because

$$\int_T^{T+\tau} \frac{1}{t}\, dt = \log \frac{T+\tau}{T}.$$

Thus, a hazard lower bound of the form $h(t) \geq c'/t$ with some constant $c'$ implies the power-law survival bound

$$S(\tau \mid T) \leq \left(\frac{T}{T+\tau}\right)^{c'}.$$

The tail integral of this power-law survival function bound is finite precisely when $c' > 1$. Hence the scale $1/t$ is constant-sensitive: a constant factor strictly above one certifies finite mean recovery through this argument, whereas a unit or smaller constant does not. By contrast, any hazard that grows asymptotically faster than $1/t$, for example $h(t) = \Omega((\log t)/t)$, lies strictly on the finite-mean side and yields an integrable survival tail.

To see the choices of $p(t)$, we now apply $h(t) = \Omega((\log t)^{K-1}p(t)^K)$ to a general power-log and iterated-logarithm family. For any fixed $c > 0$ (with the feasibility constraint that $p(t) < 1$ given the other parameters), consider Equation 12 for any fixed positive integer $m$. Substitution gives

$$h(t) = \Omega\left(c^K t^{-aK}(\log t)^{K-1-Kb_1}(\log\log t)^{-Kb_2}\cdots(\log^{(m)} t)^{-Kb_m}\right).$$

Since $c^K$ is a fixed positive number, it does not affect the polynomial exponent, the logarithmic exponent vector, nor the asymptotic size of $h(t)$. It only alters the leading constant of the hazard.

## C.2. Critical Condition on the Polynomial Exponent

The polynomial exponent is decisive at the first level. Forming the ratio of the hazard lower bound to the critical scale $1/t$:

$$\frac{h(t)}{1/t} = \Omega\left(c^K t^{1-aK}(\log t)^{K-1-Kb_1}(\log\log t)^{-Kb_2}\cdots(\log^{(m)} t)^{-Kb_m}\right).$$

The polynomial factor $t^{1-aK}$ determines the dominant magnitude. If $a < 1/K$, then $1 - aK > 0$ and $t^{1-aK} \to \infty$. The remaining logarithmic factors, regardless of their signs, diverge or decay at rates that are of strictly lower order than any polynomial and are ultimately absorbed by the polynomial divergence. Consequently, there exists $T_0$ such that for all $t \geq T_0$, $h(t) \geq (1+\varepsilon)/t$, and finite expected recovery holds. If $a > 1/K$, then $1 - aK < 0$, $t^{1-aK} \to 0$, the product tends to zero, and this lower bound eventually falls below $(1+\varepsilon)/t$, so this lower-bound argument does not certify finite expected recovery. Hence, the critical polynomial exponent is

$$a = \frac{1}{K}.$$

In this critical case, the polynomial factor cancels out ($t^0 = 1$), and the decision shifts to the logarithmic factors. In this regime,

$$h(t) \geq \frac{Ac^K}{t}(\log t)^{K-1-Kb_1}(\log\log t)^{-Kb_2}\cdots(\log^{(m)} t)^{-Kb_m}.$$

Let $M(t)$ denote the product of the above logarithmic factors. From the condition $h(t) \geq (1+\varepsilon)/t$, we require $Ac^K M(t)$ to eventually exceed $1+\varepsilon$. If $M(t) \to \infty$, any fixed $c > 0$ suffices; if $M(t) \to 0$, no fixed $c$ can make this sufficient

condition hold; if $M(t) \equiv 1$, convergence depends on whether $Ac^K$ exceeds 1. This is the origin of the sign condition and constant sensitivity. We now rigorously derive how the asymptotic behavior of $M(t)$ determines convergence via the integral structure of the cumulative hazard.

Define $\gamma_1 := K - 1 - Kb_1$, $\gamma_2 := -Kb_2$, ..., $\gamma_m := -Kb_m$, so that $M(t) = (\log t)^{\gamma_1} (\log \log t)^{\gamma_2} \cdots (\log^{(m)} t)^{\gamma_m}$. Under $a = 1/K$, the cumulative hazard, after the substitution $u = \log s$ ($ds/s = du$), transforms to

$$\int_T^x h(s) \, ds \geq Ac^K \int_{\log T}^{\log x} u^{\gamma_1} (\log u)^{\gamma_2} (\log \log u)^{\gamma_3} \cdots (\log^{(m-1)} u)^{\gamma_m} \, du.$$

Let $v := \log x$, $L_0 := \log T$, and denote by $\mathcal{I}(v; L_0)$ the integral on the right-hand side above. From the survival function upper bound $S(\tau \mid T) \leq \exp(-\int_T^{T+\tau} h(t) \, dt)$ and the change of variables $x = e^v$ ($dx = e^v dv$),

$$\mathbb{E}[\tau \mid T] \leq \int_{\log T}^{\infty} \exp\Big(v - Ac^K \cdot \mathcal{I}(v; L_0)\Big) \, dv. \tag{13}$$

The integrand in the exponent contains two terms: $\exp(v)$ from the Jacobian, and $\exp(-Ac^K \cdot \mathcal{I})$ from the cumulative hazard. The integral converges if and only if the growth of $\mathcal{I}(v; L_0)$ is sufficient to dominate the linear term $v$ in the exponent.

We now analyze the asymptotic behavior of $\mathcal{I}(v; L_0)$ case by case. Define the vector

$$\Gamma(b_1, \ldots, b_m) := (\gamma_1, \gamma_2, \ldots, \gamma_m) = (K - 1 - Kb_1, -Kb_2, \ldots, -Kb_m).$$

**Case 1: $\gamma_1 > 0$.** Although the higher-order logarithmic factors may contain negative powers, they are of strictly lower order than any polynomial power of $u$. Hence, for $\gamma_1/2$, there exists $V_0 \geq L_0$ such that, for all $u \geq V_0$,

$$(\log u)^{\gamma_2} (\log \log u)^{\gamma_3} \cdots (\log^{(m-1)} u)^{\gamma_m} \geq u^{-\gamma_1/2}.$$

Therefore, for all sufficiently large $v \geq V_0$,

$$\begin{aligned}
\mathcal{I}(v; L_0) &= \int_{L_0}^v u^{\gamma_1} (\log u)^{\gamma_2} (\log \log u)^{\gamma_3} \cdots (\log^{(m-1)} u)^{\gamma_m} \, du \\
&\geq \int_{V_0}^v u^{\gamma_1/2} \, du \\
&= \frac{v^{1+\gamma_1/2} - V_0^{1+\gamma_1/2}}{1 + \gamma_1/2}.
\end{aligned}$$

Since $V_0$ is fixed, for sufficiently large $v$ this implies

$$\mathcal{I}(v; L_0) \geq Cv^{1+\gamma_1/2}$$

for some constant $C > 0$. Substituting this into Eq.(13) gives

$$\exp\big(v - Ac^K \mathcal{I}(v; L_0)\big) \leq \exp\Big(v - C'v^{1+\gamma_1/2}\Big)$$

for some constant $C' > 0$. Since $1 + \gamma_1/2 > 1$, the negative superlinear term dominates the linear Jacobian term $v$, and the integral converges. In this case, any fixed $c > 0$ guarantees $\mathbb{E}[\tau \mid T] < \infty$. The condition $\gamma_1 > 0$ is equivalent to $b_1 < (K-1)/K$.

**Case 2: $\gamma_1 < 0$.** We distinguish two subcases.

*Subcase 2a: $-1 < \gamma_1 < 0$* (equivalently, $(K-1)/K < b_1 < 1$). The higher-order logarithmic factors may contain positive powers, so we cannot simply upper bound the integrand by $u^{\gamma_1}$. However, these factors are still of lower order than any positive polynomial power of $u$. Hence, for $\eta = -\gamma_1/2 > 0$, for all sufficiently large $u$,

$$(\log u)^{\gamma_2} (\log \log u)^{\gamma_3} \cdots (\log^{(m-1)} u)^{\gamma_m} \leq u^{-\gamma_1/2}.$$

Therefore, for sufficiently large $v$,

$$\mathcal{I}(v; L_0) \leq C + \int^v u^{\gamma_1/2}\, du$$
$$= O\left(v^{1+\gamma_1/2}\right) = o(v),$$

since $\gamma_1/2 < 0$. Substituting this into Eq.(13) shows that the exponent

$$v - Ac^K \mathcal{I}(v; L_0)$$

still grows linearly in $v$ up to a lower-order correction. Thus the survival upper bound obtained from this hazard lower bound is not integrable, and this argument does not certify finite expected recovery.

*Subcase 2b:* $\gamma_1 \leq -1$ (equivalently, $b_1 \geq 1$). Again, the higher-order logarithmic factors are dominated by any positive polynomial power of $u$. In particular, for all sufficiently large $u$,

$$(\log u)^{\gamma_2} (\log \log u)^{\gamma_3} \cdots (\log^{(m-1)} u)^{\gamma_m} \leq u^{1/2}.$$

Hence the integrand in $\mathcal{I}(v; L_0)$ is at most $u^{\gamma_1+1/2} \leq u^{-1/2}$ for large $u$, and therefore

$$\mathcal{I}(v; L_0) = O(v^{1/2}) = o(v).$$

Consequently, the term $Ac^K \mathcal{I}(v; L_0)$ cannot dominate the linear Jacobian term $v$, and the resulting survival upper bound is not integrable. Thus this hazard lower bound does not certify finite expected recovery.

**Case 3: $\gamma_1 = 0$ and $\gamma_2 \neq 0$.**  In this case,

$$\mathcal{I}(v; L_0) = \int_{L_0}^{v} (\log u)^{\gamma_2} (\log \log u)^{\gamma_3} \cdots (\log^{(m-1)} u)^{\gamma_m}\, du.$$

If $\gamma_2 > 0$, the remaining factors are of lower order than any power of $\log u$. Hence, for all sufficiently large $u$,

$$(\log \log u)^{\gamma_3} \cdots (\log^{(m-1)} u)^{\gamma_m} \geq (\log u)^{-\gamma_2/2}.$$

Thus

$$\mathcal{I}(v; L_0) \geq Cv(\log v)^{\gamma_2/2}$$

for some constant $C > 0$ and all sufficiently large $v$. Substituting into Eq.(13) gives

$$\exp\left(v - Ac^K \mathcal{I}(v; L_0)\right) \leq \exp\left(v - C'v(\log v)^{\gamma_2/2}\right),$$

which is integrable because $(\log v)^{\gamma_2/2} \to \infty$. Therefore, $\gamma_2 > 0$ certifies finite expected recovery for every fixed $c > 0$.

If $\gamma_2 < 0$, then for all sufficiently large $u$,

$$(\log \log u)^{\gamma_3} \cdots (\log^{(m-1)} u)^{\gamma_m} \leq (\log u)^{-\gamma_2/2}.$$

Hence

$$\mathcal{I}(v; L_0) = O\left(v(\log v)^{\gamma_2/2}\right) = o(v),$$

and the cumulative-hazard term does not dominate the linear Jacobian term. Thus this hazard lower bound does not certify finite expected recovery.

**Case 4: All $\gamma_i = 0$.** In this case, $M(t) \equiv 1$ and

$$\mathcal{I}(v; L_0) = v - L_0.$$

Substituting into Eq.(13) yields

$$\mathbb{E}[\tau \mid T] \leq \int_{\log T}^{\infty} \exp\big(v - Ac^K(v - \log T)\big)\, dv$$

$$= T^{Ac^K} \int_{\log T}^{\infty} \exp\big(v(1 - Ac^K)\big)\, dv.$$

The right-hand side is finite exactly when

$$Ac^K > 1.$$

In that case,

$$\mathbb{E}[\tau \mid T] \leq \frac{T}{Ac^K - 1}.$$

If $Ac^K \leq 1$, this particular survival upper bound is not integrable, and the present lower-bound argument does not certify finite expected recovery. This is the constant-sensitive boundary: at the exact logarithmic boundary, the exponents alone no longer decide the finite-mean condition; the leading constant $c$ also matters.

In summary, at the polynomial boundary $a = 1/K$, the finite-mean behavior certified by this hazard lower-bound argument is determined by the first nonzero component of $\Gamma$. If the first nonzero component is positive, then $\mathcal{I}(v; L_0)$ grows superlinearly in $v$, and the cumulative-hazard term dominates the Jacobian term $\exp(v)$ in Eq.(13). Hence finite expected recovery is certified for every fixed $c > 0$. If the first nonzero component is negative, then $\mathcal{I}(v; L_0) = o(v)$, and this lower-bound argument does not certify finite expected recovery.

Translating this criterion into conditions on the logarithmic exponents $(b_1, \ldots, b_m)$, the first coordinate of $\Gamma$ satisfies

$$K - 1 - Kb_1 > 0 \quad \Longleftrightarrow \quad b_1 < \frac{K - 1}{K}.$$

If $b_1 = (K - 1)/K$, then the first coordinate vanishes and the second coordinate determines the sign:

$$-Kb_2 > 0 \quad \Longleftrightarrow \quad b_2 < 0.$$

If also $b_2 = 0$, the same reasoning propagates to $b_3$, and so forth. Thus, for every fixed $c > 0$, the finite-mean side certified by this argument can be compactly expressed as

$$(b_1, b_2, \ldots, b_m) \prec_{\text{lex}} \left(\frac{K - 1}{K}, 0, 0, \ldots\right),$$

where $\prec_{\text{lex}}$ denotes the lexicographic order: the left-hand vector is smaller if, at the first coordinate where it differs from $((K - 1)/K, 0, 0, \ldots)$, its coordinate is smaller. The exact boundary

$$(b_1, b_2, \ldots, b_m) = \left(\frac{K - 1}{K}, 0, 0, \ldots\right)$$

is constant-sensitive. In this case, the hazard lower bound reduces to $h(t) \geq Ac^K \frac{1}{t}$, and this argument gives the sufficient condition $Ac^K > 1$ for finite expected recovery. Equivalently, the boundary case is certified when

$$c > A^{-1/K} = \frac{2K}{1 - e^{-1}}\, C_L^{-(K-1)/K}.$$

## C.3. The Parameter Choice in This Paper

Algorithm 3 chooses the simpler form

$$p(t) = \frac{1}{t^{1/K}(\log t)^{(K-2)/K}},$$

which corresponds to

$$a = \frac{1}{K}, \qquad b_1 = \frac{K-2}{K}, \qquad b_2 = b_3 = \cdots = 0.$$

Equivalently, the more general constant-scaled family

$$p_c(t) = \frac{c}{t^{1/K} (\log t)^{(K-2)/K}}$$

shares the same exponents. In particular, substituting this schedule into the hazard

$$h(t) = \Omega\left( c^K (\log t)^{K-1} \left( \frac{1}{t^{1/K} (\log t)^{(K-2)/K}} \right)^K \right) = \Omega\left( c^K \frac{\log t}{t} \right).$$

Hence, any fixed $c > 0$ remains on the same finite-mean side: the $\log t$ factor diverges and eventually overcomes any fixed constant penalty. The algorithm in the paper, Algorithm 3, corresponds to the special case $c = 1$. Since $\frac{K-2}{K} < \frac{K-1}{K}$, this choice lies strictly on the finite-mean side of the boundary. This is the reason for using the exponent $(K-2)/K$: it avoids the constant-sensitive boundary while maintaining a sublinear steady-state probing cost,

$$\sum_{t=1}^{T} L(t)p(t) = O\left( T^{1-1/K} (\log T)^{2/K} \right).$$

Under the constant-scaled schedule $p_c(t) = c\, p(t)$, the same calculation acquires only an extra multiplicative factor $c$:

$$\sum_{t=1}^{T} L(t)p_c(t) = O\left( cT^{1-1/K} (\log T)^{2/K} \right).$$

Since $c$ is a fixed constant, the asymptotic exponent of the regret remains unchanged.

## D. Way-SE-2S WorkFlow and Illustrative Example

### D.1. Way-SE-2S Flow Diagram

Figure 3 summarizes the execution logic of Algorithm 3. The diagram is intended as a readability aid and does not introduce additional algorithmic steps beyond those specified in Algorithm 3.

### D.2. Illustrative Example: Set Dynamics and Bi-directional Tracking under Departure

To intuitively demonstrate how the Way-SE-2S algorithm resolves deadlocks and tracks the Player-Optimal Stable Matching (POSM) in dynamic environments, we construct a local network instance with structural shocks (churn). Consider a market with an arm set $\mathcal{K} = \{a_1, a_2, a_3\}$ and an initial player set $\mathcal{M}_{init} = \{P_1, P_2\}$. A new player $P_3$ dynamically enters, and later, an incumbent player $P_1$ departs. The latent true preferences (unknown to the agents and learned via UCB) are defined as follows:

- **Players' Preferences:** $P_1, P_2 : a_1 \succ a_2 \succ a_3$; and $P_3 : a_2 \succ a_1 \succ a_3$.

- **Arms' Preferences:** $a_1 : P_1 \succ P_2 \succ P_3$; $a_2 : P_3 \succ P_2 \succ P_1$; with $a_3$ being indifferent.

We divide the system evolution into three distinct stages, focusing on the state transitions and the dynamics of the core sets $(S_i, O_i, \mathcal{P}_i, D_i)$ for player $P_2$, who acts as the primary observer of the market shocks.

**Stage I: Cold Start and Downward Relaxation.** Upon initialization, both $P_1$ and $P_2$ are in the EXPLORE state with full candidate sets $S = \{a_1, a_2, a_3\}$ and empty observation sets (cold palace) $O = \emptyset$. Driven by the optimistic UCB initialization, both players attempt to lock $a_1$. However, learning from proposals, arm $a_1$ identifies $P_1 \succ P_2$, thus consistently accepting $P_1$ while rejecting $P_2$. Once $P_2$'s consecutive rejections exceed the tolerance threshold $W_{lock}$, an *Eviction* is triggered: $a_1$ is removed from $S_2$ and placed into $O_2$. $P_2$ subsequently relaxes its expectation and successfully locks $a_2$ from the remaining potential set. The system reaches an initial POSM: $\mu = \{(P_1, a_1), (P_2, a_2)\}$. Although $P_2$ is exploiting $a_2$, $a_1$ remains

## Way-SE-2S Algorithm Flow Diagram

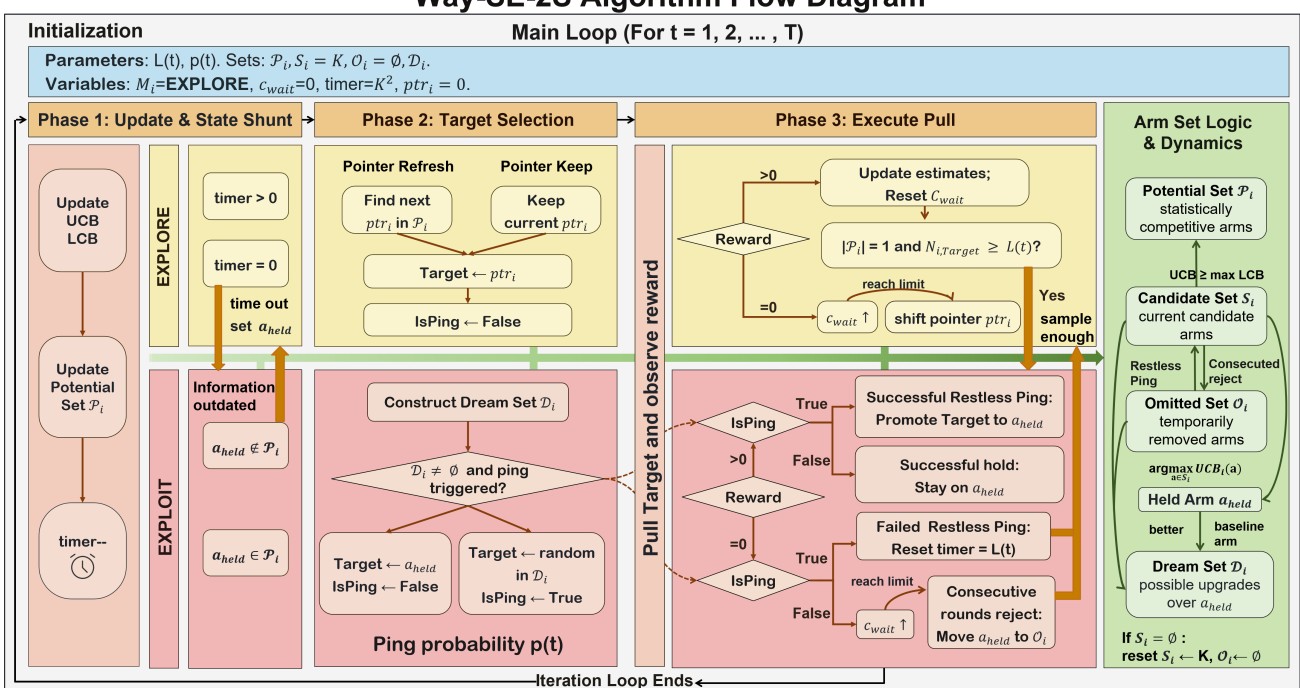

*Figure 3.* Flow diagram of Way-SE-2S. The algorithm proceeds through three phases in each round: update and state shunt, target selection, and execution with conflict handling. The right panel summarizes the arm-set dynamics among the Potential Set $\mathcal{P}_i$, Candidate Set $\mathcal{S}_i$, Omitted Set $\mathcal{O}_i$, held arm $a_{\text{held}}$, and Dream Set $\mathcal{D}_i$.

in $P_2$'s dream set ($D_2$) due to its high UCB. $P_2$ will occasionally probe (ping) $a_1$ with probability $p(F)$, which fails and increases the fatigue $F$, maintaining macroscopic stability.

**Stage II: Dynamic Entry and Cascading Eviction.** A new player $P_3$ enters the market with high initial UCBs, targeting $a_2$. Arm $a_2$ evaluates $P_3 \succ P_2$ and begins accepting $P_3$, consequently rejecting the incumbent $P_2$. This structural shock forces $P_2$ to face continuous rejections again, triggering a secondary eviction. Arm $a_2$ is moved to $O_2$, leaving $S_2 = \{a_3\}$. $P_2$ is forced to match with the baseline arm $a_3$. The system naturally gravitates downward to absorb the new competition, settling into a sub-optimal stable matching $\mu' = \{(P_1, a_1), (P_3, a_2), (P_2, a_3)\}$.

**Stage III: Incumbent Exit and Upward Excitation.** Assume $P_1$ unexpectedly departs, leaving $a_1$ vacant. $P_2$, currently matched with $a_3$, still retains $a_1 \in O_2$ with a high confidence bound, meaning $a_1 \in D_2$. When $P_2$ initiates an active *Restless Ping* towards $a_1$, the request is immediately accepted by the vacant arm. This success triggers the algorithm's *Resurrection and Preemption* logic: $a_1$ is pardoned from $O_2$ and restored to $S_2$, and $P_2$'s state directly transitions to EXPLOIT with $k_{held} = a_1$. The system successfully breaks the sub-optimal deadlock, climbing upward to converge at the new global POSM: $\mu'' = \{(P_2, a_1), (P_3, a_2)\}$.

The table highlights how the algorithm's memory mechanism (the $O$ set) prevents permanent truncation of true preferences. Unlike standard Gale-Shapley dynamics where a rejection implies permanent elimination, the WAY-SE-2S algorithm preserves highly valued but temporarily unavailable arms in $O_2$. This design is the key enabler for the upward excitation (resurrection) observed in Stage III, allowing the system to bidirectionally track the Player-Optimal Stable Matching (POSM) without centralized coordination.

# E. Experiments and Empirical Validation

In this section, we empirically test the theoretical guarantees of Way-SE and Way-SE-2S in dynamic matching markets. We demonstrate that both algorithms satisfy the predicted regret bounds while maintaining liveness and stability. The experiments are conducted in a decentralized environment where agents can only communicate through collision and rejection signals.

*Table 1.* Evolution of Player $P_2$'s Internal Sets and State under Dynamic Churn

| Stage & Event | Action / Feedback | Candidate Set ($S_2$) | Observation Set ($O_2$) | Dream Set ($D_2$) | Matched Arm ($k_{held}$) |
|---|---|---|---|---|---|
| **Initialization** | System starts; $P_1, P_2$ enter. | $\{a_1, a_2, a_3\}$ | $\emptyset$ | $\emptyset$ | None |
| **Stage I: Collision** | $P_1$ & $P_2$ pull $a_1$. $P_2$ rejected. | $\{a_1, a_2, a_3\}$ | $\emptyset$ | $\emptyset$ | None |
| **Stage I: Eviction** | $C_{wait} \geq W_{lock}$ for $a_1$. $P_2$ descends. | $\{a_2, a_3\}$ | $\{a_1\}$ | $\{a_1\}$ | $a_2$ |
| **Stage II: Entry** | $P_3$ enters, targets $a_2$. $P_2$ rejected. | $\{a_2, a_3\}$ | $\{a_1\}$ | $\{a_1\}$ | $a_2$ |
| **Stage II: Eviction** | $C_{wait} \geq W_{lock}$ for $a_2$. $P_2$ descends. | $\{a_3\}$ | $\{a_1, a_2\}$ | $\{a_1, a_2\}$ | $a_3$ |
| **Stage III: Exit** | $P_1$ leaves. $a_1$ becomes vacant. | $\{a_3\}$ | $\{a_1, a_2\}$ | $\{a_1, a_2\}$ | $a_3$ |
| **Stage III: Ping** | $P_2$ pings $a_1 \in D_2$. Ping accepted! | $\{a_3\}$ | $\{a_1, a_2\}$ | $\{a_1, a_2\}$ | None (*Transit*) |
| **Stage III: Resurrection** | $a_1$ revived from $O_2$. $P_2$ matches $a_1$. | $\{a_1, a_3\}$ | $\{a_2\}$ | $\{a_2\}$ | $a_1$ |

## E.1. One-Sided Learning: Logarithmic Regret

We evaluate the Way-SE algorithm in a one-sided learning environment, where 15 arms' preferences are known in advance, while players must learn their preferences. The system initializes with 6 players, with subsequent entries adding competition up to 11 concurrent players.

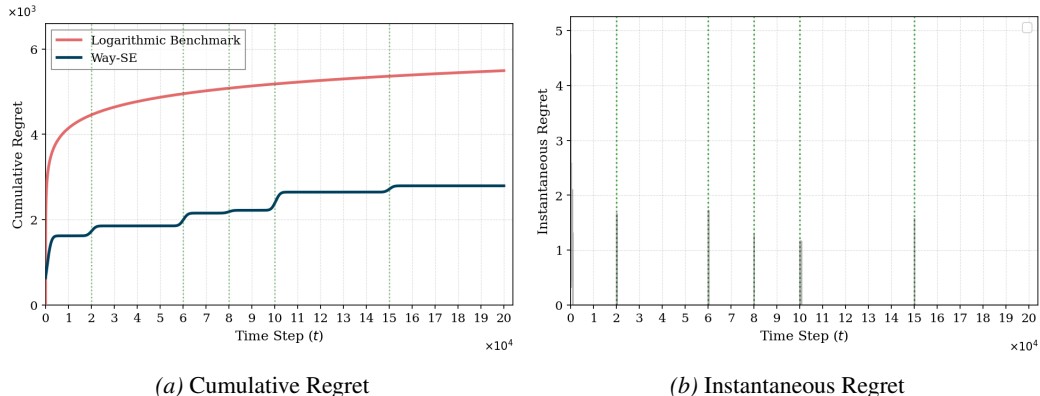

*(a)* Cumulative Regret                    *(b)* Instantaneous Regret

*Figure 4.* Way-SE in one-sided learning ($K = 15$). The left panel shows the cumulative regret tracking the logarithmic benchmark. The right panel illustrates the instantaneous regret, where sharp spikes correspond to player entry events, followed by rapid decay to zero.

As illustrated in Figure 4, the performance of the algorithm satisfies the theoretical upper bound predictions. The cumulative regret curve in the left panel maintains a trajectory exhibiting a logarithmic growth pattern consistent with the $O(\log T)$ bound. The right panel provides a microscopic view of the instantaneous regret at each time step. Each vertical dashed line represents a structural shock caused by a new player entering the market. These events trigger an immediate spike in instantaneous regret as the existing stability is disrupted and players are displaced. However, the system exhibits rapid recovery behavior; the regret spikes are transient and narrow, rapidly decaying back to zero. This return to a zero-regret state indicates that the market has re-converged to the new POSM, validating the effectiveness of the algorithm in handling dynamic player arrivals.

## E.2. Two-Sided Learning: Sublinear Regret

We evaluate the Way-SE-2S algorithm (Algorithm 3) in a two-sided learning environment where neither players nor arms know their true preferences in advance. The market initializes with $K = 15$ arms and 3 players, subjected to continuous structural shocks (player arrivals and departures) across three volatile scenarios. We compare our algorithm against two static decentralized baselines: RR-ETC (Zhang & Fang, 2024) and Epoch-based CA-ETC (Pagare & Ghosh, 2023).

As shown in Figure 5, Way-SE-2S consistently achieves sublinear cumulative regret and bounded instantaneous regret across all scenarios, significantly outperforming the baselines. While the static baselines may exhibit faster initial convergence before any population change, they fundamentally fail to adapt to dynamic environments. The asynchronous entry of new players disrupts their rigid phase synchronization, and the departure of players shifts the Player-Optimal Stable Matching (POSM). Consequently, even if a baseline algorithm coincidentally reaches a stable state, subsequent structural changes

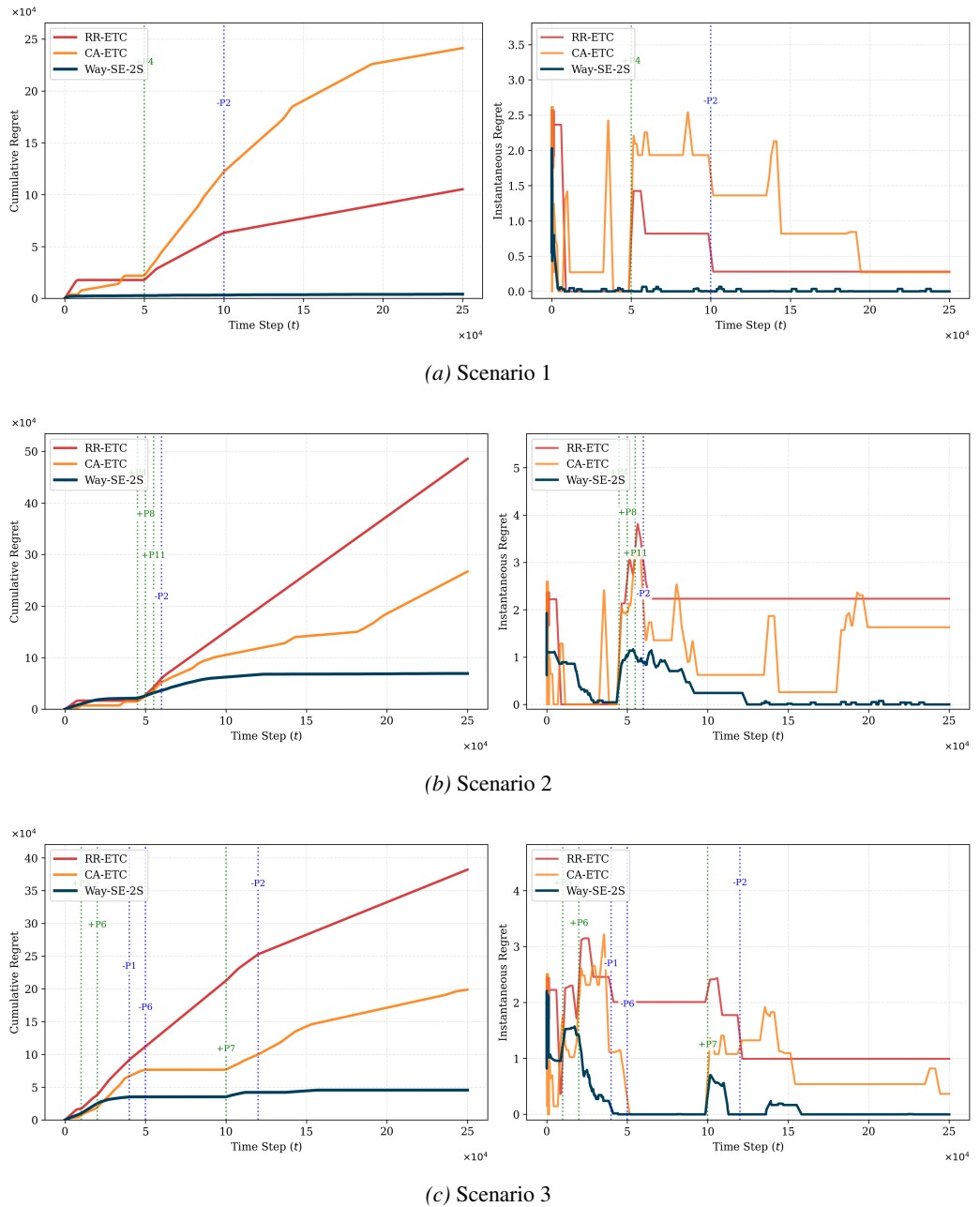

*(a)* Scenario 1

*(b)* Scenario 2

*(c)* Scenario 3

*Figure 5.* Way-SE-2S in two-sided learning ($K = 15$). The cumulative (left) and instantaneous (right) regret across three scenarios.

inevitably trap it in suboptimal matchings, causing its instantaneous regret to remain persistently positive and its cumulative regret to diverge linearly.

Conversely, the detailed instantaneous regret dynamics (the right panels of Figure 5) confirm the resilience of Way-SE-2S. Although structural shocks trigger immediate spikes in instantaneous regret when current POSM changes, the Restless Ping mechanism guarantees that players become aware of the situation and re-explore the new matching. Thus, these spikes are narrow and decay rapidly to zero in a finite time, confirming that the system swiftly resolves local conflicts and safely re-converges to the newly established POSM.

### E.3. Full Preference Landscape for $K = 15$ Experiments

To provide transparency regarding the simulation environment, we list the complete preference rankings for all players and arms used in the experiments with $K = 15$, for both Way-SE and Way-SE-2S. Table 2 details the player-side preferences derived from the underlying utility matrix, where at most 15 players are used. Table 3 details the arm-side preferences. These tables illustrate the conflicts in the matching instance, where efficient convergence requires resolving conflicts over popular choices.

*Table 2.* **Player Preference Queues.** Preference ranking for all 15 players over arms.

| P | Full Preference Queue ($K = 15$) |
|---|---|
| $P_0$ | $A_{12} \succ A_{10} \succ A_{11} \succ A_1 \succ A_7 \succ A_2 \succ A_9 \succ A_3 \succ A_8 \succ A_0 \succ A_{13} \succ A_{14} \succ A_4 \succ A_5 \succ A_6$ |
| $P_1$ | $A_{13} \succ A_{12} \succ A_1 \succ A_7 \succ A_3 \succ A_4 \succ A_{14} \succ A_8 \succ A_2 \succ A_5 \succ A_9 \succ A_0 \succ A_{11} \succ A_{10} \succ A_6$ |
| $P_2$ | $A_1 \succ A_{10} \succ A_0 \succ A_3 \succ A_2 \succ A_{13} \succ A_8 \succ A_{11} \succ A_7 \succ A_{12} \succ A_9 \succ A_5 \succ A_{14} \succ A_4 \succ A_6$ |
| $P_3$ | $A_{11} \succ A_4 \succ A_{10} \succ A_2 \succ A_{14} \succ A_8 \succ A_{13} \succ A_9 \succ A_3 \succ A_0 \succ A_7 \succ A_6 \succ A_{12} \succ A_1 \succ A_5$ |
| $P_4$ | $A_7 \succ A_9 \succ A_5 \succ A_{14} \succ A_{11} \succ A_8 \succ A_6 \succ A_{13} \succ A_0 \succ A_{12} \succ A_3 \succ A_1 \succ A_{10} \succ A_4 \succ A_2$ |
| $P_5$ | $A_{11} \succ A_{14} \succ A_{13} \succ A_9 \succ A_8 \succ A_3 \succ A_4 \succ A_{12} \succ A_6 \succ A_7 \succ A_0 \succ A_2 \succ A_1 \succ A_{10} \succ A_5$ |
| $P_6$ | $A_{10} \succ A_5 \succ A_{14} \succ A_1 \succ A_2 \succ A_9 \succ A_3 \succ A_{12} \succ A_0 \succ A_6 \succ A_{13} \succ A_7 \succ A_4 \succ A_8 \succ A_{11}$ |
| $P_7$ | $A_4 \succ A_{11} \succ A_5 \succ A_2 \succ A_6 \succ A_8 \succ A_{13} \succ A_{12} \succ A_7 \succ A_3 \succ A_{14} \succ A_{10} \succ A_0 \succ A_9 \succ A_1$ |
| $P_8$ | $A_{14} \succ A_3 \succ A_{10} \succ A_8 \succ A_5 \succ A_2 \succ A_{11} \succ A_9 \succ A_{12} \succ A_1 \succ A_7 \succ A_6 \succ A_4 \succ A_0 \succ A_{13}$ |
| $P_9$ | $A_7 \succ A_{14} \succ A_5 \succ A_{10} \succ A_8 \succ A_3 \succ A_9 \succ A_2 \succ A_{13} \succ A_{12} \succ A_1 \succ A_6 \succ A_{11} \succ A_0 \succ A_4$ |
| $P_{10}$ | $A_4 \succ A_{14} \succ A_6 \succ A_{11} \succ A_7 \succ A_9 \succ A_0 \succ A_3 \succ A_2 \succ A_8 \succ A_{10} \succ A_1 \succ A_5 \succ A_{12} \succ A_{13}$ |
| $P_{11}$ | $A_5 \succ A_{14} \succ A_8 \succ A_0 \succ A_{10} \succ A_6 \succ A_{13} \succ A_4 \succ A_9 \succ A_{12} \succ A_3 \succ A_7 \succ A_1 \succ A_{11} \succ A_2$ |
| $P_{12}$ | $A_{14} \succ A_{13} \succ A_8 \succ A_3 \succ A_1 \succ A_{11} \succ A_9 \succ A_{10} \succ A_4 \succ A_{12} \succ A_0 \succ A_5 \succ A_7 \succ A_6 \succ A_2$ |
| $P_{13}$ | $A_{14} \succ A_{13} \succ A_9 \succ A_{10} \succ A_4 \succ A_7 \succ A_3 \succ A_8 \succ A_1 \succ A_2 \succ A_6 \succ A_0 \succ A_{11} \succ A_{12} \succ A_5$ |
| $P_{14}$ | $A_9 \succ A_{13} \succ A_8 \succ A_{10} \succ A_5 \succ A_{14} \succ A_7 \succ A_{11} \succ A_0 \succ A_{12} \succ A_2 \succ A_4 \succ A_1 \succ A_6 \succ A_3$ |

*Table 3.* **Arm Preference Queues.** Preference ranking for all 15 arms over players.

| A | Full Preference Queue ($K = 15$) |
|---|---|
| $A_0$ | $P_1 \succ P_7 \succ P_{14} \succ P_5 \succ P_4 \succ P_0 \succ P_2 \succ P_8 \succ P_9 \succ P_{10} \succ P_{11} \succ P_3 \succ P_6 \succ P_{12} \succ P_{13}$ |
| $A_1$ | $P_{11} \succ P_6 \succ P_0 \succ P_{10} \succ P_{14} \succ P_4 \succ P_2 \succ P_5 \succ P_7 \succ P_9 \succ P_8 \succ P_{12} \succ P_1 \succ P_{13} \succ P_3$ |
| $A_2$ | $P_{12} \succ P_{10} \succ P_0 \succ P_{14} \succ P_{13} \succ P_3 \succ P_6 \succ P_1 \succ P_4 \succ P_9 \succ P_{11} \succ P_5 \succ P_2 \succ P_8 \succ P_7$ |
| $A_3$ | $P_0 \succ P_2 \succ P_{10} \succ P_3 \succ P_{14} \succ P_{13} \succ P_5 \succ P_{11} \succ P_2 \succ P_9 \succ P_4 \succ P_8 \succ P_7 \succ P_6 \succ P_1$ |
| $A_4$ | $P_3 \succ P_9 \succ P_1 \succ P_0 \succ P_{10} \succ P_{14} \succ P_8 \succ P_5 \succ P_{11} \succ P_2 \succ P_6 \succ P_{12} \succ P_{13} \succ P_4 \succ P_7$ |
| $A_5$ | $P_{14} \succ P_{11} \succ P_8 \succ P_9 \succ P_{12} \succ P_6 \succ P_{13} \succ P_0 \succ P_1 \succ P_7 \succ P_{10} \succ P_2 \succ P_3 \succ P_4 \succ P_5$ |
| $A_6$ | $P_4 \succ P_9 \succ P_3 \succ P_0 \succ P_7 \succ P_8 \succ P_{14} \succ P_2 \succ P_1 \succ P_{13} \succ P_{10} \succ P_{11} \succ P_5 \succ P_6 \succ P_{12}$ |
| $A_7$ | $P_{13} \succ P_0 \succ P_1 \succ P_4 \succ P_{11} \succ P_7 \succ P_9 \succ P_{10} \succ P_{12} \succ P_5 \succ P_8 \succ P_6 \succ P_{14} \succ P_2 \succ P_3$ |
| $A_8$ | $P_0 \succ P_{13} \succ P_2 \succ P_8 \succ P_9 \succ P_7 \succ P_{11} \succ P_1 \succ P_4 \succ P_5 \succ P_{10} \succ P_3 \succ P_6 \succ P_{14} \succ P_{12}$ |
| $A_9$ | $P_4 \succ P_7 \succ P_3 \succ P_0 \succ P_{10} \succ P_1 \succ P_9 \succ P_{12} \succ P_5 \succ P_{14} \succ P_{13} \succ P_8 \succ P_6 \succ P_{11} \succ P_2$ |
| $A_{10}$ | $P_0 \succ P_{12} \succ P_3 \succ P_{13} \succ P_4 \succ P_1 \succ P_2 \succ P_{11} \succ P_8 \succ P_5 \succ P_{10} \succ P_9 \succ P_7 \succ P_6 \succ P_{14}$ |
| $A_{11}$ | $P_{11} \succ P_9 \succ P_1 \succ P_5 \succ P_0 \succ P_{12} \succ P_{14} \succ P_3 \succ P_2 \succ P_8 \succ P_7 \succ P_{13} \succ P_4 \succ P_{10} \succ P_6$ |
| $A_{12}$ | $P_1 \succ P_9 \succ P_{11} \succ P_{10} \succ P_{14} \succ P_7 \succ P_5 \succ P_8 \succ P_4 \succ P_3 \succ P_{13} \succ P_{12} \succ P_0 \succ P_2 \succ P_6$ |
| $A_{13}$ | $P_9 \succ P_0 \succ P_6 \succ P_5 \succ P_3 \succ P_7 \succ P_2 \succ P_{10} \succ P_8 \succ P_{13} \succ P_4 \succ P_{12} \succ P_{11} \succ P_1 \succ P_{14}$ |
| $A_{14}$ | $P_{11} \succ P_0 \succ P_1 \succ P_4 \succ P_2 \succ P_9 \succ P_7 \succ P_3 \succ P_{14} \succ P_8 \succ P_{13} \succ P_{12} \succ P_6 \succ P_5 \succ P_{10}$ |

### E.4. Two-Sided Learning: Robustness in Large-Scale Environments

To evaluate the adaptability and robustness of our algorithm, we move beyond isolated shock scenarios and construct a complex, continuously evolving stochastic environment. Our objective is to demonstrate parameter sensitivity and stress-test the algorithm's resilience under relentless structural disruptions compounded by large state spaces and statistical noise.

**Stochastic Environment Modeling.** We model the dynamic market participation as a continuous stochastic process. Specifically, structural shock events (players' arrival and departure) sample following a Poisson process with an arrival rate of $\lambda_{\text{event}} = 0.001$ per time step before $T_{\text{stable}}$, where the value of $T_{\text{stable}}$ varies across different scenarios. When a shock event is triggered, its type is determined by a uniform distribution ($p = 1/2$ for arrival, $p = 1/2$ for departure). Furthermore,

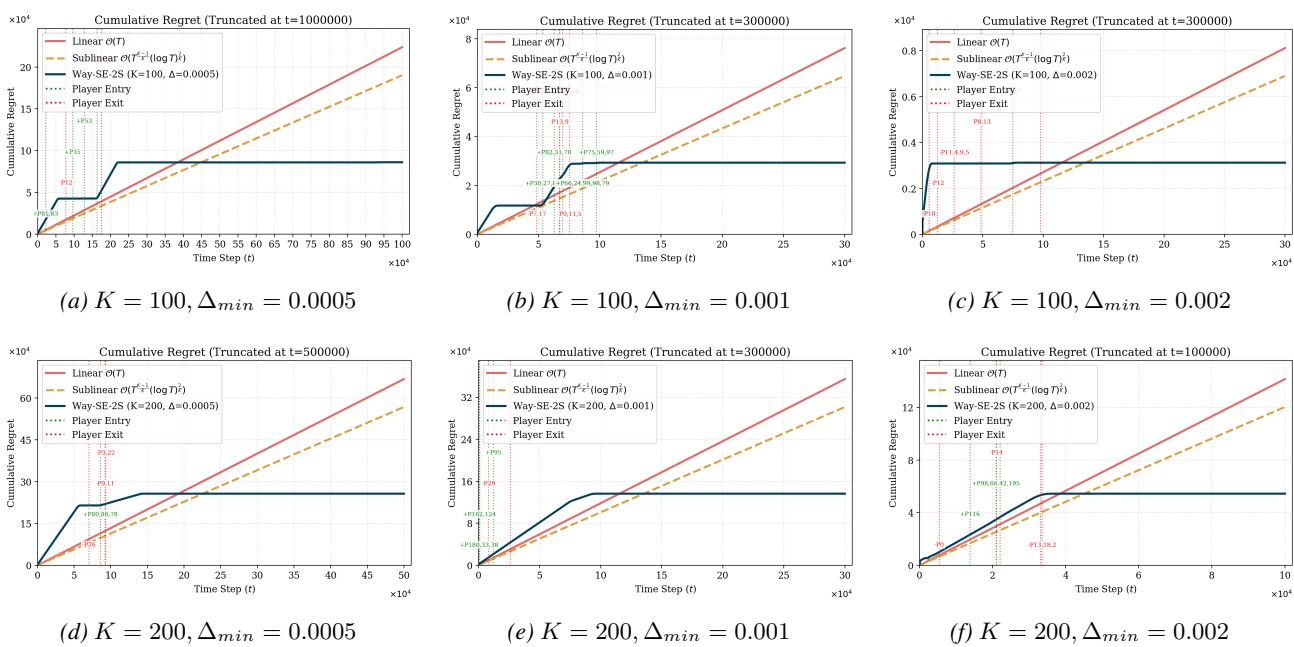

*Figure 6.* **Performance under Stochastic Dynamics:** Cumulative regret across large market sizes ($\{K, N_{\text{init}}\} \in \{\{100, 20\}, \{200, 30\}\}$) and varying $\Delta_{min}$'s. Way-SE-2S successfully absorbs the relentless stochastic disruptions and maintains a sublinear regret trajectory.

to simulate realistic burst traffic, the volume of the shock—i.e., the exact number of agents entering or leaving during the event—is sampled from a shifted Poisson distribution, strictly defined as $N_{\text{shift}} \sim \text{Poisson}(\lambda_{\text{num}} = 1.5) + 1$. Within this volatile and shifting population dynamics, we scale the quantity of players and arms to $\{K, N_{\text{init}}\} \in \{\{100, 20\}, \{200, 30\}\}$ and control the learning hardness via the minimum utility gap $\Delta_{\text{min}} \in \{0.0005, 0.001, 0.002\}$.

**Adaptability to Continuous Stochastic Shocks.** The combination of large state spaces ($K = 100$ or $200$) and continuous stochastic population shifts can trigger persistent deadlock cycles for static algorithms. The constant arrivals and departures of agents modeled by Poisson processes prevent static algorithm from reaching a steady phase. However, as demonstrated across all panels in Figure 6, Way-SE-2S can still converge to POSM. The Restless Ping mechanism swiftly breaks local deadlocks formed by random arrivals and prevents a permanent deviation from POSM.

**Conclusion.** These simulations confirm that the Way-SE-2S algorithm is resilient to stochastic market volatility. It can dynamically track the evolving POSM and maintain sublinear regret growth.

## F. An $L_{i,k}(t)$-Adaptive Variant of Way-SE-2S

In section 5, our Way-SE-2S algorithm uses the uniform sample separation threshold

$$L(t) = \left\lceil \frac{48 \log t}{\Delta_{\min}^2} \right\rceil + 1, \tag{14}$$

where $\Delta_{\min}$ is the minimum preference gap in the market. This appendix introduces the direct local-threshold variant obtained by replacing this uniform threshold with a player-arm specific threshold. For each pair $(p_i, a_k)$, define

$$L_{i,k}(t) = \left\lceil \frac{48 \log t}{\min\{\Delta_i, \Delta_k^{(a)}\}^2} \right\rceil + 1, \tag{15}$$

where $\Delta_i$ is the minimum player-side utility gap for player $p_i$, and $\Delta_k^{(a)}$ is the minimum arm-side utility gap for arm $a_k$.

We describe a local-threshold version of Way-SE-2S, in which the uniform threshold $L(t)$ in the main algorithm is replaced $L_{i,k}(t)$. The resulting procedure keeps the same state update, target selection, Restless Ping, and conflict-handling structure as Algorithm 3, while using the local threshold only in the corresponding waiting, timer reset, and sample threshold conditions.

## F.1. Modified Algorithm

---

**Algorithm 7** Way-SE-2S-$L_{i,k}$ (from the view of player $p_i$)

---

1: **Input:** Arm set $\mathcal{K} = \{a_k\}_{k=0}^{K-1}$, player-side gap $\Delta_i$, arm-side gaps $\{\Delta_k^{(a)}\}_{k=0}^{K-1}$, and parameter $\delta_g$.
2: **Global Variables:** Time step $t \leftarrow 1$, waiting counter $c_{wait} \leftarrow 0$, $\mathcal{S}_i \leftarrow \mathcal{K}$, $\mathcal{O}_i \leftarrow \emptyset$, $M_i \leftarrow$ EXPLORE, $timer \leftarrow K^2$, $a_{\text{held}} \leftarrow \perp$, $ptr_i \leftarrow 0$.
3: **Global Variables:** $\hat{\mu}_{i,k} \leftarrow 0$, $N_{i,k} \leftarrow 0$, $\text{UCB}_{i,k} \leftarrow 1$, $\text{LCB}_{i,k} \leftarrow 0$, $\forall k \in \{0, \ldots, K-1\}$.
4: **while** $t \leq T$ **do**
5:    *// Phase 1: State Update*
6:    $(\mathcal{P}_i, \{L_{i,k}(t)\}_{k=0}^{K-1}) \leftarrow$ STATEUPDATELOCAL$(\mathcal{K}, \Delta_i, \{\Delta_k^{(a)}\}_{k=0}^{K-1}, \delta_g)$;                    //Alg. 8
7:    $timer \leftarrow timer - 1$;
8:    **if** $M_i =$ EXPLORE $\wedge$ $timer = 0$ **then**
9:       $M_i \leftarrow$ EXPLOIT; $c_{wait} \leftarrow 0$;
10:       $a_{\text{held}} \leftarrow \arg\max_{a_k \in \mathcal{S}_i} \text{UCB}_{i,k}$;
11:    **else if** $M_i =$ EXPLOIT $\wedge$ $a_{\text{held}} \notin \mathcal{P}_i$ **then**
12:       $timer \leftarrow L_{i,a_{\text{held}}}(t)$;
13:       $M_i \leftarrow$ EXPLORE; $c_{wait} \leftarrow 0$;
14:       $a_{\text{held}} \leftarrow \perp$;
15:    **end if**
16:    *// Phase 2: Target Arm Selection*
17:    $(Target, IsPing) \leftarrow$ SELECTLOCAL$(\mathcal{K}, \mathcal{P}_i)$;                    //Alg. 9
18:    *// Phase 3: Execution & Conflict Handling*
19:    **Pull** $Target$; Observe $X_i(t)$;
20:    **if** $X_i(t) > 0$ **then**
21:       Update $\hat{\mu}_{i,Target}, N_{i,Target}$; $c_{wait} \leftarrow 0$;
22:       **if** $IsPing$ **then**
23:          $a_{\text{held}} \leftarrow Target$;                    //Ping Success
24:       **end if**
25:       **if** $(M_i =$ EXPLORE$) \wedge (N_{i,Target} \geq L_{i,Target}(t)) \wedge (|\mathcal{P}_i| = 1)$ **then**
26:          $a_{\text{held}} \leftarrow Target$; $M_i \leftarrow$ EXPLOIT;                    // Stabilize
27:       **end if**
28:    **else**
29:       CONFLICTLOCAL$(Target, IsPing, \mathcal{K})$.                    //Alg. 10
30:    **end if**
31:    $t \leftarrow t + 1$;
32: **end while**

---

The local threshold replaces the uniform threshold in exactly the places where the main algorithm waits for new samples or resets the timer after a rejected proposal. During exploration, a successful pull of $Target$ is sufficient for stabilization only after $N_{i,Target}(t) \geq L_{i,Target}(t)$ and $|\mathcal{P}_i| = 1$. If a Restless Ping fails, the player releases its held arm and returns to exploration with $timer \leftarrow L_{i,Target}(t)$. If a held arm rejects the player for $K$ consecutive rounds, the held arm is moved from $\mathcal{S}_i$ to $\mathcal{O}_i$, and the next timer reset uses $L_{i,a_{\text{old}}}(t)$, where $a_{\text{old}}$ is the arm that was just released. Finally, when an exploited held arm drops out of the potential set, the player also returns to exploration using the threshold for the held arm before releasing it.

## F.2. Analysis

The uniform threshold in section 5 is driven by the global worst-case quantity $\Delta_{\min}$. When $\Delta_{\min}$ is only a conservative lower bound, $L(t)$ can be larger than necessary for many player-arm interactions. Intuitively, the role of the threshold is unchanged after this replacement. In the main algorithm, $L(t)$ makes a player wait long enough for the confidence intervals to separate reliable candidates from inferior arms before stabilizing or after returning to exploration. The local threshold $L_{i,k}(t)$ serves the same purpose for the specific interaction $(p_i, a_k)$: it still enforces enough observations for the corresponding player-side and arm-side gaps, but avoids calibrating every interaction to the worst global lower bound $\Delta_{\min}$.

---

**Algorithm 8** STATEUPDATELOCAL

---

1: **Input:** Arm set $\mathcal{K}$, player-side gap $\Delta_i$, arm-side gaps $\{\Delta_k^{(a)}\}_{k=0}^{K-1}$, and parameter $\delta_g$.
2: **Global Input:** Candidate set $\mathcal{S}_i$, current time $t$.
3: $\delta_t \leftarrow \frac{3\delta_g}{\pi^2 K^2 t^2}$;
4: **for** $a_k \in \mathcal{K}$ **do**
5: $\quad L_{i,k}(t) \leftarrow \max\left\{ K, \left\lceil \frac{48\log t}{\min\{\Delta_i, \Delta_k^{(a)}\}^2} \right\rceil + 1 \right\}$;
6: **end for**
7: **for** $a_k \in \mathcal{K}$ **do**
8: $\quad$ **if** $N_{i,k} > 0$ **then**
9: $\quad\quad$ $\text{Rad}_{i,k} \leftarrow \frac{1}{2}\sqrt{\frac{1+N_{i,k}}{N_{i,k}^2}\left(1 + 2\log\left(\frac{K\sqrt{1+N_{i,k}}}{\delta_t}\right)\right)}$;
10: $\quad\quad$ $\text{UCB}_{i,k} \leftarrow \min(1, \hat{\mu}_{i,k} + \text{Rad}_{i,k})$;
11: $\quad\quad$ $\text{LCB}_{i,k} \leftarrow \max(0, \hat{\mu}_{i,k} - \text{Rad}_{i,k})$;
12: $\quad$ **else**
13: $\quad\quad$ $\text{UCB}_{i,k} \leftarrow 1; \text{LCB}_{i,k} \leftarrow 0$;
14: $\quad$ **end if**
15: **end for**
16: $\text{LCB}_{\max} \leftarrow \max_{a_j \in \mathcal{S}_i} \text{LCB}_{i,j}$;
17: $\mathcal{P}_i \leftarrow \{a_k \in \mathcal{S}_i \mid \text{UCB}_{i,k} \geq \text{LCB}_{\max}\}$;
18: **Return** $\mathcal{P}_i, \{L_{i,k}(t)\}_{k=0}^{K-1}$.

---

**Algorithm 9** SELECTLOCAL

---

1: **Input:** Arm set $\mathcal{K}$, Potential Set $\mathcal{P}_i$.
2: **Global Input:** $\mathcal{S}_i, \mathcal{O}_i, a_{\text{held}}, t, ptr_i, c_{wait}, M_i, \text{UCB}_{i,k}$.
3: **if** $M_i = \text{EXPLORE}$ **then**
4: $\quad$ **if** $(c_{wait} = 0) \lor (a_{ptr_i} \notin \mathcal{P}_i)$ **then**
5: $\quad\quad$ $d^* \leftarrow \min\{1 \leq d \leq K \mid a_{(ptr_i+d) \pmod{K}} \in \mathcal{P}_i\}$;
6: $\quad\quad$ $ptr_i \leftarrow (ptr_i + d^*) \pmod{K}; c_{wait} \leftarrow 0$;
7: $\quad$ **end if**
8: $\quad$ **Return** $(a_{ptr_i}, \text{False})$.
9: **else**
10: $\quad$ $\mathcal{D}_i \leftarrow \{a_k \in (\mathcal{S}_i \cup \mathcal{O}_i) \setminus \{a_{\text{held}}\} \mid \text{UCB}_{i,k} > \text{UCB}_{i,a_{\text{held}}}\}$;
11: $\quad$ $p(t) \leftarrow \frac{1}{t^{1/K}(\log t)^{(K-2)/K}}$;
12: $\quad$ **if** $\mathcal{D}_i \neq \emptyset \land \text{Rand}(\mathcal{U}(0,1)) < p(t)$ **then**
13: $\quad\quad$ $Target \leftarrow RandomSample(\mathcal{D}_i)$;
14: $\quad\quad$ $\mathcal{S}_i \leftarrow \mathcal{S}_i \cup \{Target\}; \mathcal{O}_i \leftarrow \mathcal{O}_i \setminus \{Target\}$;
15: $\quad\quad$ **Return** $(Target, \text{True})$.
16: $\quad$ **end if**
17: $\quad$ **Return** $(a_{\text{held}}, \text{False})$.
18: **end if**

---

---

**Algorithm 10** CONFLICTLOCAL

---

1: **Input:** $Target$, $IsPing$, arm set $\mathcal{K}$.
2: **Global Input:** $\mathcal{S}_i, \mathcal{O}_i, a_{\text{held}}, c_{wait}, M_i, timer, \{L_{i,k}(t)\}_{k=0}^{K-1}$.
3: $c_{wait} \leftarrow c_{wait} + 1$;
4: **if** $IsPing$ **then**
5:     $a_{\text{held}} \leftarrow \perp$;
6:     $c_{wait} \leftarrow 0$;
7:     $M_i \leftarrow \text{EXPLORE}$;
8:     $timer \leftarrow L_{i,Target}(t)$;                         // Ping Failed: target-specific timer reset
9: **else if** $M_i = \text{EXPLOIT} \wedge c_{wait} \geq K$ **then**
10:     $a_{\text{old}} \leftarrow a_{\text{held}}$;
11:     $\mathcal{S}_i \leftarrow \mathcal{S}_i \setminus \{a_{\text{old}}\}$; $\mathcal{O}_i \leftarrow \mathcal{O}_i \cup \{a_{\text{old}}\}$;
12:     $a_{\text{held}} \leftarrow \perp$;
13:     $M_i \leftarrow \text{EXPLORE}$; $timer \leftarrow L_{i,a_{\text{old}}}(t)$;
14:     **if** $\mathcal{S}_i = \emptyset$ **then**
15:         $\mathcal{S}_i \leftarrow \mathcal{K}$; $\mathcal{O}_i \leftarrow \emptyset$;                                // Candidate Reset
16:     **end if**
17: **else if** $M_i = \text{EXPLORE} \wedge c_{wait} \geq K$ **then**
18:     $c_{wait} \leftarrow 0$;
19: **end if**

---

This appendix does not change the worst-case theorem in the main text: replacing $L(t)$ by $L_{i,k}(t)$ gives a local-threshold variant whose analysis follows the same confidence-separation argument, with the global threshold recovered by the inequality $L_{i,k}(t) \leq L(t)$ whenever $\Delta_{\min} \leq \min\{\Delta_i, \Delta_k^{(a)}\}$.

