# OpenReview forum: "Decentralized Bandits without Global Clock for Dynamic Matching Market"
_ICML.cc/2026/Conference — ICML 2026 regular_

### Official Review · Reviewer_cQBk · 2026-03-11

**Soundness:** 3
**Presentation:** 3
**Significance:** 3
**Originality:** 4
**Overall Recommendation:** 4
**Confidence:** 3

**Summary:**

Paper is the first to offer a theoretical guarantee for stable matching in a decentralized and uncoordinated bandit market, making the method more applicable in real-world settings as it does not require assumptions in previous works such as a fixed participant set or a synchronized starting clock. Player arrivals and departures are arbitrary in this setting, which has been an open problem as it is much more complex than previous settings. Notably, they obtain sublinear theoretical guarantees on the regret in this setting.

**Compliance With Llm Reviewing Policy:**

Affirmed.

**Final Justification:**

Authors have addressed most of my concerns; I am happy to maintain my positive score.

**Key Questions For Authors:**

1.	One of this paper’s main contributions is that it gets rid of assumptions that were used in previous works. I would like to see more discussion of specific applications in which it becomes necessary to relax these structural assumptions.
2.	 Why were the scenarios in 6.3 chosen? Perhaps you could simulate many random scenarios for each of a) single entry-exit b) delayed exit c) high volatility, to show that the method performs well consistently in volatile scenarios beyond these specific anecdotes.
3.	I would be interesting in more emphasis of practical application. They are briefly mentioned in the introduction, but specific examples of when considering arbitrary player arrivals and departures would be beneficial would show the significance of the paper.

**Limitations:**

A future directions section is provided, which implicitly highlights current limitation of this work, and an impact statement is provided highlighting that the paper is very theoretical with no obvious social consequences.

**Strengths And Weaknesses:**

The authors provide well-supported theoretical regret guarantees in one-sided and two-sided learning settings.

Soundness:
Authors provide both rigorous theoretical justification and empirical validation. Algorithms are clearly stated and expanded upon in the appendix. Assumptions are also supported with literature reference. It would be interesting to see performance on at least one real data example to justify importance in practical application, although maybe it is difficult to obtain such data for use in this context.

Presentation:
Notation is very clear, paper does well at highlighting the contributions and uniqueness of this method. Theoretical results are expanded upon in the appendix, but proof sketches within the main manuscript are appreciated. Figures clearly show superior performance with respect to accumulated regret.

Significance:
Cumulative regret scales in this paper sub linearly, outperforming the linear baseline. This is a significant result, but I would be interested to see mentions of specific domains where matching markets are inherently dynamic.

---

> ### Author Rebuttal · Authors · 2026-03-31
>
> We sincerely thank the reviewer for recognizing the complexity of our settings, the importance of providing rigorous theoretical justification and empirical validation, and the clarity of our presentation.
>
> On providing a real data example, we agree with the reviewer and are currently studying real-world datasets to see how to analyze them in our model. Due to the limited amount of rebuttal time we haven't finished this part yet, but definitely plan to do so.
>
> For the 3 key questions:
>
> Q1: On specific applications
>
> This is indeed important and we have expanded Sections 1 and 7 to discuss applications where relaxing the assumptions is critical.
> (i) Decentralized job matching: In a job market, workers may enter the market at different times, apply to firms asynchronously, accept offers, and/or withdraw without sending any public signal or having any coordination with others. Firms also have their own timelines in their job posts and do not follow a common global clock to coordinate with other firms.
> (ii) Decentralized dating platforms: On decentralized dating platforms, users join and leave asynchronously, their interactions with each other are not publicly revealed, a player leaving the platform is typically without sending a public signal to everybody, and rejections or successful matches are typically not broadcast to others either.
> In both cases, the market evolves through private, uncoordinated local interactions rather than synchronized clocks or public signals. This is the type of environment our model tries to reflect.
>
> Q2: Random-scenario simulation
>
> This is a great suggestion. For each of the three categories, we have added experiments with randomized entry/exit scenarios in the new version of our paper.
> In particular, we consider the events as Poisson processes with different parameters, to simulate rare events and highly volatile events.
> Moreover, when an event happens, we also use Poisson distributions for the number of arrivals and/or departures.
> In addition, we repeat each configuration several times to examine the empirical consistency of our algorithm under such randomness.
>
> Q3: Practical applications
>
> This question is related to Q1 about the specific applications where removing the existing assumptions would be crucial. Indeed, the two applications,
> decentralized job matching and decentralized dating platforms are good examples where players arrive and depart arbitrarily without coordination. We have added this discussion in Sections 1 and 7 to highlight the significance of the dynamic matching setting, and we have also updated our Impact Statement to better reflect the practical significance of the dynamic setting beyond its current theoretical emphasis.
>
> Meanwhile, to further bridge theory and practice, we are investigating the dataset used in the 2021 JMLR paper suggested by another reviewer, as well as other possible real-world sources, and we plan to incorporate a real-data analysis in the final version of our paper.

---

> > ### Author Rebuttal · Reviewer_cQBk · 2026-04-02
> >
> > Thank you for the response. I will maintain my positive score.

---

> > > ### Author Response · Authors · 2026-04-08
> > >
> > > Thank you!

---

### Official Review · Reviewer_QpYU · 2026-03-15

**Soundness:** 3
**Presentation:** 3
**Significance:** 3
**Originality:** 3
**Overall Recommendation:** 3
**Confidence:** 4

**Summary:**

This paper studies decentralized bandit learning in two-sided matching markets where players arrive and depart at arbitrary times without access to a global clock or any shared signal. The authors propose two algorithms: Way-SE for one-sided learning with arrivals only, achieving $O(K^2 \log{T} / \Delta_{min}^2)$ regret, and Way-SE-2S for the fully decentralized two-sided setting with arbitrary arrivals and departures, achieving $O(T^{1-1/K} \log{T})$ regret. The key technical ideas are an index-free cyclic exploration mechanism for implicit coordination without synchronization, and a Restless Ping mechanism for detecting and adapting to structural changes in the stable matching caused by player departures.

**Compliance With Llm Reviewing Policy:**

Affirmed.

**Key Questions For Authors:**

1. Can the $\Delta_{\min}^2$ be removed with more advanced algorithms? Do we have any lower bound result.

2. How to determine the optimal $K$ consecutive rounds for all users, cross-validation?

3. On the exploration rate $p(t) = t^{-1/K}$: This rate is chosen to ensure a particular tail integral converges.

4. On the $T_{stable}$ assumption: Your regret bound decomposes into terms that are constant $(R_{cog}, R_{struct})$ and a term that grows with $T (R_{steady})$. This decomposition only works because T_stable is finite. What happens if players arrive and depart at a rate of, say, one event per $\Theta(T^{\alpha})$ time steps for some alpha in (0,1)? Can you characterize regret as a function of both $T$ and the frequency of structural changes?

5. More related works:
Two-sided Competing Matching Recommendation Markets With Quota and Complementary Preferences Constraints, ICML 2024
Learning Strategies in Decentralized Matching Markets under Uncertain Preferences, 2021 JMLR

If the author can provide reasonable response, I will consider to accept this paper.

**Limitations:**

1. The paper provides no lower bound analysis, so it is unknown whether this rate is fundamental to the problem or an artifact of the Restless Ping approach.

2. The assumption that the active player set eventually stabilizes (T_stable exists and N_t is fixed for all t >= T_stable) is unrealistic and critically important to the analysis, yet underemphasized.

**Strengths And Weaknesses:**

1. The removal of the global clock assumption and the dynamic arrivals and departures is a meaningful practice. However, it only provides simulation data analysis and lacks real data analysis.

2. It has a good algorithmic design with sound intuition. The index-free cyclic exploration uses random pointer initialization ensures that with constant probability, two players' cyclic sweeps are offset, yielding a $1/K^2$ sampling rate guarantee without any communication.  But the $O(T^{1-1/K} \log{T})$ regret bound for Way-SE-2S is quite weak and its tightness is entirely open. For even moderate K (say K=10), this is $O(T^{0.9} \log{T})$, which is barely sublinear.
3. The presentation is good and originality is solid.

---

> ### Author Rebuttal · Authors · 2026-03-31
>
> We sincerely thank the reviewer for the thoughtful review and constructive feedback.
>
> Following the reviewer’s comments, we have carefully revised the paper with added discussions, experiments, and related-work comparisons, summarized below:
>
> **Firstly, on the $O(T^{1-1/K}\log T)$ regret bound and its tightness.**
> We actually conjecture that our upper bound is not too far from being tight. Due to lack of space, we ask the reviewer to kindly refer to our responses to reviewer VWjk on bullet point 1 there.
>
> Related to the discussion about the regret bound, we feel this is a good place to also reply to the reviewer's key question 3, which is about the design of $p(t)$.
> The tail-integral view indeed reflects a useful interpretation of our choice of $p(t)=t^{-1/K}$.
> Roughly speaking, $1-(1-p(t))^{L(t)} \approx L(t)p(t)$ is the probability for one player to re-explore once in $L(t)$ time. After the first re-exploration by any player, which happens with probability $p(t)$, $(L(t)p(t))^{K-1}$ is roughly the probability that the other $K-1$ players all re-explore in the next $L(t)$ interval. Thus $(L(t))^{K-1} (p(t))^K$ is roughly the probability that the deadlock can be resolved "at and after time t''. Moreover, because some players may have just arrived, the corresponding arm needs enough samples to learn about it, so we set $L(t)$ to be the sample lower-bound, roughly $\log t/\Delta_{min}^{2}$, as in the literature for static settings.
>
> Then what does it mean for the deadlock-resolution probability to be "large enough"? We require that for any T, there exists a time length $\tau$ such that the good event happens at least once, which corresponds to the integral from $T$ to $T+\tau$ being at least 1. $p(t) = t^{-1/K}$ is a natural choice as then $(p(t))^K = 1/t$. Moreover, if $p(t)$ is smaller than this by a polynomial factor, then the integral can't be $\geq 1$ even with the extra $(L(t))^{K-1}$ term. $p(t)$ can be smaller than this by some poly-logarithmic factor when $K\geq 2$, which we now clarify in Section 5 of the paper.
> In general, we have elaborated on the parameter choices used in our analysis in Section 5.
>
> **Secondly, on real data analysis.** Although most related studies we noticed were based on simulations,
> we think it's a great idea to incorporate a real-data analysis in the work, and we noticed that the 2021 JMLR paper suggested by the reviewer considered a relevant dataset. We are currently investigating this and other datasets in detail, and we will try to incorporate an analysis of them under our model (unfortunately in the limited rebuttal time we couldn't finish this part, but we definitely plan to do so).
>
> **On the 5 Key Questions.**
>
> Q1: $\Delta_{min}^2$ and lower bound
>
> For Way-SE, due to a lack of space, we ask the reviewer to kindly refer to our responses to reviewer VWjk on bullet points 1 and 3 there.
>
> For Way-SE-2S, $\Delta_{min}^{-2}$ appears as a sample lower-bound as discussed above. The difference from Way-SE is that the latter doesn't depend on $L(t)$. Unless there is a different way to measure when the arm has learnt about a player, we doubt the dependence on $\Delta_{min}^{-2}$ can be further improved.
>
> Q2: The optimal $K$ consecutive rounds
>
> The number of consecutive rounds is set to be the number of arms $K$. This is to argue that, if a player is rejected by an arm for $K$ times, then it's not because the shift of the other $K-1$ players, but must be because the arm is consistently occupied by a player whom it prefers. For Way-SE, this means this arm cannot be in the player’s POSM and can be eliminated.
> For Way-SE-2S, the player will temporarily eliminate this arm and may re-explore it later.
>
> Q3: Exploration rate $p(t)$
>
> See the discussion above on the regret bound and its tightness.
>
> Q4: On $T_{stable}$
>
> Great question, and we can accommodate this setting to some extent.
> Indeed, as the interval between the arrival/departure events grows larger as $T$ grows, the events will become sufficiently rare. Then, after each event, the system can pretend this is the last one (thus with a finite $T_{stable}$), and the algorithm will converge as long as the convergence speed is faster than the event speed. Currently, the convergence is roughly the time needed to resolve the deadlock, which is about $T/polylog(T)$. Thus, if the event interval is $T/polylog(T)$ and is larger than the convergence time, then our algorithm applies without any change. A $T^\alpha$ event interval is shorter than the convergence time, thus the algorithm may not converge. Further improving the convergence speed is left for future studies. We have added discussions about this in Section 7.
>
> Q5: More related work
>
> We thank the reviewer for the pointers. We have integrated a discussion about them in Sections 1 and 2. They are very helpful for better positioning our work.

---

> > ### Author Rebuttal · Reviewer_QpYU · 2026-04-03
> >
> > I would like to thank the authors for their detailed responses. I am inclined to maintain my current score.

---

> > > ### Author Response · Authors · 2026-04-08
> > >
> > > Thank you for the acknowledgment.

---

### Official Review · Reviewer_VWjk · 2026-03-16

**Soundness:** 3
**Presentation:** 2
**Significance:** 2
**Originality:** 3
**Overall Recommendation:** 3
**Confidence:** 3

**Summary:**

This paper studies decentralized bandit learning in two-sided dynamic matching markets where players arrive and depart at arbitrary times without access to a global clock. The authors propose two algorithms: Way-SE for one-sided learning with arrival-only dynamics, achieving O(K^2 log T / Δ^2_min) regret, and Way-SE-2S for the fully decentralized two-sided learning setting with both arrivals and departures, achieving O(T^{(K-1)/K} log T) regret. The key technical contributions include an index-free cyclic exploration mechanism for implicit coordination without synchronization, and a Restless Ping mechanism for detecting and adapting to structural shifts in the stable matching caused by player departures.

**Compliance With Llm Reviewing Policy:**

Affirmed.

**Key Questions For Authors:**

Same as weakness.

**Limitations:**

The paper acknowledges the lack of lower bounds and mentions incentive compatibility and reward non-stationarity as future directions. However, the limitation regarding the known Δ_min assumption and the practical scalability to large K deserve more explicit discussion.

**Strengths And Weaknesses:**

**Strengths:**
1. The paper addresses a genuinely important open problem in the matching bandits literature. The dynamic, uncoordinated setting where players lack a global clock and may arrive/depart arbitrarily is well-motivated by real-world applications, and prior work has not provided theoretical guarantees in this regime. The problem formulation is clean and the distinction between Setting I (one-sided, arrivals only) and Setting II (two-sided, arrivals and departures) provides a natural progression of difficulty.
2. The index-free cyclic exploration mechanism is a clever and elegant solution to the coordination problem. The idea that players with randomly offset pointers sweeping cyclically through arms can implicitly avoid persistent collisions — combined with the K-round waiting timeout as a discriminator for permanent blocking — is technically appealing and well-analyzed. Lemma A.6 establishing the necessity and sufficiency of the timeout condition is a particularly clean result.
3. The Restless Ping mechanism for the two-sided setting is a novel contribution that addresses the fundamental challenge that elimination strategies fail when the stable matching structure shifts due to departures.

**Weaknesses:**
1. The regret bounds provided in the paper are weak. The bound for Way-SE has a Δ^2_min dependence which is not ideal. But the regret bound O(T^{(K-1)/K} log T) for Way-SE-2S is a very weak bound. While the authors acknowledge this is due to the persistent exploration needed to detect departures, there is no lower bound analysis to indicate whether this rate is tight. Without a matching lower bound, it is unclear whether the polynomial dependence on T is inherent to the dynamic two-sided setting or an artifact of the algorithmic design.
2. The presentation of the two-sided algorithm (Algorithm 3 and its subroutines in Algorithms 4-6) is quite dense and difficult to follow. The interplay between the four arm sets (S_i, P_i, O_i, D_i), the mode switching logic, the timer mechanism, and the Restless Ping protocol involves many moving parts. A clearer high-level walkthrough with a concrete small example tracing the algorithm's execution through an arrival and departure event would significantly improve readability.
3. The assumption that Δ_min is known as input to Algorithm 3 (Way-SE-2S) is quite restrictive and somewhat at odds with the decentralized, information-limited spirit of the setting.
4. The experimental setup uses a relatively small number of arms (K=6 for two-sided, K=15 for one-sided) and a specific utility generation model with Δ_min = 0.01. It would be valuable to see how the algorithms scale with K, particularly since the regret exponent (K-1)/K depends directly on it, and to test sensitivity to Δ_min.
5. Several proof steps in the appendix rely on asymptotic arguments (e.g., "for sufficiently large t") without explicitly tracking the constants or specifying the required regime. For instance, Theorem B.4 defines T_alg with expressions involving N, K, and Δ_min, but the dependence is complex and the practical magnitude of these thresholds is not discussed.

---

> ### Author Rebuttal · Authors · 2026-03-31
>
> We sincerely thank the reviewer for recognizing the importance of the problem and the novelty of our algorithms. Following the 5 concerns raised by the reviewer, we have carefully revised the corresponding parts of the paper, with added discussions and experiments. We summarize our responses below:
>
> 1: The regret bounds
>
> For Way-SE, the $\Delta_{\min}^{-2}$ dependence is a consequence of the player needing to distinguish persistent preference inferiority from transient mismatches within our asynchronous environment.
> For algorithms in a static setting, once an arm is confidently suboptimal, it can be removed early, and its regret is naturally charged at its own gap, yielding the usual $1/\Delta$ regret. However, in our asynchronous model, a rejection is ambiguous: it may indicate true inferiority, but it may also arise from transient effects such as temporary occupation or collisions. Therefore, before an arm can be safely eliminated, the player must distinguish the two cases. Our current analysis thus uses a worst-case confidence scale tied to $\Delta_{\min}$, rather than arm-local gap charging, leading to the $\Delta_{\min}^{-2}$ dependence. We have added these discussions to Section 4. Similar to the paradigm in the static setting, which started from $\Delta_{min}^{-2}$ and later improved to $\Delta_{min}^{-1}$, we believe the quadratic dependence may motivate future studies and eventually be improved.
>
> For Way-SE-2S, we agree with the reviewer that it would be great to have a lower bound for this challenging problem.
> However, given that this is the very first time the setting is studied, and given the technical complexity that our results already have, we feel it's better to leave the lower bound as a separate study. Nonetheless, we actually conjecture that our upper bound is not too far from being tight, and we provide some intuitions on why this may be the case.
> Firstly, because of the dynamic departure, the player must maintain a probability p of random re-exploration, or the learning procedure won't converge.
> Secondly, a bad case caused by the departure of a player and the arrival of a new player is that all remaining K players are not at their optimal matching, and they actually form a Deadlock of length K. Such a Deadlock can only be resolved by all K players simultaneously moving up to better arms, which requires their re-explorations to all overlap. Thus, p must be large enough so that $p^K$ is not too small. Simply by having such a large p, the regret of each player, which is at least $pT$, seems to have a polynomial dependence on T, such as $T^{\frac{K-1}{K}}$.
>
> 2: The readability of our algorithms
>
> We have added a high-level flow diagram for Algorithm 3 in Appendix C, including the interplay between the different pieces mentioned in the review. We also added a concrete small example to track the algorithm execution in Appendix C. Besides, we have added discussions surrounding them to help the reader understand how different parts connect. We hope those new materials will improve our paper's readability.
>
> 3: About $\Delta_{\min}$.
>
> $\Delta_{min}$ is used because the algorithm uses a separation horizon whose length depends on the $\Delta$'s of individual players and arms. So we took the minimum of them in the worst-case analysis, as well as in the algorithm's construction. The information about $\Delta_{min}$, however, doesn't need to be so accurate. Firstly, any lower bound on it would work. Secondly, in implementation, we could replace the term $L(t)$ in Algorithm 4 by a player-arm specific $L_{i,k}(t)$. In this way, the algorithm doesn't need the global $\Delta_{min}$. It can decide which arm and player's information to use on the run. The player only needs to know $\Delta_{k}^{(a)}$ for some specific arms where $L_{i,k}(t)$ needs to be explicitly computed, which is more in line with the decentralized and uncoordinated setting.
> This doesn't change our worst-case analysis, so we didn't do it in the paper. We have added this discussion to Section 7.
>
> 4: Experimental Setup
>
> Section 6 now includes experiments with $K=100$ and 200, and with $\Delta_{\min}$ from 0.001, 0.002, and 0.0005.
> Discussions about the effects of those parameters are also added, where the effects are highly consistent with the analysis. The sensitivity of $\Delta_{min}$ is actually better than $\Delta_{min}^{-2}$ as the analysis is an upper bound.
>
> 5: Asymptotic arguments
>
> The asymptotic argument is used mainly because those quantities have no closed-form solution. To give a more concrete sense of them, we have added discussions in corresponding parts of the Appendix for the quantities where "sufficiently large" statements are used.
> For the quantity $T_{alg}$, we have added figures to plot its practical magnitude and its relationship with its parameters.

---

> > ### Author Rebuttal · Reviewer_VWjk · 2026-04-05
> >
> > I am still not convinced that the dependence on \Delta_{\min}^{-2} is needed and that $T^{\frac{K-1}{K}}$ bound is close to tight. It would be nice to have lower bounds to make these claims.

---

> > > ### Author Response · Authors · 2026-04-08
> > >
> > > We acknowledge the reviewer's comments.
> > > We agree that a matching lower bound would be very valuable for this hard problem, and we will continue working on it.
> > > That said, since this is, to our best knowledge, the first theoretical study of this challenging setting, we feel it is somewhat too much to require a matching lower bound as a component of the present paper.
> > > The upper-bound alone is already a highly non-trivial technical result and is a significant first step in this direction.

---

### Official Review · Reviewer_w42R · 2026-03-19

**Soundness:** 2
**Presentation:** 3
**Significance:** 3
**Originality:** 3
**Overall Recommendation:** 4
**Confidence:** 2

**Summary:**

Authors consider two-sided matching markets where players learn their preferences through iterative interactions. In this work authors study algorithms for dynamic environment, where agents could freely enter and leave the market. The key contribution is two algorithms (one for one-sided learning with arrival only players and one for two-sided learning, i.e. for main problem setup) supported by technical analysis.

**Compliance With Llm Reviewing Policy:**

Affirmed.

**Key Questions For Authors:**

Questions [pls see S/W part]:

1] Why can we use algorithms from restless bandits in this setup?

2] The regret rates include \triangle_{min}. Can we get more refined regret rates, that depends on combination of \triangle_i?

**Limitations:**

Yes

**Strengths And Weaknesses:**

Paper with good presentation, limited but important problem setup, requires more experiments and some clarifications.

Soundness: The submission includes theoretical and experimental results.
The theoretical part is technically correct (at least the rates are expected and I didn't find any mistakes in proofs in the limited time).
The experimental results from my perspective is lacking:
- No benchmarking. I understand the claim, that there are no algorithms that do not utilize global clock, and hence there are nothing to compare (at least no algorithm , that is designed for this setup). But currently it is not clear (for me) why one could not simple run some bandits algorithms locally for each player. At least to see how it would behave. There is a claim about conflicts, but why we can't handle them by using your Algorithm 6.
- Small setups. All experiments consider small number of arms: 6, 15. This limit our ability to understand how algorithm performance scales with large number of arms. At least in matching markets one usually expect K to be in range from 10^3 (small dating services) to 10^8 (marketplaces and social networks). So I think it is reasonable to expect experiments that shows how algorithm scales with increasing number of arms, at least up to K~10^2.

Presentation: I think the submission is clearly written (at least it was. easy to follow, understandable, good structured).

Significance: From my perspective, the original problem statement regarding players who learn their preferences in matching setting has limited significance. But still they have their share in applications such as recommended systems in marketplaces for users without specific needs.
On the other hand, I think that authors work in dynamic setup, where users come and go freely is a significant step toward practical problem setups.

Currently the main metric that is studied is individual regret for a player. The main theorems (4.1 and 5.1) bounds individual regret for any player. On the other hand, while reading paper intro and motivation, I would expect some kind of \textit{system} metric.
Authors mention, that "As typical in the literature, the objective of our learning algorithms is to minimize each player’s cumulative regret". But I think this choice of metrics, actually, reduce the significance.

Originality: It is hard for me to estimate the originality of the work, since do not specialize in this topic.
The algorithmic part, at least its bandits core, is more or less standard. Regret rates are also standard. But I think that is not a stopper, since the main novelty should come from dynamic arrivals of new players.

---

> ### Author Rebuttal · Authors · 2026-03-31
>
> We sincerely thank the reviewer for the positive feedback and the constructive comments. We have taken care of the comments carefully in the new version of our paper, with expanded discussions and experiments. We summarize our responses below:
>
> For the experimental results:
>
> 1: No benchmarking
>
> In Sections 6 and 7, we have added discussions and experiments on several existing bandit algorithms that rely on a global clock and static participants as a benchmark for our setting. We show that under our more challenging setting, they do not converge to player-optimal stable matching (POSM), and their cumulative regrets are at least linear.
> We don't know how to easily combine our Algorithm 6 with those algorithms to make them work, as Algorithm 6 is designed as a subroutine of the whole algorithm and needs input parameters from other parts, such as $c_{wait}$, $a_{held}$, $timer$, etc. If we revise those simple bandit algorithms to work around Algorithm 6's requirements, then essentially, we will get back to our entire algorithm. Indeed, conflict resolution is not only by Algorithm 6, and the entire algorithm should be aware of it and work around it. The core difficulty is designing exploration and timing rules so that decentralized players do not remain trapped in long conflict patterns when no global clock coordinates them. This is the main challenge of the setting and what our algorithm targets.
>
> 2: Small setups
>
> We have added in Section 6 experiments with $K=100$ and $K=200$ to show how our algorithm behaves as $K$ increases. Although existing decentralized matching-bandit studies typically evaluate small values of $K$ (mostly less than 20), we agree with the reviewer that doing larger experiments speaks more directly to the scalability of our algorithm. And we thank the reviewer for suggesting this.
>
> 3: System metric
>
> We get the confusion about the regret being the main metric. Indeed, the common system goal in related studies is to eventually converge to POSM of the market. The studies try to minimize the accumulative regret under the prerequisite that the POSM is achieved, which is a larger social goal. We have added in Sections 3, 4 and 5 more discussions about the role of POSM, and Theorems 4.1 and 5.1 now highlight that the algorithm converges to POSM.
>
> Q1: Restless bandits
>
> We realize that the term "restless" is overloaded in this case. We neither study the restless-bandit problem nor import a restless-bandit algorithm. Our bandit model is still the standard model, and "Restless Ping" is only a name we use for a subroutine: it keeps a small amount of probing after the interaction appears near-stable so that a player can notice when a previously blocked arm becomes relevant again once others leave. To resolve this confusion, we have explicitly clarified this distinction in the Introduction and added a brief clarification in the Conclusion.
>
> Q2: More refined regret rates
>
> $\Delta_{min}$ appears because the algorithm uses a separation horizon whose length depends on the $\Delta$'s of individual players and arms. So we took the minimum of them all in the worst-case analysis. The information about $\Delta_{min}$, however, doesn't need to be so accurate. Firstly, any lower bound on it would work. Secondly, in implementation, we could replace the general term $L(t)$ in Algorithm 4 by a player-arm specific $L_{i,k}(t)$, with the algorithm taking $\Delta_{i}$ and $\Delta^{(a)}_{k}$ as extra inputs. In this way, the algorithm can decide which arm and player's information to use on the run, and even acquire such information only as needed. More discussions on this aspect have been added in Section 7. We thank the reviewer for raising this question and making us think about these aspects more clearly.

---

> > ### Author Rebuttal · Reviewer_w42R · 2026-04-01
> >
> > Thank you for your answer.
> >
> > 2) Could you pls share results on additional experiments with $K=100$ and $K=200$?
> >
> > 4) As other reviewers and myself mentioned, $\Delta_{min}$ gives a very weak bound. I would suggest to try and make instant-dependent regret bound with $\Delta_{I}$.
> >
> > I understand that it is hard to do 4) in given time, but 2) should not be a problem, right?

---

> > > ### Author Response · Authors · 2026-04-08
> > >
> > > Thank you again for the helpful suggestion. Firstly, as the reviewer pointed out, it's indeed hard to get to an explicit instant-dependent regret bound in the given time, even though the algorithm can be made instant-dependent and only use the $\Delta_i$s.
> > > More precisely, the algorithm can use $L_{i, k}(t)$ instead of $L(t)$, where
> > > $$
> > >     L_{i,k}(t) =\left\lceil \frac{48\log t}{\min(\Delta_i,\Delta_k^{(a)})^2} \right\rceil+1.
> > > $$
> > >
> > > Still, motivated by the reviews, after the rebuttal period we have been thinking about the regret bound, and we have improved it in the $\log T$ factor. In particular, the persistent exploration schedule can be sharpened from $p(t)=t^{-1/K}$ to $p(t)=\frac{1}{t^{1/K}(\log t)^{(K-2)/K}}$. Here the exponent for $\log t$ can be any positive value less than $(K-1)/K$, and we took $(K-2)/K$ in our analysis. This correspondingly improves the regret upper bound from $O\left(\frac{K T^{1-\frac{1}{K}}\log T}{\Delta^2_{min}}\right)$ to $O\left(\frac{K T^{1-\frac{1}{K}}(\log T)^{2/K}}{\Delta^2_{min}}\right)$.
> > >
> > > We have redone our experiments with respect to this new design. Below we provide two experiments as data tables, with $\Delta_{min}=0.0005$. One has $(K,N)=(100,20)$ and one has $(200,30)$. Here $N$ is the expected number of players used in Poisson processes for random entry/exit events.
> > >
> > > To present the long-horizon ($T=10^6$) within the space limit for our response, the data points in the tables are non-uniformly taken from our logs. Specifically, we extract dense points around the moments of shape changes, when players enter or exit, and when the regret transits between steep learning curves and flat trends. During long stable periods, we use sparser time intervals.
> > > The reviewer can plot the corresponding curves using the data points by tools such as Python, MATLAB, or Excel. They exhibit a similar behavior to those of small $K$s and are better than the worst-case regret as in our bound.
> > >
> > > Table 1: K = 100, N = 20
> > > |TimeStep($t$)|Accumulative Regret($R(t)$)|
> > > |-|-|
> > > |10|78.63|
> > > |510|1174.22|
> > > |2.51K|2667.49|
> > > |8.51K|7147.33|
> > > |15K|11993.03|
> > > |19.63K|15449.97|
> > > |20.51K|16107.02|
> > > |21.13K|16569.93|
> > > |21.63K|16947.84|
> > > |22.13K|17434.62|
> > > |23.63K|18554.58|
> > > |26.63K|20794.51|
> > > |31.63K|24527.71|
> > > |35K|27043.89|
> > > |40K|30777.10|
> > > |45K|34510.31|
> > > |50K|38243.51|
> > > |56.01K|42502.40|
> > > |58.01K|42502.40|
> > > |60.01K|42502.40|
> > > |66.01K|42502.40|
> > > |70K|42502.40|
> > > |75.23K|42502.40|
> > > |76.73K|42502.40|
> > > |77.23K|42502.40|
> > > |77.73K|42502.40|
> > > |79.23K|42502.40|
> > > |82.23K|42502.40|
> > > |87.23K|42502.40|
> > > |95K|42502.40|
> > > |100K|42561.40|
> > > |125K|42561.40|
> > > |150K|42619.80|
> > > |161.01K|42619.80|
> > > |163.01K|42809.65|
> > > |165.01K|44362.44|
> > > |171.01K|49020.80|
> > > |175K|52118.61|
> > > |183.01K|58337.53|
> > > |200K|71528.47|
> > > |218.51K|85835.86|
> > > |220.51K|85835.86|
> > > |222.51K|85835.86|
> > > |228.51K|85835.86|
> > > |240.51K|85835.86|
> > > |250K|85835.86|
> > > |275K|85835.86|
> > > |300K|85835.86|
> > > |400K|85835.86|
> > > |500K|85835.86|
> > > |600K|85835.86|
> > > |700K|85835.86|
> > > |800K|85835.86|
> > > |900K|85835.86|
> > > |928.51K|85835.86|
> > > |930.01K|85900.06|
> > > |930.51K|86024.75|
> > > |932.01K|86024.75|
> > > |932.51K|86024.75|
> > > |934.01K|86024.75|
> > > |938.51K|86024.75|
> > > |940.01K|86024.75|
> > > |950.51K|86024.75|
> > > |952.01K|86024.75|
> > > |1M|86024.75|
> > >
> > > Table 2: K = 200, N = 30
> > > |TimeStep($t$)|Accumulative Regret($R(t)$)|
> > > |-|-|
> > > |10|158.02|
> > > |510|4307.71|
> > > |2.51K|11897.66|
> > > |8.51K|34667.51|
> > > |15K|59296.89|
> > > |20.51K|80207.19|
> > > |25K|97246.63|
> > > |30K|116221.50|
> > > |35K|135196.37|
> > > |40K|154171.24|
> > > |45K|173146.11|
> > > |50K|192120.98|
> > > |56.01K|214708.67|
> > > |58.01K|214708.67|
> > > |60.01K|214708.67|
> > > |66.01K|214708.67|
> > > |68.23K|214708.67|
> > > |69.73K|214708.67|
> > > |70.23K|214708.67|
> > > |70.73K|214708.67|
> > > |72.23K|214708.67|
> > > |75.23K|214708.67|
> > > |78.01K|214708.67|
> > > |80.23K|214708.67|
> > > |83.57K|214708.67|
> > > |84.01K|214708.67|
> > > |85.07K|214708.67|
> > > |85.57K|214714.04|
> > > |86.01K|215272.60|
> > > |87.57K|216444.22|
> > > |88.01K|216774.68|
> > > |90.57K|218697.33|
> > > |94.01K|221280.90|
> > > |95.57K|222452.52|
> > > |100K|225779.62|
> > > |106.01K|230293.35|
> > > |125K|244555.56|
> > > |141.01K|256579.67|
> > > |143.01K|256961.19|
> > > |145.01K|256961.19|
> > > |151.01K|256961.19|
> > > |163.01K|256961.19|
> > > |175K|256961.19|
> > > |200K|256961.19|
> > > |225K|256961.19|
> > > |250K|256961.19|
> > > |275K|256961.19|
> > > |300K|256961.19|
> > > |400K|256961.19|
> > > |500K|256961.19|
> > > |600K|256961.19|
> > > |700K|256961.19|
> > > |800K|256961.19|
> > > |900K|256961.19|
> > > |1M|256961.19|

---

### Official Review · Reviewer_FkNX · 2026-03-26

**Soundness:** 3
**Presentation:** 3
**Significance:** 2
**Originality:** 3
**Overall Recommendation:** 4
**Confidence:** 2

**Summary:**

The authors study distributed, uncoordinated player arrivals in a dynamic two-sided market. The set of arms is fixed and publicly known, and there is a global stable time after which there are no more arrivals or departures. Arms and players each have strict preference rankings over the other side. At each (discrete) time step, each present player proposes to one arm, each arm gets a set of proposals and picks one player, and there is exactly 1 match eventually with stochastic rewards on both ends, and other players/arms get zero.

The authors start with the simplified one-sided setting, where only players need to learn their preferences, and arms have fixed, known preferences. The authors propose Way-SE, which maintains confidence bounds for each player-arm pair, and uses them to decide on whether to "exploit with elimination" or continue with "index-free cyclic exploration". The authors provide regret bounds for each player.

The authors then study the two-sided setting where both side learn their preferences, and proposed the player-side algorithm Way-SE-2S, which uses StateUpdate, Select, and Conflict subroutines. The authors provide regret bounds for each player.

Finally, the authors provide numerical simulation results to demonstrate the proposed algorithms.

**Compliance With Llm Reviewing Policy:**

Affirmed.

**Final Justification:**

The rebuttal partially addressed my concern around $\Delta_{\min}$ and the algorithm being centralized. The new discussion around the local version that the authors added (in the comment and in the revision) is helpful. Still, the algorithm is "gap-dependent" as it needs to know this minimum gap, a lower bound, or a local proxy of it, to achieve the regret guarantee based on current analysis. This remains a restrictive assumption. A gap-independent algorithm/variant or additional analysis on the necessity of this dependence will be very helpful.

For modeling choice and example, the "AI companies" example seems the most plausible compared to any other examples we came up with so far. I believe adding it to the next version will help justify the setting.

**Key Questions For Authors:**

My main question/concern is around the modeling choice.

In this paper, arms are entirely passive (they do not propose), run identical UCB-like algorithms upon buyer proposals, and make matching decisions. I find it difficult to fit this into any application, e.g., job-seeking, recommendation, ride-sharing, etc. All of them fundamentally deviate from this paper's settings: the platform (not arms) often makes the decisions that may or may not align with buyers' or arms' incentives; arms are very likely to have different objectives and be strategic, in which case a mechanism design framework may be necessary instead of optimizing buyer-side regrets; arms may also have arrival/departure intervals.

While I don't think the current paper needs to get to all of these, I find it necessary to get to 1-2 concrete modeling examples showing that this paper's mid-ground setting aligns with something genuinely plausible. An example might be allocation of public goods (schools, rationing, etc.), where "arms" are constantly present and make passive choices on proposers. However, for these, unknown preferences and a fully online learning setting both seem very off.

**Limitations:**

yes

**Strengths And Weaknesses:**

**Strengths**

The problem setting is genuinely challenging, and the authors provide a nontrivial algorithm and corresponding regret bounds.

Theorem 4.1, especially Lemma A.6 in its proof ($\geq K$ regections <==> "stable incumbent"), seems novel and nontrivial. This + local i.i.d. pointers are essential in establishing Theorem 4.1 without a global clock.

The Restless Ping / random exploration rate in Eq. (4) helps with the stable matching lattice argument while still ensure a sublinear regret.

**Weaknesses**

The algorithm seem to rely on $\Delta_{\min}$, as well as other parts of the regret behind the big-O term in Theorem 5.1 (essentially the Theorem 4.1 bound). For a very small $\Delta_{\min}$, Way-SE-2S + StateUpdate --> there will be a lot of pure exploration?

The modeling assumptions seem very synthetic and not aligned with the cited practical motivation in the introduction. See "Questions" for more details.

---

> ### Author Rebuttal · Authors · 2026-03-31
>
> We sincerely thank the reviewer for recognizing that the problem is challenging and the technical complexity of our algorithm.
> Following the reviewer’s comments, we have revised the paper with added discussions and clarifications, which are summarized below:
>
> Firstly, on $\Delta_{min}$ in the theorems. The reviewer is correct, and both Theorems 5.1 and 4.1 depend on it. We absorbed it in the statement of 5.1 to highlight the dependence on $T$, but kept it in the proofs. We have revised 5.1's statement to explicitly include $\Delta_{min}$, which is quadratic as in 4.1.
>
> On the modeling choice, which is the reviewer's key question:
>
> In application scenarios such as the job market, we consider companies as arms and applicants as players who send their applications to companies. For many jobs of a daily base, a player can only do one job at a time step (i.e., one day). The companies mainly have the job posted and examine the applications they receive ---like the more passive arm-role.
>
> Other job-market scenarios may be different, such as the high-profile headhunters would reach out to people more actively. In this case, we can flip the roles and consider headhunters as the players who propose. Indeed, high-profile professionals in a field are playing the arm-role, which exist long-term and may be passive in looking for positions.
>
> However, we certainly do not claim that our model captures all aspects of a matching market. For example, the strategic considerations of arms as mentioned by the reviewer could also exist and deserves careful studies on its own. But most of the related works we noticed in the online learning literature focus on the stable matching perspective and aim at minimizing the participants' accumulative regret, under the prerequisite that the learning algorithm converges to the player-optimal stable matching. This is the scenario where our model tries to remove several important assumptions, such as static players and a global clock for coordination. Although we are interested in the mechanism design perspective as well, and also arms' dynamic arrival and departure, they are out of scope for this study.
>
> We have added discussions in Sections 1 and 3 about the job-market scenarios in terms of applicants as players and companies as arms, as well as when headhunters as players and professionals as arms. In both cases, the players arrive and departure dynamically, and the arms are there long-term. We have also added in Section 7, when discussing other important directions, several different perspectives besides stable matching and regret minimizing. We hope those discussions better place our work in the literature.

---

> > ### Author Rebuttal · Reviewer_FkNX · 2026-04-04
> >
> > Thanks for the response. I still find it very hard to envision a concrete example that fits modeling choice. Consider "companies as arms and applicants as players who send their applications to companies": Under the paper's setting, companies are running the same UCB algorithms, and applicants send applications repeatedly; for each (candidate A, company B) pair, learning only happens on days when the A applies to B and gets accepted. And this has to happen many times, together with rejections, for the same pair, to accumulate data. This feels unrealistic and does not seem to capture any job market setting. One possible setting I could think of: highly seasonal and temporary work where many similar employers and workers interact frequently and repeatedly over time? For example, in summer tourism destinations, jobs may be posted / performed / paid strictly on a daily basis, and, for sone reason, each employer prefers making hiring decisions on a daily basis instead of hiring a worker for a fixed time period. In short, it is very hard to think of a "repeated learning"-friendly job market.
> >
> > Regarding $\Delta_{min}$ in Algorithms 3 and 4 (not only the theorems): It seems the response did not cover this concern? It is a restrictive assumption to have the algorithm use an unobservable / hard-to-observe global parameter (especially given the "passive arms" assumption, which naturally means that arms don't communicate with each other or players). While the key contributions of the paper remove the need of a global clock and observability / communication among arms and players, this global knowledge requirement seems to be a significant tradeoff.

---

> > > ### Author Response · Authors · 2026-04-08
> > >
> > > We sincerely thank the reviewer for the tourism example (highly seasonal and temporary daily work), which we found to be a good example and have incorporated it into the Introduction.
> > > Regarding the assumption that arms are passive, we just followed the typical setup in the related works, where arms are normally considered as accessing players based on UCB.
> > > To further illustrate why players undergo repeated applications and rejections with the same arms in reality, we believe batch-based crowdsourcing platforms such as Amazon Mechanical Turk also provide a good example. In this scenario, AI companies or research institutions (as arms) repeatedly release massive batches of similar data annotation tasks. Crowdsourcing workers (as players) constantly apply to take on these tasks. If a worker is "rejected", it simply indicates that the platform allocated the current batch to other workers based on their current estimated reliability (UCB). The worker is not permanently blocked and can continue to apply for new task batches posted by the same institution in the next hour. This high-frequency, asynchronous, repeated interaction scenario aligns well with our dynamic matching formulation.
> > >
> > > Regarding the dependence on $\Delta_{\min}$ in our algorithms, let us describe its role more precisely. In our current formulation, $\Delta_{\min}$ appears because the algorithm uses a separation horizon whose length depends on the $\Delta$'s of individual players and arms.
> > > So we took the minimum of them in the worst-case analysis, as well as in the algorithm's construction.
> > > In fact, the information about $\Delta_{\min}$ does not need to be accurate: any valid lower bound is sufficient, and using a smaller lower bound only makes the separation horizon more conservative.
> > > More importantly, the actual mechanism does not inherently depend on global knowledge.
> > > In implementation, we could replace the term $L(t)$ in Algorithm 4 by a player-arm specific $L_{i,k}(t)$. In this way, the algorithm doesn't need the global $\Delta_{min}$. It can decide which arm and player's information to use on the run. The player only needs to know $\Delta_{k}^{(a)}$ for some specific arms where $L_{i,k}(t)$ needs to be explicitly computed, which is more in-line with the decentralized and uncoordinated setting.
> > > We did not develop this local version in the paper because it does not improve the worst-case bound of our current analysis. Nevertheless, such a refinement is both natural and meaningful, and we have added this discussion to Section 7.

---

### Decision · Program_Chairs · 2026-04-30

**Decision:**

Accept (regular)

**Comment:**

This paper studies a challenging and practically relevant extension of matching bandits: dynamic two-sided markets with asynchronous arrivals and departures, no global clock, and decentralized learning. The paper proposes nontrivial algorithmic ideas, especially the index-free cyclic exploration mechanism and the Restless Ping component for adapting to departures, and provides the first sublinear-regret guarantee in this fully decentralized uncoordinated setting.

The paper is not without weaknesses. The regret guarantee in the two-sided setting is relatively weak, and the absence of a matching lower bound leaves open whether this dependence is inherent or mainly an artifact of the current analysis. Reviewers also raised reasonable concerns about the modeling assumptions and the strength of the motivating applications. The experiments are also somewhat limited in scale, though the rebuttal added useful larger-$K$ evidence and clarifications.

Overall, however, I lean positive. My main reason is that the paper appears to make a genuinely new theoretical step on a hard open problem, and the remaining concerns are mostly about the sharpness of the guarantees and the breadth of applicability rather than about core soundness. As a first theoretical treatment of this decentralized dynamic setting, I believe the contribution (just about) clears the bar for a weak accept.